# De novo inter-regional coactivations of preconfigured local ensembles support memory

Hiroyuki Miyawaki 📧 1✉ & Kenji Mizuseki 📧 1✉

Neuronal ensembles in the amygdala, ventral hippocampus, and prefrontal cortex are involved in fear memory; however, how inter-regional ensemble interactions support memory remains elusive. Using multi-regional large-scale electrophysiology in the aforementioned structures of fear-conditioned rats, we found that the local ensembles activated during fear memory acquisition are inter-regionally coactivated during the subsequent sleep period, which relied on brief bouts of fast network oscillations. During memory retrieval, the coactivations reappeared, together with fast oscillations. Coactivation-participating-ensembles were configured prior to memory acquisition in the amygdala and prefrontal cortex but developed through experience in the hippocampus. Our findings suggest that elements of a given memory are instantly encoded within various brain regions in a preconfigured manner, whereas hippocampal ensembles and the network for inter-regional integration of the distributed information develop in an experience-dependent manner to form a new memory, which is consistent with the hippocampal memory index hypothesis.

---

[1] Department of Physiology, Osaka City University Graduate School of Medicine, Asahimachi 1-4-3, Abeno-ku, Osaka 545-8585, Japan.
✉email: miyawaki.hiroyuki@med.osaka-cu.ac.jp; mizuseki.kenji@med.osaka-cu.ac.jp

Animals acquire memory through experience during wakefulness, and in the subsequent sleep period, the acquired labile memory is transformed into a stable form via a process called memory consolidation[1]. Cell ensembles within local circuits that were activated at the time of memory acquisition also become active during memory retrieval[2], implying that memory-encoding cell ensembles are maintained stably throughout the memory process. In contrast, memory-responsible brain regions shift with time[3], suggesting that the global circuit changes dynamically during memory consolidation. However, how memory-encoding ensembles interact inter-regionally during memory consolidation and retrieval remains unclear. Moreover, neuronal activity patterns during awake periods are spontaneously reactivated during the subsequent non-rapid eye movement sleep (NREM) in various brain regions[4–6], and this reactivation plays an essential role in memory consolidation[7–9]. However, it remains controversial whether sleep reactivation occurs synchronously[5,6,10–13] or independently[14] in different brain regions.

Ensemble reactivations in the dorsal hippocampus (dHPC) occur during short bouts of fast oscillations (110–200 Hz), known as sharp-wave ripples (SWRs)[8]. The co-occurrence of SWRs and cortical oscillatory events, such as cortical ripples (cRipples; 90–180 Hz), has been proposed as the inter-regional information transfer mechanism[15]. SWRs are also observed in the ventral hippocampus (vHPC). In the dHPC and vHPC, SWRs occur largely asynchronously, have distinct physiological properties[16], and affect activity in downstream regions differently[17]. Moreover, fast oscillations (90–180 Hz) are also observed in the amygdala, where the oscillatory events are referred to as high-frequency oscillations (HFOs)[18]. However, how fast network oscillations in various regions control ensemble reactivations and whether these oscillations regulate inter-regional communication remain unknown. Additionally, it has been proposed that cortical slow-waves (0.5–4 Hz) facilitate information transfer by coordinating various oscillatory events[1]. However, it remains unclear whether slow-waves support inter-regional ensemble communications.

Furthermore, it remains controversial whether an ensemble activity similar to that occurring during behaviour exists prior to experience[19–21] or not[22]. Moreover, whether inter-regional ensemble coordination exists prior to experience and whether the cells contributing to coordination are intrinsically distinct remain to be determined.

This study aimed to investigate the inter-regional interactions of local ensemble activities and elucidate their regulation mechanisms and physiological functions in the memory process using fear conditioning as a model. Fear memory involves the vHPC CA1 region (vCA1), basolateral nucleus of the amygdala (BLA), and prelimbic cortex (PL)[23]. Although these brain regions are anatomically interconnected, direct projection from the PL to the vCA1 is lacking[23]. Our analysis of simultaneous recordings in the vCA1, BLA, and PL suggests that elements of a fear memory are instantly encoded in preconfigured local ensembles and that de novo inter-regional ensemble coactivations bind these elements together and support memory retrieval.

## Results

### Simultaneous recording of neuronal activity from multiple single cells in the vCA1, BLA, and PL of fear-conditioned rats.
We performed simultaneous large-scale electrophysiological recordings in the vCA1, BLA, PL layer 5 (PL5), and adjacent regions (Fig. 1a, b and Supplementary Fig. 1) and examined local field potentials (LFPs) and 1220 well-isolated units (Supplementary Table 1) in 15 freely moving rats. Excitatory and inhibitory cells were classified based on spike transmission/suppression and

waveforms (Supplementary Fig. 2). Recordings were obtained continuously throughout baseline, conditioning, context-retention, cue-retention/extinction, retention-of-extinction, and homecage sessions that preceded, were interleaved, and followed the behavioural sessions (Fig. 1c, d). The proportion of time spent in freezing behaviour indicated that the rats had learned an association between cues and shocks and retrieved the association during the retention sessions (Fig. 1c and Supplementary Fig. 3). This study focused on neuronal activity during conditioning sessions, cue-retention/extinction sessions, and homecage sessions flanking conditioning sessions. Data from the context-retention sessions, retention-of-extinction sessions, and homecage sessions flanking retention-of-extinction sessions were excluded from further analysis, unless otherwise stated.

**Memory-encoding ensembles in different brain regions are synchronously reactivated during NREM sleep after fear conditioning.** To investigate whether memory-encoding ensembles in different brain regions interact, we identified neuronal ensembles, which reflect prominent cofiring of multiple neurons within a short time window (20 ms), in each brain region using an independent component analysis (ICA) of spike trains[24,25] during conditioning sessions (Supplementary Table 2). The member cells of each ensemble were defined as cells with the top five projection weights (Fig. 2a). First, using bin label shuffling (see "Methods" for details), we assessed whether the firing of each ensemble-participating-cell (i.e., a cell that was a member of at least one ensemble as determined during the conditioning session) was modulated by freezing behaviour/cue presentation. In the cue-retention/extinction sessions, the firing rates of most BLA, vCA1, and PL5 ensemble-participating-cells were modulated by freezing behaviour/cue presentation (Supplementary Fig. 4a). Next, we estimated the instantaneous ensemble activation strength, which reflects how cofiring patterns within each time bin are similar to those observed during behaviour[5,24,25], in both pre-conditioning (pre-cond) and post-conditioning (post-cond) homecage sessions (Fig. 2a, b). We also estimated the instantaneous activation strength of the ensembles in the cue-retention/extinction sessions and observed that most BLA, vCA1, and PL5 ensembles altered their activity in response to freezing behaviour/cue presentation (Supplementary Fig. 4b). These results suggest that most of the ICA-identified ensembles in the BLA, vCA1, and PL5 are involved in the fear memory.

Inter-regional interactions of the ensembles were assessed using cross-correlogram (CCG) analysis of instantaneous ensemble activation strength (Figs. 2c, d and 3a). We evaluated the significance of the actual CCG peaks/troughs based on the distribution of the maximum deflection of shuffled surrogates (Fig. 2c and Supplementary Table 3). The inter-regional synchronous ensemble activation during NREM was significantly enhanced after fear conditioning in BLA–PL5 and vCA1–PL5 pairs (Figs. 2c–e, and 3a–c). These coactivations were detected in individual rats with sufficient ensemble pairs examined (Supplementary Table 4). Regarding the proportion of BLA–PL5 and vCA1–PL5 pairs with significant CCG peaks or troughs, we confirmed the robustness of the results by repeating the analysis by excluding animals one by one (Supplementary Fig. 5a). Consistently, correlations between spike trains of BLA–PL5 and vCA1–PL5 cell pairs during conditioning sessions tended to correlate more strongly with those in post-cond NREM than in pre-cond NREM (Supplementary Fig. 5b). Although a small number of negatively correlated pairs were identified (Figs. 2d, e, 3a, b, and Supplementary Table 3), herein, we focused on positively correlated pairs. Henceforth the ensemble pairs displaying significant coactivation during post-cond NREM are referred to as coupled-ensemble-pairs, and the composing ensembles as coactivation-participating-ensembles (Supplementary Table 5).

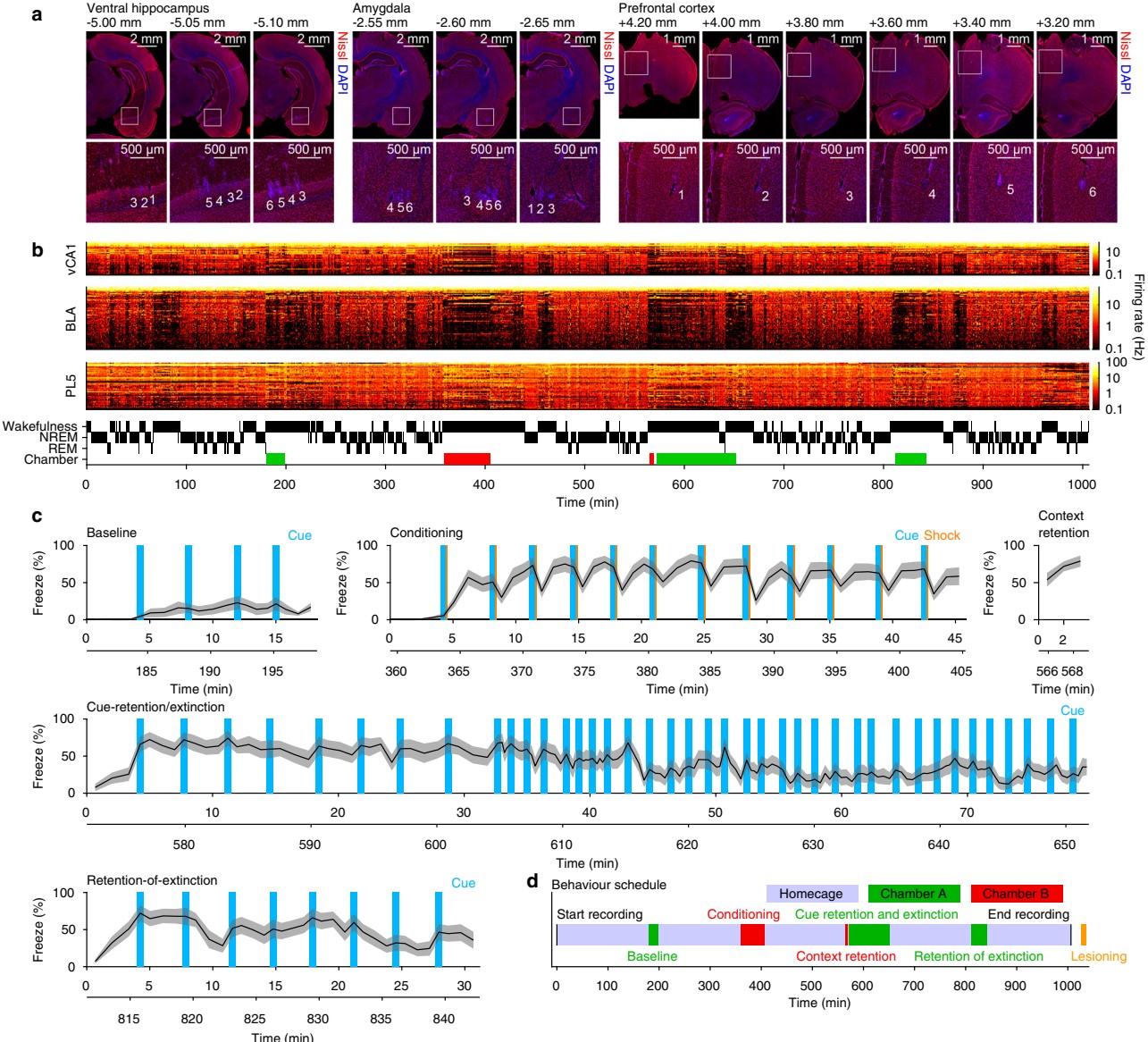

**Fig. 1 Multi-regional large-scale electrophysiological recordings on fear-conditioned rats. a** Electrode tip locations in a representative rat. The areas indicated by white squares in the top panels are shown in the bottom panels at higher magnification. The numbers indicate the shank indices within each probe. The A–P axis coordinate from the bregma is shown at the top-left corner of each micrograph. **b** Unit activity and hypnogram from the same rat. Each row represents the firing rates of individual units in 10 s bins ($n = 35$, 67, and 51 cells in the vCA1, BLA, and PL5, respectively). The periods of behavioural sessions in safe and shock chambers are indicated by green and red bands on the bottom of the bars, respectively. **c** Proportion of time spent in freezing in each behavioural session. The black lines and grey shaded areas indicate the mean and SEM, respectively ($n = 15$ rats). Time from the session onset is shown on the x-axes, and time from the beginning of the recording is superimposed beneath the x-axes, corresponding to the x-axis in (**d**). Source data are provided as a Source Data file. **d** Schedule of the experiments. We performed approximately 17 h continuous recording during which the animals underwent behavioural sessions (highlighted with green and red), and immediately after the recording, the animals were anesthetised, and micro-lesions were made to mark the recording sites. Electrophysiological recording, behavioural sessions, and micro-lesions were performed on the same day. The black vertical lines indicate the start and end times of the recordings. Neuronal activities obtained from context-retention sessions, retention-of-extinction sessions, and the homecage sessions flanking retention-of-extinction sessions were not analysed in this study, unless otherwise stated.

A few coupled-ensemble-pairs in vCA1–BLA (4 of 257 pairs) and other region pairs (Supplementary Table 3) were also identified; however, the change of coactivation between pre- and post-cond NREM was not significant at the population level (Fig. 3b, c, Supplementary Fig. 6, and Supplementary Table 3).

In contrast to the findings obtained during NREM, the proportions of coactivated ensemble pairs did not significantly differ between pre- and post-cond rapid eye movement (REM) sleep (Supplementary Fig. 7a–c). Although REM was shorter than NREM in both pre- and post-cond homecage sessions [REM/

NREM durations were $16.7 \pm 1.7/81.5 \pm 3.3$ min and $12.4 \pm 2.0/67.8 \pm 6.0$ min in pre- and post-cond homecage sessions, respectively, mean ± standard error of the mean (SEM), $n = 15$ rats], the absence of coactivation during REM cannot be attributed solely to its short duration, as BLA–PL5 ensemble coactivation during NREM remained significant when the NREM duration analysed was matched to the REM duration (Supplementary Fig. 7d–f). Furthermore, the same analysis on vCA1–PL5 ensemble pairs showed an increasing trend for the coactivation (Supplementary Fig. 7d–f).

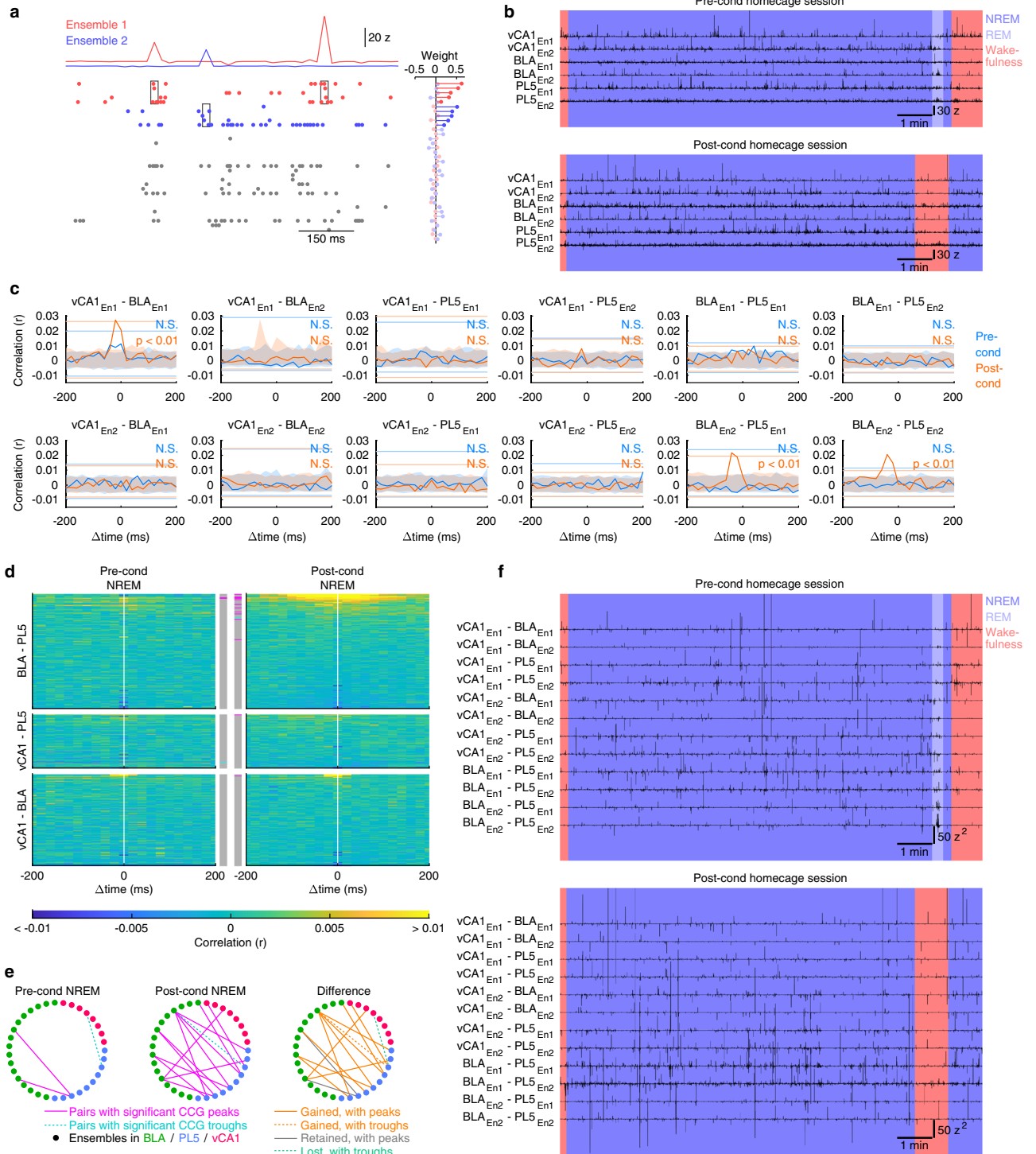

The distribution of CCG peak time among the BLA–PL5 coupled-ensemble-pairs demonstrated that reactivation in PL5 ensembles tended to follow that in BLA (18.4 ± 6.6 ms, mean ± SEM; $n = 38$ pairs) ensembles (Fig. 3d). Considering that monosynaptic transmission latencies from the BLA to the prefrontal cortex are approximately 7 ms[26] and that a brain region could require tens of milliseconds to activate local ensembles through local circuit interactions in response to direct inputs received from an upstream region[27], the temporal delays between ensemble activations suggest that direct projections from the BLA to the PL5 support the inter-regional coactivations. The distribution of CCG peak time among coupled vCA1–PL5

ensemble pairs (8.3 ± 8.3 ms, mean ± SEM; $n = 12$ pairs) is also consistent with the notion that direct projections from the vCA1 to the PL5 play a role in inter-regional coactivation, considering that monosynaptic transmission latencies from the vHPC to the prefrontal cortex are approximately 15 ms[28] and anatomical connectivity from vHPC to PL is unidirectional[23].

Because emotionally arousing experiences are remembered more strongly than neutral ones[29], we hypothesised that a robust aversive experience enhances the coactivation of inter-regional ensembles more prominently than a neutral experience. Thus, to determine whether ensembles that are active during the baseline session are also inter-regionally coactivated after the baseline

**Fig. 2 Representative examples of ensemble reactivations and inter-regional coactivations. a** A representative example of a vCA1 spike raster plot from an NREM epoch of the rat presented in Fig. 1a, b. The weight of the projection vector of two example ensembles (group of cofiring neurons in the same region identified in the conditioning session) is shown on the right, where the member cells (top five cells with the largest weight of the projection vector) are highlighted with vivid colours. Instantaneous ensemble activation strength traces detected from the spike trains are shown at the top. On the raster plots, the spikes from the member cells are highlighted with colours, and the spikes of the member cells at significant ensemble activation events are highlighted with boxes. **b** Instantaneous activation strength of local ensembles in pre- and post-cond homecage sessions detected in the same rat. Two representative ensembles (En1 and En2) from each region are shown. The background colours indicate behavioural states. Ensembles from vCA1 correspond to ensembles 1 and 2 in (**a**). **c** Inter-regional CCGs of instantaneous activation strength traces presented in (**b**) during pre- and post-cond NREM. The significance of the peak for each CCG (determined with 99% CI of peak/trough within ±100 ms in shuffled surrogates; shown by horizontal lines) is superimposed on the top right corner. Coloured shades indicate 99% CI of shuffled surrogates at each time point. **d** All inter-regional CCGs of the instantaneous activation strength of ensembles obtained from the same rat ($n = 170$, 80, and 136 for BLA–PL5, vCA1–PL5, and vCA1–BLA ensemble pairs, respectively). Each row represents the CCG for one inter-regional ensemble pair. Ensemble pairs are sorted based on the peak heights of CCGs during post-cond NREM. The coloured bars in the middle indicate pairs with significant peaks (magenta) or troughs (cyan), as tested using shuffling analysis ($p < 0.01$; see "Methods"). **e** Diagrams representing coactivation networks among ensembles obtained from the same rat. The solid and dashed lines reflect ensemble pairs with significant CCG peaks or troughs, respectively. The right panel illustrates changes from pre- to post-cond NREM. **f** Instantaneous coactivation strength of the inter-regional ensemble pairs presented in (**b**).

session (in which rats are exposed to novel environments and tones without electrical shocks), we identified neuronal ensembles during the baseline session using the same method used for ensemble identification in the conditioning session. We found that the proportion of significantly coactivated ensemble pairs changed between the pre- and post-baseline NREM sessions (Supplementary Fig. 8a, b); however, CCG peak height of ensemble pairs did not change significantly at the population level (Supplementary Fig. 8c). These results indicate that a subset of neuronal ensembles becomes coactivated across brain regions after novel experiences. Notably, the extent of change in the coactivation caused through the conditioning sessions was significantly different from that through the baseline sessions (Supplementary Fig. 8d, e), suggesting that emotional experience induces larger changes in inter-regional coactivation.

Significant changes in ensemble coactivation between pre- and post-cond NREM were observed only in BLA–PL5 pairs (examined in seven rats) and vCA1–PL5 pairs (examined in seven rats). Thus, further analysis of coactivations was restricted to these region pairs. The behaviour of the subsets of rats with implants in the BLA and PL5 or in the vCA1 and PL5 did not differ significantly from the average behaviour of all the examined rats (Supplementary Fig. 3). The proportion of coupled-ensemble-pairs in BLA–PL5 pairs was approximately three times as large as that in vCA1–PL5 pairs (Supplementary Table 3). In both region pairs, the number of analysed pairs (product of the numbers of simultaneously recorded local ensembles in the involved regions) increased linearly with the product of the numbers of simultaneously recorded neurons in the involved regions with similar slopes (Supplementary Fig. 9). Alternatively, the slopes of regression lines between the numbers of analysed and coupled-ensemble-pairs varied between BLA–PL5 and vCA1–PL5 pairs (Supplementary Fig. 9). These observations suggest that the difference in the proportion of identified coupled-ensemble-pairs across the region pairs cannot be fully explained by the difference in the number of recorded neurons or analysed ensemble pairs.

**Amygdalar HFOs, hippocampal SWRs, and prelimbic cRipples contribute to inter-regional ensemble coactivation during NREM.** Next, we searched for network activity patterns during which the ensemble coactivation among the vCA1, BLA, and PL5 occurred. Visual inspection suggested that BLA–PL5 coactivation accompanies fast (~130 Hz) oscillations in the BLA LFP (Fig. 4a), which are known as amygdalar HFOs[18] (Supplementary Fig. 10). The periods of HFOs partially overlapped with those of SWRs and cRipples (Supplementary Fig. 10c, d). HFOs strongly

modulated the firings of cells in the amygdala and other regions (Supplementary Fig. 11a) and enhanced ensemble activations in the BLA (Supplementary Fig. 11b). Similarly, the strength of vCA1 and PL5 ensemble activation transiently increased at SWR and cRipple peaks, respectively (Supplementary Fig. 11b). Moreover, vCA1–PL5 ensemble coactivations frequently coincided with SWRs (Fig. 4a), whereas both BLA–PL5 and vCA1–PL5 ensemble coactivations frequently co-occurred with cRipples (Fig. 4a).

To quantify these observations, we calculated the instantaneous coactivation strength (Fig. 2f), detected individual coactivation events, and calculated the ensemble-coactivation-triggered average of LFP wavelet power (Fig. 4b). We detected a strong peak of ~130 Hz in BLA wavelet power at BLA–PL5 ensemble coactivations, which reflects a coincidence between BLA–PL5 ensemble coactivations and HFOs. At vCA1–PL5 ensemble coactivations, strong peaks of ~150 Hz and broad peaks of ~15 Hz were observed in hippocampal wavelet power, which is consistent with ripples and accompanying sharp-waves[8] in the vHPC. Additionally, we detected peaks of ~130 Hz in the PL5 LFP at BLA–PL5 and vCA1–PL5 ensemble coactivations (Fig. 4b). Consistent with this, coactivation events transiently increased during cRipples (Fig. 4c, d). Furthermore, BLA–PL5 and vCA1–PL5 ensemble coactivations were transiently enhanced during HFOs and SWRs, respectively (Fig. 4c, d). These enhancements were more prominent in the post- than pre-cond NREM (Fig. 4e). In contrast, the occurrence rates of HFOs and SWRs did not change significantly, and those of cRipples increased moderately between pre- and post-cond NREM (Supplementary Fig. 12), suggesting that the development of inter-regional ensemble coactivation (Figs. 2 and 3) can be attributed to the enhancement of ensemble coactivations during fast oscillations, rather than to an increase in fast oscillation event rates.

When CCG analysis was restricted within fast oscillatory events, the CCG peaks were significantly higher than those across the entire post-cond NREM (Fig. 4f). Consistently, CCG analysis of instantaneous ensemble activation strength outside fast oscillatory events (Supplementary Fig. 13) revealed the contribution of fast oscillations in the coactivations. We observed that 84.2% of BLA–PL5 and 83.3% of vCA1–PL5 coupled-ensemble-pairs displayed a significant CCG peak reduction when the time bins that contained HFOs and SWRs were excluded, respectively (Fig. 4g). Similarly, the exclusion of cRipple-containing bins significantly decreased CCG peaks in 81.6% of BLA–PL5 and 75.0% of vCA1–PL5 coupled-ensemble-pairs (Fig. 4g). Furthermore, the CCG peaks were no longer significant after excluding HFO-/SWR-containing bins in 50.0% of BLA–PL5 and 66.7% of

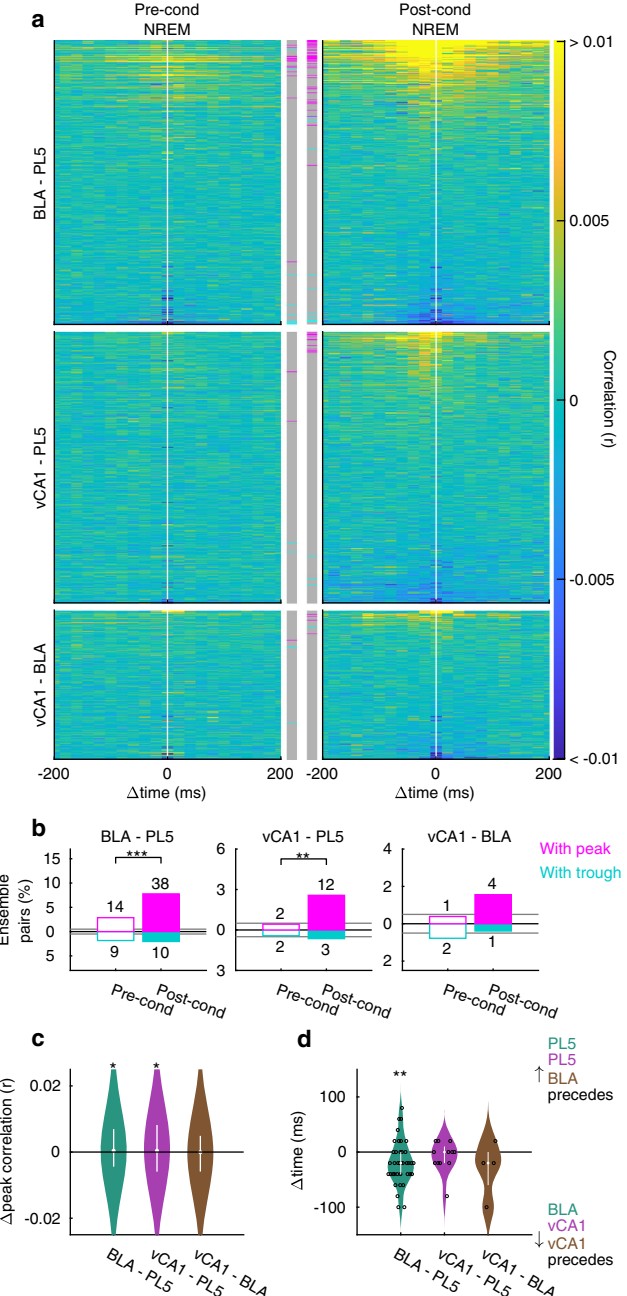

**Fig. 3 Memory-encoding ensembles in different brain regions are synchronously reactivated during non-REM sleep after fear conditioning.** **a** Inter-regional CCGs of the instantaneous activation strength of local ensembles, as in Fig. 2d, but for data pooled across rats. **b** The proportions of ensemble pairs with significant peaks/troughs among the pairs shown in (**a**). ***$p < 0.001$, **$p < 0.01$, Fisher's exact test. Horizontal bars indicate chance levels (0.5% for each). The number of ensemble pairs is superimposed on each bar. **c** Violin plots indicating the changes in CCG peak heights from pre- to post-cond NREM for all ensemble pairs that are shown in (**a**). White dots and lines represent median and upper/lower quartiles, respectively. *$p < 0.05$, WSR-test. **d** Violin plots of peak time on the CCGs of coupled-ensemble-pairs. White dots and lines represent median and upper/lower quartiles, respectively. **$p < 0.01$, WSR-test. Open circles indicate individual data points (small horizontal jitters are added for visualisation). The numbers of tested pairs and those in each rat are summarised in Supplementary Table 4. In addition, detailed statistics are shown in Supplementary Data 1. Source data are provided as a Source Data file.

vCA1–PL5 coupled-ensemble-pairs, respectively (Fig. 4h). The removal of cRipple-containing bins resulted in a loss of significant peaks in 23.7% of BLA–PL5 and 33.3% of vCA1–PL5 coupled-ensemble-pairs (Fig. 4h). These results indicate that amygdalar HFOs and hippocampal SWRs contribute to BLA–PL5 and vCA1–PL5 ensemble coactivations during NREM, respectively, and that cRipples contribute to ensemble coactivations in both region pairs.

**Cortical slow-waves coordinate inter-regional ensemble coactivation during NREM.** NREM sleep is characterised by slow oscillations (<1 Hz) and delta waves (1–4 Hz), which are associated with the alternation of silent and active periods of large cortical neuronal populations[30,31]. The peaks of slow-waves recorded in the deep layers of the neocortex were concomitant with generalised silent periods, known as DOWN states, in the neocortex (Fig. 5a)[30,32,33]. PL slow-waves also strongly modulated neuronal firing in the BLA but only weakly modulated neuronal firing in the vCA1 (Supplementary Fig. 14a). We observed that the occurrence rates of HFOs peaked at $200 \pm 11$ ms (mean $\pm$ SEM, $n = 15$ rats) prior to PL slow-wave peaks in post-cond NREM (Fig. 5b). Consistent with a tight relationship with HFOs (Fig. 4), BLA–PL5 ensemble coactivations were also enhanced at a similar time ($241 \pm 14$ ms prior to slow-wave peaks in post-cond NREM, mean $\pm$ SEM, $n = 38$ pairs; Fig. 5c). These time gaps were longer than the typical interval between slow-wave onset to peak ($113 \pm 4.1$ ms in post-cond NREM, mean $\pm$ SEM, $n = 15$ rats) but shorter than that between slow-wave offset and the next slow-wave peak ($1,317 \pm 49$ ms, mean $\pm$ SEM, $n = 15$ rats). Qualitatively similar results were also observed in the mean coactivation strength aligned to the centre of OFF states, which are equivalent to DOWN states but were detected purely based on spiking activity[32–34] (Supplementary Fig. 14b, c). These results indicate that BLA–PL5 ensemble coactivations are immediately followed by UP–DOWN transitions.

Similar to previous observations in the dHPC[5,9,35,36], vHPC SWR occurrences increased moderately around UP–DOWN transitions (maxima were reached $167 \pm 22$ ms prior to slow-wave peaks in post-cond NREM, mean $\pm$ SEM, $n = 14$ rats; Fig. 5b). In contrast, the coactivations of vCA1–PL5 ensemble pairs peaked at $75 \pm 12$ ms after slow-wave peaks (mean $\pm$ SEM, $n = 12$ pairs; Fig. 5c), presumably around DOWN–UP transitions (Fig. 5a; the interval from slow-wave peak to offset was $110 \pm 5.9$ ms [mean $\pm$ SEM, $n = 15$ rats] in post-cond NREM). Similar results were obtained when the centres of OFF states were used as triggers (Supplementary Fig. 14c). These observations suggest that subsets of SWRs occurring around DOWN–UP transitions are preferentially involved in vCA1–PL5 ensemble coactivations.

In summary, vCA1–PL5 and BLA–PL5 ensemble coactivations occur preferentially at distinct time lags with respect to slow-waves, suggesting that slow-waves coordinate the timing of ensemble coactivations in a brain region combination-dependent manner.

**BLA–vCA1–PL5 triple-activation is enhanced in post-cond NREM.** In addition to the coactivation of inter-regional ensemble pairs, we observed nearly simultaneous activation of BLA, vCA1, and PL5 ensembles during post-cond NREM (Fig. 6a). In rats with implants in all three regions (BLA, vCA1, and PL5; 6 rats), we quantified this observation by expanding CCG analysis to triplets by calculating "triple CCG", which was defined as the mean of the products of three activation strengths with various time shifts (Fig. 6b). The significance of the peak of actual triple CCG was evaluated based on the distribution of triple CCG peaks

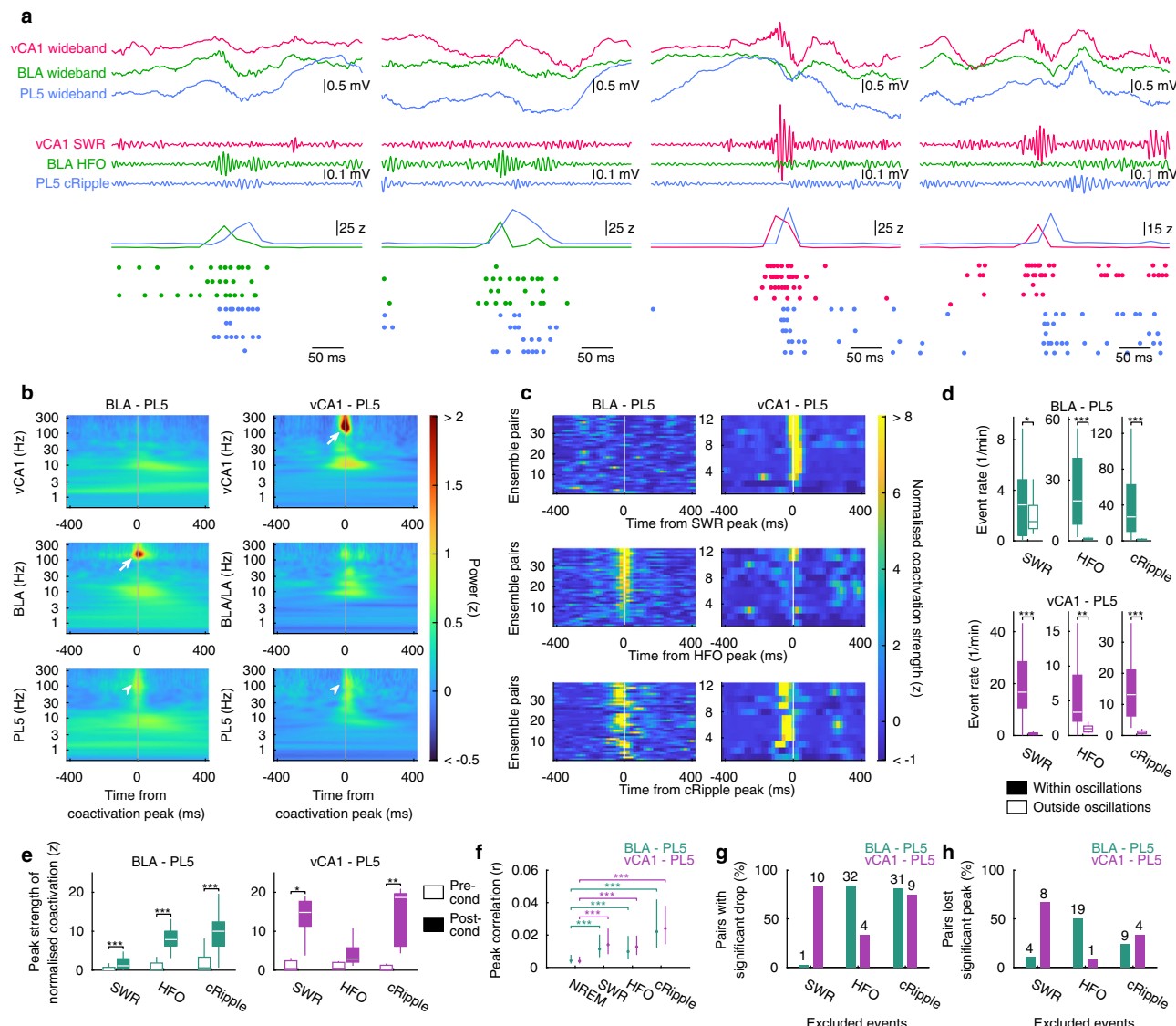

**Fig. 4 HFOs, SWRs, and cRipples are associated with inter-regional ensemble coactivations during NREM. a** Representative examples of BLA–PL5 coactivation (left two panels) and vCA1–PL5 coactivation (right two panels). Wideband and filtered LFP recorded from vCA1, BLA, and PL5 are shown at the top. The instantaneous ensemble activation strength and spikes from member cells of the ensembles are shown at the bottom. **b** BLA–PL5 and vCA1–PL5 coactivation-triggered average of LFP wavelet powers during post-cond NREM. Peaks corresponding to HFOs and SWRs (white arrows) are observed at the BLA–PL5 and vCA1–PL5 coactivations, respectively, and peaks reflecting cRipples (white arrowhead) are apparent at the coactivations of both region pairs. **c** Oscillatory event peak-triggered average BLA–PL5 and vCA1–PL5 coactivation strength. The BLA–PL5 and vCA1–PL5 ensemble pairs are sorted based on the peak height around time 0 of HFO- and SWR- triggered averages, respectively. **d, e** Box plots indicate coactivation event rates within/outside oscillatory events during post-cond NREM (**d**) and the coactivation strength peaks within oscillatory events during pre- and post-cond NREM (**e**). The centre lines and limits of the boxes represent median and quartiles, respectively, and whiskers extend to the maxima and minima with outliers excluded. ***$p < 0.001$, **$p < 0.01$, *$p < 0.05$, WSR-test. **f** Median of peak correlation between ensemble pairs (including non-coupled pairs) within oscillatory events in post-cond NREM. Error bars indicate quartiles. ***$p < 0.001$, **$p < 0.01$, *$p < 0.05$, post hoc TK test following the Friedman test. Only statistical results of comparisons with the entire NREM are shown. **g, h** Proportions of coupled-ensemble-pairs that significantly ($p < 0.01$, random jittering analysis) reduced CCG peak height (**g**) and proportions of coupled-ensemble-pairs that lost significant peaks on CCG (**h**) by excluding the time bins containing SWRs, HFOs, or cRipples. The numbers of pairs are superimposed on the top of the bars. The numbers of pairs analysed in (**b–h**) are summarised in Supplementary Table 4. Detailed statistics are shown in Supplementary Data 1. Source data are provided as a Source Data file.

of shuffled surrogates (Fig. 6b). We identified 100 coupled-ensemble-triplets (of the 2925 possible triplet combinations), and every single rat had at least one coupled-ensemble-triplet (Supplementary Table 4). The peak position of triple CCG varied across triplets but was significantly skewed from the uniform ($p < 0.001$, $\chi^2$ test, $n = 100$ triplets). The histogram revealed a significant peak at [−60 ms, −20 ms] (Fig. 6c), suggesting that, most commonly, the vCA1 ensemble is activated first, followed by

the BLA ensemble, which, in turn, is followed by the PL5 ensemble.

Next, we examined whether the triple-activations were a mere coincidence of ensemble-coactivation events. We defined partial-pairs as pairs of ensembles participating in the coupled-ensemble-triplet of interest (each triplet had three partial-pairs). We observed that 55.8% of the coupled-ensemble-pairs (60.5%, 30.0%, and 75.0% for BLA–PL5, vCA1–PL5, and vCA1–BLA

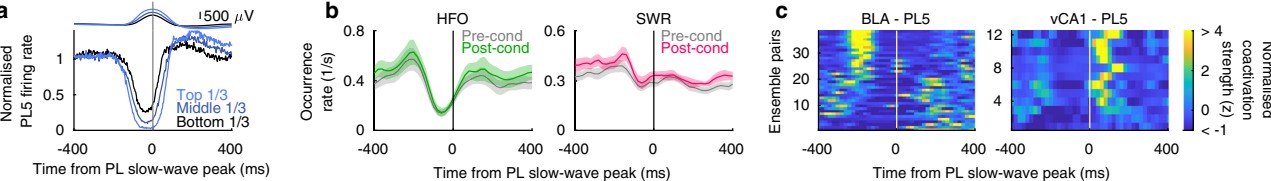

**Fig. 5 Inter-regional ensemble coactivations are modulated by cortical slow-waves. a** Slow-wave peak-triggered average population firing-rate in PL5 (in 3 ms bins, n = 8 rats). Slow-waves were separated into tertiles based on their amplitudes within each rat, and the firing rates were normalised to the mean in periods of [−2,000 ms, −1500 ms] and [+1500 ms, +2000 ms] from the slow-wave peaks. The mean PL5 LFP waveforms are presented on the top. The troughs of the firing rates precede the peaks of the cortical slow-waves, as observed previously[32,36]. **b** Slow-wave peak-triggered average of HFO and SWR occurrence rates. The lines and shaded areas indicate the mean and SEM, respectively (n = 15 and 14 rats for HFOs and SWRs, respectively). **c** Slow-wave peak-triggered average BLA–PL5 and vCA1–PL5 coactivation strength. Normalised coactivation strength was obtained as a z-score of the results in periods of ± 2000 ms range. The numbers of pairs analysed are summarised in Supplementary Table 4. Source data are provided as a Source Data file.

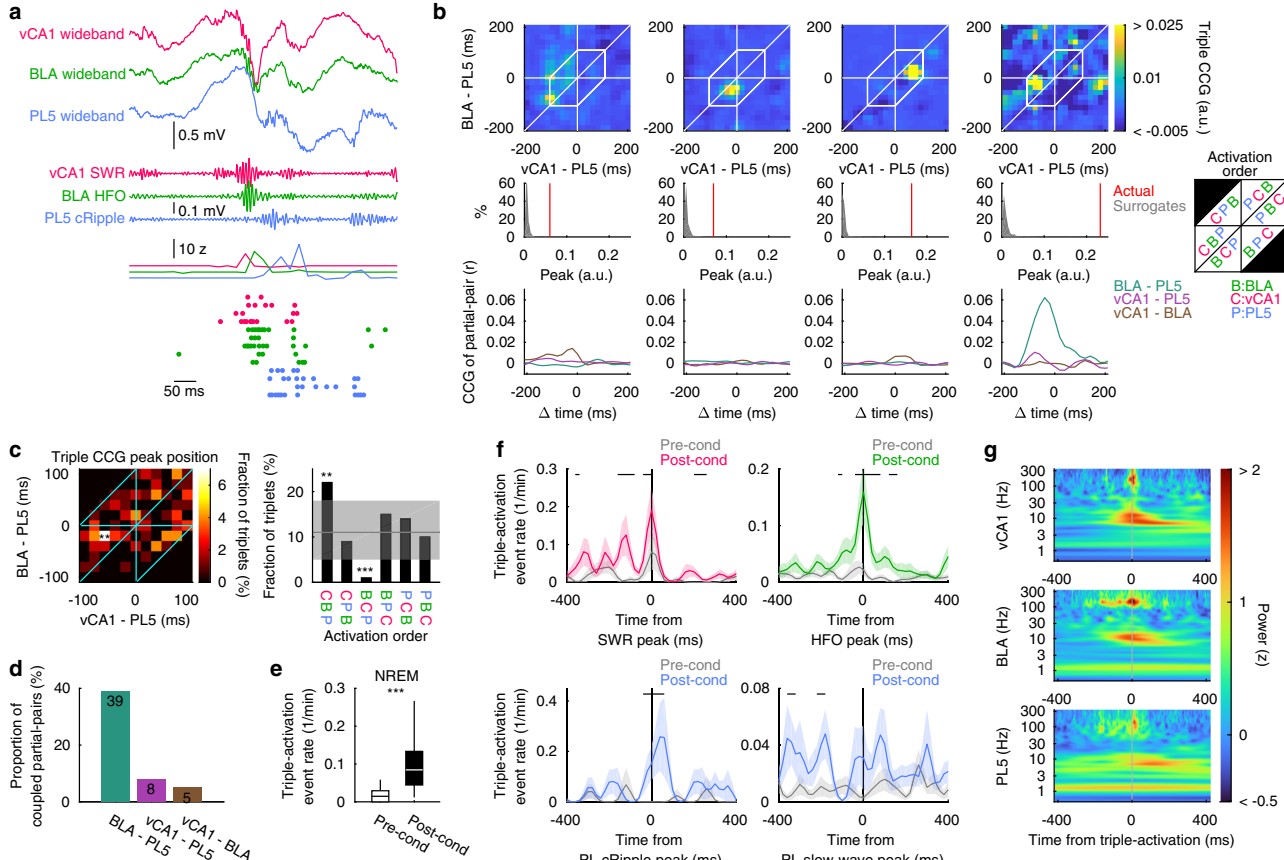

**Fig. 6 Triple-activation across the vCA1, BLA, and PL5 during post-cond NREM. a** Representative example of triple-activation across vCA1, BLA, and PL5 ensembles. Wideband and filtered LFP recorded from the vCA1, BLA, and PL5 are shown at the top. Instantaneous ensemble activation strength and spikes from the member cells of the ensembles are shown at the bottom. **b** Representative examples of triple CCGs (top), distribution of triple CCG peak height of the surrogates and the peaks of the actual data (middle), and CCGs of participating ensemble pairs (partial-pairs; bottom). The white hexagons on the top panels indicate the range of peak detection (time gaps of all combinations <100 ms). The leftmost panel illustrates the triplet shown in (**a**). The inset on the right illustrates the order of ensemble activation. **c** Distribution of triple CCG peak position (left) and ensemble activation order (right) across coupled-ensemble-triplets. The peak distribution is non-uniform (p < 0.001, χ² test) with a significant peak (**p < 0.01, based on Poisson distribution with Bonferroni correction). The grey line and shaded areas on the right panel indicate the mean and 95% confidence interval, respectively, assuming uniform distribution. ***p < 0.001, **p < 0.01, test based on Poisson distribution with Bonferroni correction. **d** The proportion of the partial-pairs of coupled-ensemble-triplets with significant peaks on their CCG during post-cond NREM. The numbers of partial-pairs are superimposed on the bars. **e** Box plots indicating the BLA–vCA1–PL5 triple-activation event occurrence rate per coupled-ensemble-triplet. The centre lines and limits of the boxes represent the median and quartiles, respectively, and whiskers extend to the maxima and minima with outliers excluded. ***p < 0.001, WSR-test. **f** Oscillatory event peak-triggered average of triple-activation event rates of coupled-ensemble-triplets. The lines and shaded areas indicate the means and SEMs, respectively. Periods with significant differences between pre- and post-cond are indicated with black ticks on the top (p < 0.05, WSR-test). **g** Triple-activation event-triggered average of wavelet power in the vCA1, BLA, and PL5. The numbers of ensemble triplets analysed are summarised in Supplementary Table 4. Detailed statistics are shown in Supplementary Data 1. Source data are provided as a Source Data file.

[$n$ = 38, 10, and 4 coupled-ensemble-pairs in 6 rats], respectively) were partial-pairs of the coupled-ensemble-triplets. In addition, only 17.3% of partial-pairs were coupled-ensemble-pairs (39%, 8.0%, and 5.0% for BLA–PL5, vCA1–PL5, and vCA1–BLA, respectively; $n$ = 100 partial-pairs each; Fig. 6b, d). Although a numerical simulation revealed that triple-activations also induced peaks of partial-pairs, the peaks of triple CCGs were more robust than the inevitable peaks of partial-pairs against "noisy" solo activation events (Supplementary Fig. 15a, b). We also confirmed that the pairwise coactivation of each partial-pair did not induce prominent peaks on the triple CCG (Supplementary Fig. 15c, d). These results suggest the existence of restricted time windows in which ensembles of the three brain regions are preferentially activated together.

Next, we sought network activity patterns during which triple-activations occur. First, we calculated the instantaneous triple-activation strength as a product of the instantaneous ensemble activation strengths with optimal time shift determined based on the triple CCG peak position, then detected triple-activation events by thresholding the trace of the instantaneous triple-activation strength. Similar to the enhancement of inter-regional coactivation in post-cond sleep (Figs. 2 and 3), triple-activation event rates significantly increased after fear conditioning (Fig. 6e and Supplementary Fig. 16). Enhancement of triple-activation in post-cond NREM was prominent at SWR, HFO, and cRipple peaks (Fig. 6f). Consistently, the triple-activation-triggered average of LFP wavelet power displayed clear peaks corresponding to SWRs, HFOs, and cRipples (Fig. 6g). However, slow-wave modulation on triple-activation event rates was not prominent (Fig. 6f). These findings indicate that triple ensemble activation events were enhanced by SWRs, HFOs, and cRipples during post-cond NREM.

**BLA–PL5 and vCA1–PL5 ensemble coactivations develop with distinct time courses**. The ensembles in BLA, PL5, and vCA1 were activated in response to the shock presentation (Supplementary Figs. 17a and 18a), suggesting that these ensembles are related to the memory of shock events. We further examined whether the coactivations (Figs. 2 and 3) and triple-activations (Fig. 6) of ensembles existed prior to memory acquisition or developed after the experience. The shock-triggered average of coactivation events revealed that coupled-ensemble-pairs (determined based on the CCGs of instantaneous activation strength during post-cond NREM) are more frequently coactivated than non-coupled ones in BLA–PL5 at shock onsets (Fig. 7a and Supplementary Fig. 17). Interestingly, the rates of BLA–PL5 coactivation events during shock presentations were highest during the first shock and gradually decreased with subsequent shocks, whereas the rates of those occurring between the shocks did not change (Supplementary Fig. 17). In contrast, coactivation event rates in inter-shock periods gradually increased in vCA1–PL5 coupled-ensemble-pairs, although the change was not statistically significant (Supplementary Fig. 17). Coactivation event rates in vCA1–PL5 ensemble pairs were not different between the coupled and non-coupled pairs, and both coupled and non-coupled vCA1–PL5 ensemble pairs were weakly but significantly activated at shock onset (Fig. 7b). Triple-activation of BLA–vCA1–PL5 coupled-ensemble-triplets was more strongly enhanced than that of non-coupled triplets at shock onset (Fig. 7c), and this enhancement remained largely constant across the conditioning session (Supplementary Fig. 17). Additionally, BLA–PL5 coupled-ensemble-pairs and BLA–vCA1–PL5 coupled-ensemble-triplets, but not vCA1–PL5 ensemble pairs, were weakly but significantly activated at cue onsets (Supplementary Fig. 19). However, neither the coactivation of ensemble pairs nor

the triple-activation of ensemble triplets was enhanced at freeze onset (Supplementary Fig. 19). These results indicate that coactivation of BLA–PL5 coupled-ensemble-pairs and triple-activation of BLA–vCA1–PL5 coupled-ensemble-triplets formed rapidly during memory acquisition. In contrast, the coactivation of vCA1–PL5 ensemble pairs evolved with a different time course. Both coupled and non-coupled vCA1–PL5 ensemble pairs were weakly coactivated during shocks; subsequently, coactivations of coupled-ensemble-pairs, but not those of non-coupled pairs, were slowly enhanced during the conditioning sessions. The coactivations of coupled vCA1–PL5 ensemble pairs were further differentiated from those of non-coupled pairs during the sleep period that followed those experiences.

Ensembles in the PL5 and vCA1 were activated more frequently in the first NREM epoch during post-cond homecage sessions (started 50.0 ± 8.0 min [mean ± SEM, $n$ = 15 rats] after the end of conditioning sessions and lasted for 435.7 ± 85.9 s [mean ± SEM, $n$ = 15 rats]) than in the last NREM epoch during pre-cond homecage sessions (lasted for 353.3 ± 72.9 s [mean ± SEM, $n$ = 15 rats] and ended 11.2 ± 3.3 min [mean ± SEM, $n$ = 15 rats] prior to the start of conditioning sessions; Supplementary Fig. 18b, c). Thus, we hypothesised that the coactivation of vCA1–PL5 coupled-ensemble-pairs was enhanced at the very beginning of sleep after the experiences. To examine this possibility, we visualised the time evolution of coactivations during NREM by aligning coactivation events to the offset/onset of NREM epochs preceding/following conditioning sessions (Fig. 7d, e). Both BLA–PL5 and vCA1–PL5 coupled-ensemble-pairs prominently increased their coactivation event rates from pre- to post-cond NREM, resulting in a more frequent coactivation of coupled-ensemble-pairs than that of non-coupled pairs in the first post-cond NREM epoch (Fig. 7d, e). In contrast, in pre-cond NREM, we detected a significant difference in coactivation event rates between coupled- and non-coupled-ensemble-pairs in BLA-PL5, but not in vCA1-PL5 (Fig. 7d, e). Furthermore, the BLA–vCA1–PL5 coupled-ensemble-triplets were triple-activated more frequently than non-coupled triplets in the first NREM epochs in the post-cond homecage sessions, but not in the last NREM epochs in the pre-cond homecage sessions (Fig. 7f). In summary, BLA–PL5 coactivation and BLA–vCA1–PL5 triple-activation, which developed during memory acquisition (Fig. 7a, c), persisted in the following NREM epochs. In contrast, vCA1–PL5 coactivation, which occurred in a rudimentary manner during memory acquisition (Fig. 7b), became more prominent in coupled-ensemble-pairs than in non-coupled pairs during post-cond NREM.

The coactivation/triple-activation event rates decayed with time (Fig. 7d–f). As early phases of sleep are dominated by NREM, it is possible that coactivations were not detected during REM (Supplementary Fig. 7) due to its temporal delay from the fear conditioning, rather than the sleep state difference. To explore this possibility, we compared the coactivation/triple-activation event rates and the proportion of pairs/triplets that coactivated significantly more often than chance level in the post-cond first REM versus following NREM (Supplementary Fig. 20). Among the coupled-ensemble-pairs/-triplets, coactivation/triple-activation event rates during the first REM were significantly lower than those during the subsequent NREM (Supplementary Fig. 20a). The proportion of significantly coactivated ensemble pairs during the first REM did not differ from chance level (0.5%), with the exception of BLA–PL5, where only 5.3% of the coupled-ensemble-pairs (2/38 pairs) were coactivated (Supplementary Fig. 20b). In contrast, a significant proportion of BLA–PL5 and vCA1–PL5 coupled-ensemble-pairs were coactivated during the NREM epoch following the first REM (36.8% [14/38 pairs] and 25.0% [3/12 pairs] for BLA–PL5 and vCA1–PL5 coupled-ensemble-pairs, respectively). For the vCA1–BLA–PL5 triplets, the proportion of triplets with significant triple-activation was not

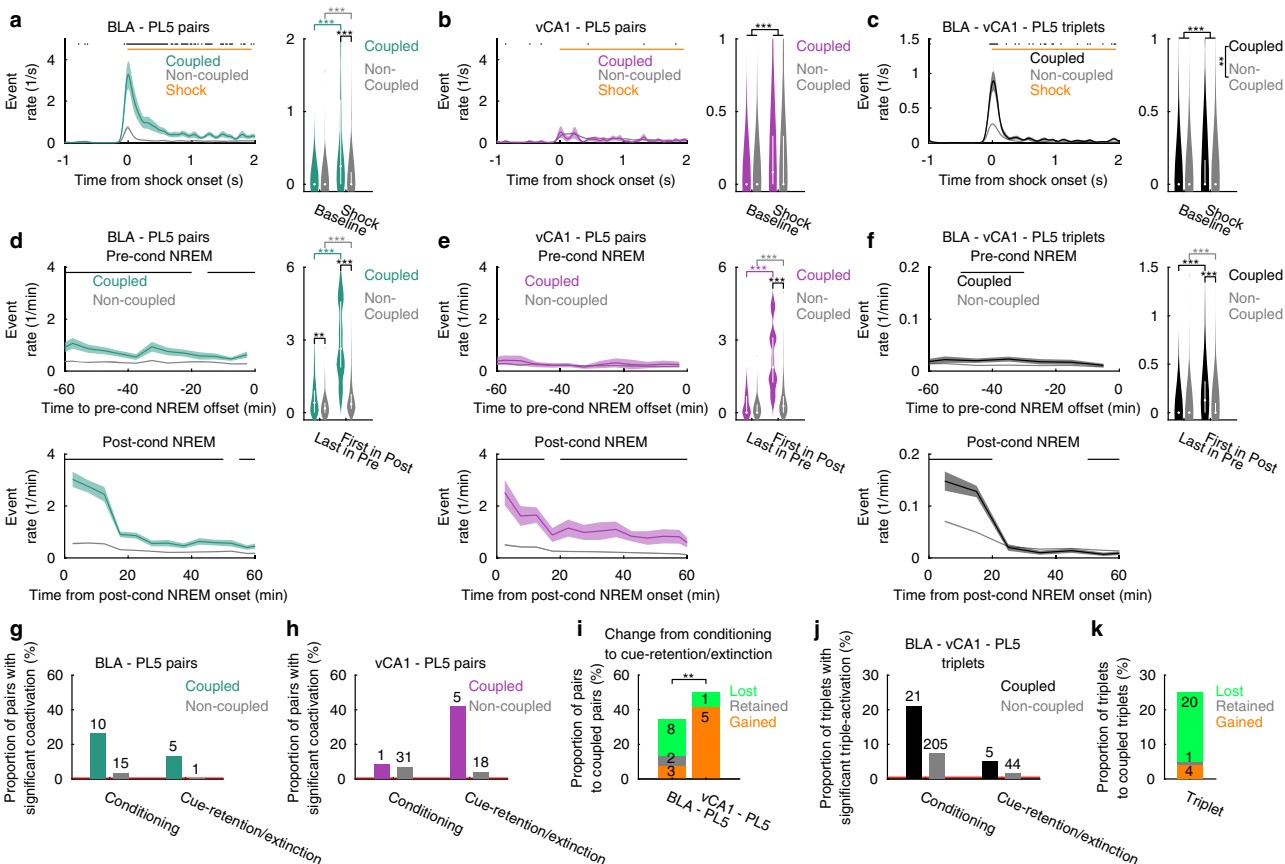

**Fig. 7 BLA–PL5 and vCA1–PL5 ensemble coactivations develop in distinct time courses. a–c** Shock-triggered average of coactivation/triple-activation event rates (left) and violin plots representing event rates in periods 0.05–0.55 s prior to and following shock onsets (baseline and shock, respectively; right). In the left panels, shaded areas represent the SEMs, and black ticks on the top indicate periods with significant differences ($p < 0.05$, WSR-test) between coupled- and non-coupled-ensemble-pairs/triplets. In the right panels, white dots and lines represent the median and quartiles, respectively. \*\*\*$p < 0.001$, \*\*$p < 0.01$, post hoc TK test following two-way ANOVA on the ranks. **d–f** Time-aligned average of coactivation/triple-activation event rates during NREM (left) and violin plots indicating coactivation/triple-activation event rates during last/first NREM epoch in pre-/post-cond homecage sessions (right). In the left panels, the means were calculated within concatenated NREM epochs during pre- or post-cond homecage sessions. The shaded areas represent SEMs, and the black ticks on the top indicate periods with significant differences ($p < 0.05$, WSR-test) between coupled- and non-coupled-ensemble-pairs/triplets. In the right panels, white dots and lines represent median and quartiles, respectively. \*\*\*$p < 0.001$, \*\*$p < 0.01$, post hoc TK test following two-way ANOVA on the ranks. **g, h** Proportion of BLA–PL5 (**g**) and vCA1–PL5 (**h**) ensemble pairs that displayed significant CCG peaks during conditioning and cue-retention/extinction sessions after the first tone onset. Analysed durations of the conditioning and cue-retention/extinction were matched (see "Methods"). The numbers of ensemble pairs are superimposed on the top of the bars. Red horizontal bars indicate the chance level (0.5%). **i** Proportion of coupled-ensemble-pairs that gained, retained, or lost significant CCG peaks from conditioning sessions to cue-retention/extinction sessions. \*\*$p < 0.01$, Fisher's exact test. The number of pairs is superimposed on the bars. **j, k** Same as (**g–i**) but for BLA–vCA1–PL5 triplets. Ensembles were detected based on neuronal firings in entire conditioning sessions and divided into coupled- and non-coupled-ensemble-pairs/triplets based on their coactivation/triple-activation in post-cond NREM. The numbers of ensemble pairs and triplets analysed are summarised in Supplementary Table 4. Detailed statistics are shown in Supplementary Data 1. Source data are provided as a Source Data file.

higher than chance level during both the first REM and the following NREM (Supplementary Fig. 20b). These results suggest that the BLA–PL5 and vCA1–PL5 coactivation occurring during REM, if any, were significantly reduced compared with that during NREM, and that this difference cannot be attributed solely to the longer temporal delay from fear conditioning for REM versus NREM.

We then investigated whether the coactivations and triple-activations also occur in other behavioural sessions. To ensure sufficient data collection for statistically reliable analysis, we restricted the analyses to the conditioning and cue-retention/extinction sessions. For each ensemble pair, the significance of CCG peaks was examined during the behavioural sessions. The proportion of BLA–PL5 coupled-ensemble-pairs with significant peaks tended to be higher in the conditioning session than in the cue-retention/extinction session, whereas that of vCA1–PL5

coupled-ensemble-pairs with significant peaks tended to be higher in the cue-retention/extinction session than in the conditioning session (Fig. 7g, h). Indeed, changes between the conditioning and cue-retention/extinction sessions in the proportion of pairs with significant CCG peaks were significantly different between BLA–PL5 and vCA1–PL5 coupled-ensemble-pairs (Fig. 7i). The triple-activation of BLA–vCA1–PL5 coupled-ensemble-triplets showed decreasing trends from the conditioning session to the cue-retention/extinction sessions (Fig. 7j, k). Collectively, these results indicate that the time evolution of inter-regional ensemble coactivation depends on the participating regions.

**Fast oscillations coordinate inter-regional ensemble coactivation during memory retrieval.** Next, we examined the network

activity patterns during which the coactivation in cue-retention/extinction sessions occurred. As triple-activation events were rare during the cue-retention/extinction sessions (88% of coupled-ensemble-triplets had ≤2 triple-activation events, and triple-activation event rates of coupled-ensemble-triplets were $0.022 \pm 0.003$ min$^{-1}$ in cue-retention/extinction sessions whereas those in conditioning sessions were $0.184 \pm 0.018$ min$^{-1}$ [mean ± SEM, $n = 100$ triplets, $p < 0.001$ Wilcoxon signed-rank test (WSR-test)]; see also Fig. 7j, k), we did not further analyse triple-activation-related activity patterns during cue-retention/extinction sessions. Similar to post-cond NREM (Fig. 4), BLA–PL5 and vCA1–PL5 ensemble coactivations were accompanied by awake HFOs (aHFOs) and SWRs, respectively (Fig. 8a). In addition, fast PL oscillations were observed at the time of coactivation (Fig. 8a).

To better understand these observations, we defined "reappeared-ensemble-pairs" as coupled-ensemble-pairs that had significant CCG peaks also during cue-retention/extinction sessions (peaks were assessed on CCGs in whole cue-retention/extinction sessions; Supplementary Table 6). Among the coupled-ensemble-pairs, 23.7% BLA–PL5 (9 of 38) and 50.0% vCA1–PL5 (6 of 12) were reappeared-ensemble-pairs. Those proportions were significantly larger than those of non-coupled pairs with significant peaks on their CCG during cue-retention/extinction sessions (Supplementary Fig. 21).

Then, we used the coactivation events of reappeared-ensemble-pairs to calculate event-triggered average LFP wavelet power and found strong peaks reflecting the aHFOs and SWRs at BLA–PL5 and vCA1–PL5 coactivations, respectively (Fig. 8b). Consistently, we observed a transient enhancement of BLA–PL5 and vCA1–PL5 ensemble coactivations around aHFO and SWR peaks, respectively (Fig. 8c). In addition, BLA–PL5 ensemble coactivations preceded cRipples ($\Delta t = -40.0 \pm 13.2$ ms, mean ± SEM, $n = 9$ pairs, $p < 0.05$, WSR-test; Fig. 8c). BLA–PL5 coactivation occurrence rates were significantly elevated during aHFOs (Fig. 8d), and vCA1–PL5 coactivations tended to occur during SWRs ($p = 0.063$, WSR-test, $n = 6$ pairs). Furthermore, 55.6% of BLA–PL5 and 83.3% of vCA1–PL5 reappeared-ensemble-pairs showed a significant decrease in CCG peaks when bins containing aHFOs and SWRs, respectively, were excluded from the analysis (Fig. 8e).

Moreover, we observed a transient increase in the PL5 LFP wavelet power at ~130 Hz (Fig. 8b), which corresponds to cRipples, at both BLA–PL5 and vCA1–PL5 ensemble coactivations. Consistently, BLA–PL5 and vCA1–PL5 ensemble coactivations occurred more frequently during cRipples (Fig. 8d), although the difference in BLA–PL5 did not reach statistical significance ($p = 0.055$, WSR-test, $n = 9$ pairs). We also observed other PL5 wavelet power peaks in the slow gamma ($\gamma_{slow}$) band (30–60 Hz) at vCA1–PL5 ensemble coactivations (Fig. 8b). Consistent with this observation, vCA1–PL5 coactivation strength was transiently enhanced around the peaks of $\gamma_{slow}$ in the PL5 (Fig. 8c), and the vCA1–PL5 ensemble-coactivation event rates were higher during $\gamma_{slow}$ epochs (Fig. 8d). In contrast, no noticeable peaks were detected on the PL fast gamma ($\gamma_{fast}$; 60–90 Hz)-triggered average coactivation strength (Fig. 8c). These findings indicate that fast network oscillations also contributed to coactivations during memory retrieval.

To examine whether these coactivations detected during cue-retention/extinction sessions were related to memory retrieval, we calculated the proportion of reappeared-ensemble-pairs whose coactivations were significantly modulated by freezing behaviour or cue presentation during the entire cue-retention/extinction sessions (Fig. 8f). Strikingly, coactivations of all vCA1–PL5 reappeared-ensemble-pairs were significantly enhanced during freezing behaviour. In addition, significant proportions of

BLA–PL5 reappeared-ensemble-pairs had enhanced or suppressed coactivations during cue presentation (Fig. 8f). The BLA–PL5 reappeared-ensemble-pairs displayed transient enhancement of coactivation events at cue onset (Fig. 8g). The cue-onset enhancement of BLA–PL5 reappeared-ensemble-pairs coactivation was also significant when the analysis was restricted to the cue-retention—but not in the extinction—part of the session (Supplementary Fig. 22). The activation of PL5 ensembles participating reappeared-ensemble-pairs showed similar patterns (Supplementary Fig. 22). Such cue-onset modulation was not observed in the coactivation of vCA1–PL5 reappeared-ensemble-pairs (Fig. 8h and Supplementary Fig. 22). On the other hand, we did not observe an enhancement of ensemble activation/coactivation at freeze onset during the cue-retention or extinction parts of the session (Supplementary Fig. 22). In addition, coactivations of the reappeared-ensemble-pairs were not significantly modulated by cue presentation during baseline sessions or awake immobility in post-cond homecage sessions (Supplementary Fig. 23). These observations suggest that the BLA–PL5 and vCA1–PL5 coactivations that re-emerged during cue-retention/extinction sessions are related to memory retrieval, rather than simply reflecting sensory inputs or overt behaviour.

Overall, these results suggest that inter-regional coactivations accompany fast network oscillations in the participating regions and support memory retrieval.

**Cell ensembles are configured prior to conditioning in the BLA and PL5 but not in the vCA1.** Recent studies have suggested that memory-encoding cells are more excitable prior to experience, and that memory-encoding ensembles may be configured before experience[37,38]. To determine whether ensembles participating in inter-regional coactivation (Supplementary Table 7) are pre-determined, we first examined the significance of ensemble activation event rates by comparing them with those of surrogate ensembles (Fig. 9a). In the BLA and PL5, the proportion of significantly activated ensembles was significantly higher than chance, even in the pre-cond NREM, suggesting that cell ensembles that become active during fear conditioning are pre-configured in these regions. In contrast, vCA1 ensembles coupled with PL5 were activated significantly more than chance level only after fear conditioning, suggesting that vCA1 ensembles participating in vCA1–PL5 coactivation developed in an experience-dependent manner. Activation event rates of BLA ensembles were comparable between pre- and post-cond NREM, whereas vCA1 ensembles coupled with PL5 and PL5 ensembles coupled with BLA were activated more frequently in post- than in pre-cond NREM (Fig. 9b).

Next, we examined whether the excitability of cells that contribute to the ensemble coactivation is pre-determined. We identified the member cells of each ensemble as the cells with the top five weights of the projection vector and considered member cells of coactivation-participating-ensembles as ensemble-coactivation-contributing cells. The coupled region of each ensemble-coactivation-contributing cell was defined as the partner region of the coactivation-participating-ensemble of which the cell was a member. Although a given PL5 cell can have multiple coupled regions, most cells had one or no coupled region (Supplementary Table 8 and Supplementary Fig. 24a). Inhibitory cells had more coupled regions than excitatory cells (Supplementary Fig. 24a), whereas the proportion of inhibitory cells among ensemble-coactivation-contributing cells were similar to those of other ensembles (Supplementary Fig. 24b). In the BLA, vCA1, and PL5, ensemble-coactivation-contributing excitatory cells fired faster than other excitatory cells (Fig. 9c and Supplementary Fig. 24c); however, this difference was not

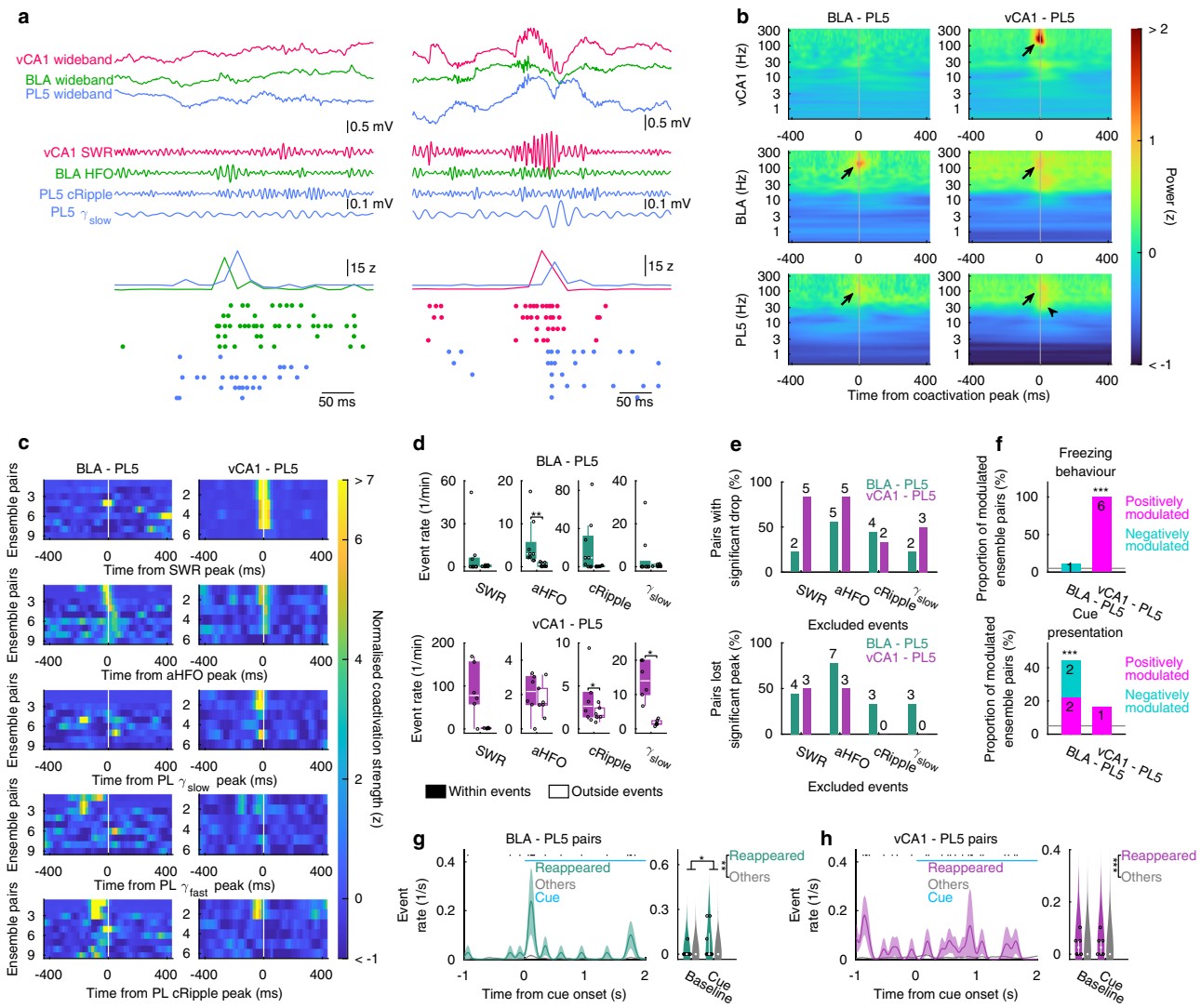

**Fig. 8 Fast oscillations host coactivation during memory retrieval. a** Representative examples of BLA–PL5 (left) and vCA1–PL5 (right) coactivation, as in Fig. 4a, but during cue-retention/extinction sessions. **b, c** Coactivation-triggered average of LFP wavelet power (**b**), and oscillatory event-triggered average of instantaneous coactivation strength (**c**) during cue-retention/extinction sessions. Only reappeared-ensemble-pairs were used. The black arrows and arrowhead in (**b**) indicate the peaks reflecting SWRs/aHFOs/cRipples in vCA1/BLA/PL5 and $\gamma_{slow}$ in PL5, respectively. In panel (**c**), the BLA–PL5 and vCA1–PL5 ensemble pairs were sorted based on the height of HFO- and SWR- triggered average peaks around time 0, respectively. **d** Box plots indicate coactivation event rates of reappeared-ensemble-pairs within/outside oscillatory events during cue-retention/extinction sessions. The centre lines and limits of the boxes represent median and quartiles, respectively, and whiskers extend to the maxima and minima with outliers excluded. Open circles indicate individual data points, including outliers, with random horizontal jitters for visualisation. ***$p < 0.001$, **$p < 0.01$, *$p < 0.05$, WSR-test. **e** Proportions of reappeared-ensemble-pairs that significantly ($p < 0.01$, tested using random jittering analysis; see "Methods") reduced CCG peak height and that lost significant peaks on CCG by excluding time bins containing SWRs, aHFOs, cRipples, or $\gamma_{slow}$. **f** Proportion of reappeared-ensemble-pairs for which coactivation event rates indicated significant positive or negative modulation by freezing behaviour (top) or cue presentation (bottom). Horizontal bars indicate the chance level (5%). ***$p < 0.001$, $\chi^2$ test. **g, h** Cue-onset-triggered averages of the coactivation event rates in BLA–PL5 (**g**) and vCA1–PL5 (**h**) ensemble pairs. In the left panels, the line and shaded area indicate the mean and SEM, respectively. Periods in which the event rates of reappeared-ensemble-pairs were significantly different from those of other ensemble pairs are indicated with black ticks on the top ($p < 0.05$, WSR-test). Right panels are violin plots showing event rates in periods 0.05–0.55 s prior to/following cue onsets (baseline and cue, respectively). White dots and lines represent medians and quartiles, respectively. Individual data points of reappeared-ensemble-pairs are shown with open circles with horizontal random jitters for visualisation. ***$p < 0.001$, **$p < 0.01$, *$p < 0.05$, post hoc TK test following two-way ANOVA on the ranks. For (**b**–**h**), data obtained from the entire cue-retention/extinction session were used, and the numbers of analysed ensemble pairs are summarised in Supplementary Table 6. In (**e**) and (**f**), the numbers of pairs are superimposed on the top of the bars. Detailed statistics are shown in Supplementary Data 1. Source data are provided as a Source Data file.

detected in the inhibitory cells (Supplementary Fig. 24c, d). Furthermore, firing rates were not correlated with the weights of the projection vectors (Supplementary Fig. 24e), suggesting that higher firing rates of ensemble-coactivation-contributing excitatory cells cannot be a trivial result of the analysis method. In the BLA and vCA1, excitatory cells coupled with PL5 displayed significantly higher firing rates than other cells (Fig. 9c), whereas there were no significant differences in the firing rates of excitatory cells across pre- and post-cond NREM (Fig. 9c). These results suggest that ensemble-coactivation-contributing excitatory

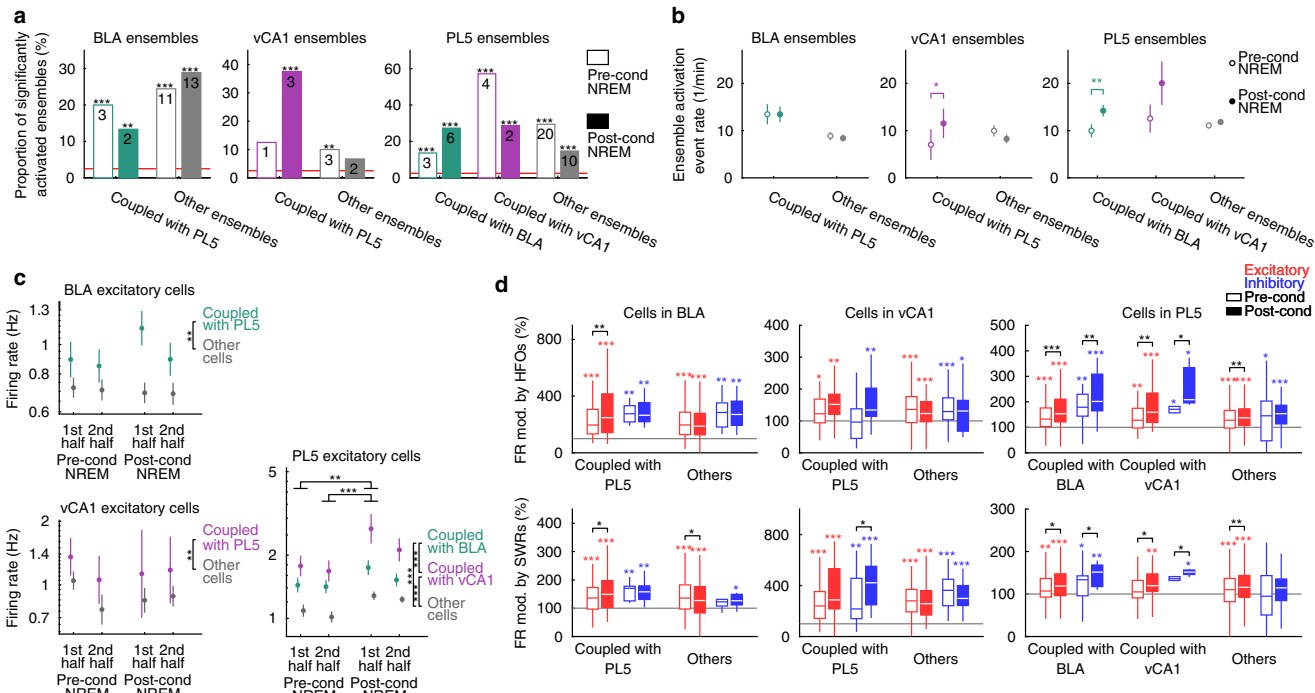

**Fig. 9 Coactivation-participating-ensembles in the BLA and PL5, and the distinct firing properties of ensemble-coactivation-contributing cells in the BLA and vCA1 are configured prior to conditioning. a** The proportion of ensembles significantly activated during pre- and post-cond NREM. Ensembles not coupled with PL5, BLA, or vCA1 are shown as other ensembles. The horizontal line indicates the chance level (2.5%), and the numbers of ensembles are superimposed on the bars. ***$p < 0.001$, **$p < 0.01$, $\chi^2$ test. **b** Mean ensemble activation event rates in NREM. Ensembles not coupled with PL5, BLA, or vCA1 are shown as other ensembles. Error bars indicate the SEM. **$p < 0.01$, *$p < 0.05$, WSR-test for difference between pre- and post-cond NREM. **c** Mean firing rates of coupled and non-coupled (others) excitatory cells during NREM in the 1st and 2nd halves of pre- and post-cond homecage sessions. Error bars indicate SEM. ***$p < 0.001$, **$p < 0.01$, *$p < 0.05$, post hoc TK test following two-way ANOVA on the logarithm of the data. The coupled regions had a significant effect in all regions, whereas periods had a significant effect in PL5 but not in BLA or vCA1. **d** The firing rate modulation (FR mod.) by HFOs and SWRs (firing-rate ratio of within over outside HFOs or SWRs during NREM) for coupled and non-coupled (others) cells in the BLA, vCA1, and PL5. The centre lines and limits of the boxes represent the median and upper/lower quartiles, respectively, and whiskers extend to the maxima and minima with outliers excluded. Horizontal grey lines indicate no modulation (100%). ***$p < 0.001$, **$p < 0.01$, *$p < 0.05$, WSR-test for modulation within pre- or post-cond NREM (coloured) and changes from pre- to post-cond NREM (black). The numbers of analysed ensembles in (**a**, **b**) and cells in (**c**, **d**) are summarised in Supplementary Tables 7 and 8, respectively. Detailed statistics are shown in Supplementary Data 1. Source data are provided as a Source Data file.

cells are more excitable in the BLA and vCA1, even before the fear conditioning.

Conversely, ensemble-coactivation-contributing cells in the PL5 developed distinct firing properties after conditioning. Two-way analysis of variance (ANOVA) on PL5 excitatory cell firing rates across pre- and post-cond NREM revealed a significant effect of the coupled region and periods, with a significant difference across cells coupled with BLA, vCA1, and cells with which no coupled region was detected (Fig. 9c). In addition, the firing rates of PL5 excitatory cells were significantly higher during NREM in the first half of post-cond homecage sessions than during NREM in pre-cond homecage sessions (Fig. 9c). These results indicate that ensemble-coactivation-contributing cells in the PL5 refine their firing activity through fear conditioning.

Because HFOs and SWRs mediated BLA–PL5 and vCA1–PL5 coactivations, we hypothesise that HFO- and SWR-modulations of ensemble-coactivation-contributing cell firings are enhanced after fear conditioning. Firing-rate modulations were measured as the ratio of firing rates within the events of interest (e.g., HFOs or SWRs) versus mean firing rates across NREM (see "Methods" for detail). In the BLA, the firings of excitatory cells coupled with PL5 were enhanced by HFOs and SWRs more strongly in post- than in pre-cond NREM (Fig. 9d). Similar trends were observed in vCA1 excitatory cells, although the changes in cells coupled with the PL5 did not reach statistical significance ($p = 0.084$ and $0.053$ for HFO- and SWR-modulation, respectively, $n = 19$ cells for

each, WSR-test). Unlike the BLA, SWR-modulation of inhibitory cells coupled with PL5 was also enhanced in the vCA1 (Fig. 9d). In the PL5, HFO- and SWR-modulation of excitatory cells and ensemble-coactivation-contributing inhibitory cells, but not that of other inhibitory cells, was enhanced by fear conditioning (Fig. 9d). Overall, these results suggest that the inter-regional ensemble coactivations after conditioning are associated with the enhanced recruitment of ensemble-coactivation-contributing cells to HFOs and SWRs.

## Discussion

Recent studies have suggested that memory-encoding cell ensembles in local circuits are configured before memory acquisition[19,37], although post-acquisition stabilisation may occur[20,21]. Consistently, our results demonstrated that coactivation-participating-ensembles in the BLA and PL5 were activated more than chance level prior to conditioning (Fig. 9a). Moreover, significant inter-regional ensemble coactivations were observed during post- but not pre-cond NREM (Figs. 2 and 3) and re-emerged during cue-retention/extinction sessions, in which coactivation strength was modulated by freezing behaviour and cue presentation (Figs. 7 and 8). Based on these findings, we propose that elements of a given memory are instantly encoded in preconfigured cell ensembles in various brain regions and that de novo inter-regional ensemble coactivations bind these elements

together to form a new memory (Fig. 10). Furthermore, we hypothesise that vHPC ensembles coupled with the prefrontal cortex work as indices of memory contents[39] represented by prefrontal ensembles, which become more active after memory acquisition (Fig. 9b) owing to the input from vHPC indices. Although we cannot rule out the possibility that the vCA1 stores memory contents, this hypothesis is consistent with the following observations: (1) conditioning induced the activation of vCA1 ensembles coupled with PL5 (Fig. 9b), (2) PL5 cells displayed increases in firing rates and modulation by SWRs between pre- and post-cond NREM (Fig. 9c, d), (3) the coactivation of vCA1–PL5 coupled-ensemble-pairs hosted by SWRs (Figs. 4, 6, and 8) were enhanced during post-cond NREM (Figs. 2, 3, and 7e), and (4) vCA1–PL5 ensemble coactivations reappeared during memory retrieval (Fig. 8 and Supplementary Fig. 21).

The index theory assumes that a hippocampal index develops during memory acquisition, presumably owing to the rapid synaptic changes taking place within the hippocampus[39]. Consistently, the coactivation-contributing-ensembles in the hippocampus developed in an experience-dependent manner (Fig. 9a, b). In contrast, during memory acquisition, the coupled and non-coupled vCA1–PL5 ensemble pairs were similarly coactivated by shock presentations (Fig. 7b). Based on these observations, we hypothesise that first, the hippocampal ensembles developed through experience are non-selectively and weakly linked with cortical ensembles during the process of memory acquisition and that the subset of ensemble pairs is selected and stabilised later (Fig. 10). The recurring coactivation of the selected hippocampal-cortical ensemble pairs associated with fast network oscillations in the involved regions (Fig. 4) may further invoke changes in the neocortical circuitry during the initial stage of memory consolidation, which subsequently supports systems consolidation.

The extent to which the memory representing cells/ensembles are configured prior to an experience remains controversial. We observed that inter-regional ensemble-coactivation-contributing cells in the BLA fired faster than other cells during both pre- and post-cond NREM (Fig. 9c). This observation is consistent with a previous study demonstrating that neurons showing higher excitability immediately before learning are more likely to be allocated to a memory trace in the lateral nucleus of the amygdala[40]. In contrast, vCA1 ensembles coupled with PL5 emerged in an experience-dependent manner (Fig. 9a), whereas ensemble-coactivation-contributing excitatory cells in the vCA1 were more active than other excitatory cells both before and after fear conditioning (Fig. 9c). Our results appear to be discrepant from those of a previous study reporting that population activity patterns in the dHPC are configured prior to novel experiences[19]. Although there are several differences between our study and the previous study in terms of behavioural tasks (non-spatial versus spatial), recorded regions (ventral versus dorsal), and analysis methods (coactivation versus sequential activation), these seemingly discrepant notions may describe similar phenomena in different ways. Previous studies on the dHPC have proposed that neuronal sequences become more stable after experiences[20,21]. Such stabilisations might be required for significant ensemble activation, whereas a relatively noisy pre-configured sequence could be sufficient to induce higher excitability in ensemble-coactivation-contributing cells. Further studies are required to pursue this possibility because the methods used in this study cannot evaluate the temporal order of the local population activation.

Although it remains unclear how vCA1–PL5 coupled-ensemble-pairs were selected after memory acquisition, we speculate that BLA–vCA1–PL5 triple-activation might play a role in this process. The coupled-ensemble-triplets were transiently triple-activated at shock onsets (Fig. 7c), and the coupled triplet activities were significantly more elevated during post-cond

NREM than during pre-cond NREM (Figs. 6e and 7f). Thus, the experience-dependent activation of a network of ensemble triplets may provoke further changes in an inter-regional network, including the vCA1 and PL5. Consistently, BLA activation immediately after learning facilitates memory consolidation by modulating neuronal activity outside the amygdala[29]. Although coupled and non-coupled vCA1–PL5 ensemble pairs differentiated after the emergence of BLA–vCA1–PL5 ensemble triple-activation and BLA–PL5 ensemble coactivation (Fig. 7 and Supplementary Fig. 22), further investigation is necessary to clarify whether BLA–vCA1–PL5 ensemble triple-activation and/or BLA–PL5 ensemble coactivation is required for the selection of vCA1–PL5 coupled ensembles or whether the activation of the BLA alone is sufficient for the development of vCA1–PL5 ensemble coactivation.

Accumulating evidence suggests that inter-regional interactions are coordinated by oscillatory events. Theta oscillations are prominent when animals are in a state of alertness[37], and theta-gamma coupling plays a role in inter-regional communication[41,42]. Faster oscillations were associated with inter-regional ensemble coactivation during NREM (Figs. 4, 10 and Supplementary Fig. 13), during which firing synchrony is higher than that detected during theta oscillations[37]. The SWR-associated synchronous activity should have a large influence on postsynaptic neurons; therefore, it is suitable for efficient off-line memory processing involving inter-regional communication when the brain is disengaged from environmental stimuli[8]. Furthermore, ensemble coactivation during such a synchronous epoch may result in temporal compression of neuronal sequences, which can facilitate plastic changes in synaptic connections[8]. Recent studies have suggested that SWRs can trigger changes in the network activity and synaptic connections within the dHPC[43,44]. Inter-regional recurring coactivation hosted by fast oscillations during NREM may induce further changes in the global network.

Consistent with previous reports[45,46], we only observed a weak modulation of the firing of vCA1 neurons by cortical DOWN states (Supplementary Fig. 14a). Moreover, vCA1–PL5 coactivations occurred during DOWN–UP transitions (Fig. 5c). In contrast, BLA firing activities were strongly suppressed during cortical DOWN states (Supplementary Fig. 14a), and BLA–PL5 ensemble coactivation preferentially preceded UP–DOWN transitions (Fig. 5c). In addition, a subset of vCA1–PL5 ensemble pairs was selected and its coactivations were enhanced during the sleep period that followed the conditioning sessions, whereas BLA–PL5 ensemble coactivations were quickly formed during conditioning (Fig. 7). We hypothesise that reinforcement of vulnerable inter-regional ensemble networks and maintenance of stable ones have distinct temporal windows that are separated by cortical silent periods during NREM.

The inter-regional circuits involved in fear memory shift over a period ranging from one day to several weeks[47,48]. In contrast, BLA–PL5 and vCA1–PL5 coactivations developed rapidly (Fig. 7), and their time courses were comparable to the time window of synaptic consolidation (range: seconds or minutes to hours[1]). This finding implies that BLA–PL5 and vCA1–PL5 ensemble coactivations are consequences of plastic changes in the inter-regional synaptic connections[26,28], which is consistent with our notion that the observed time lags of ensemble coactivations reflect the effect of direct inter-regional projections (Fig. 3d). Subsequently, these fast changes underlying the initial stage of memory consolidation may drive slower changes that support systems consolidation, a process that may take several days, weeks, or months. Because fear memory induced with eyelid stimulation lasts for at least 6 days after conditioning[49], the neuronal dynamics occurring during the consolidation process may be examined by adopting cutting-edge extra-long duration recording techniques[50,51] in future studies.

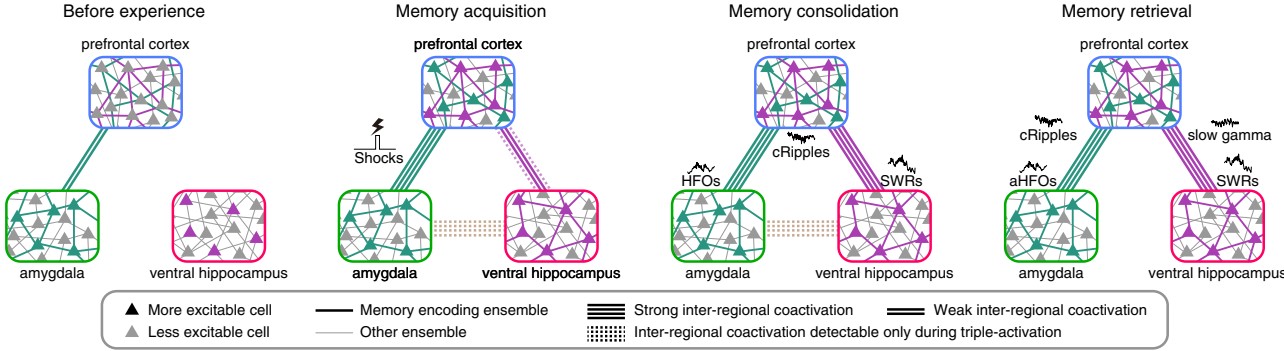

**Fig. 10 Coactivations of preconfigured local ensembles support memory.** Schematic summary of inter-regional ensemble-coactivation development suggests that elements of memories are instantly embedded in preconfigured amygdalar and prefrontal cortical ensembles, whereas the inter-regional network that binds the distributed information develops in an experience-dependent manner (left two panels). The coactivation between hippocampal and prefrontal cortical ensembles is enhanced through an initial stage of memory consolidation, which occurs during the subsequent sleep period (second right panel). During memory retrieval, the prefrontal cortical ensembles are coactivated with amygdalar/hippocampal ensembles (rightmost panel). Inter-regional ensemble coactivations during post-learning sleep and memory retrieval are associated with fast network oscillations (right two panels).

The inter-regional ensemble coactivations reappeared during memory recall (Fig. 7g, h), and the coactivations were hosted by fast oscillations in various brain regions (Fig. 8a–e). These observations suggest that inter-regional ensemble coactivations occurring during fast oscillations in wakefulness support memory retrieval, consistent with the notion that awake SWRs in the dHPC support spatial[52] and fear memory[53]. Furthermore, hippocampal ripple oscillations and neocortical HFOs are significantly coupled during successful memory retrieval in humans, and this coupling is associated with the reinstatement of memory-specific cortical representations[54–56]. In rodent dHPC, replays of firing sequences associated with awake SWRs occur in both forward and reverse orders[57], and reverse replays are selectively involved in memory updates[58]. Although it is unclear whether the diversity of reactivations, such as forward and reverse replays, also exists in vHPC SWRs, HFOs, and cRipples, only a subset of the ensemble coactivations observed here may be involved in memory updates, such as extinction learning. Thus, a precisely structured sequence of inter-regional coactivations, which could not be examined in this study, should be further scrutinised to deepen the understanding of the memory process.

Our results demonstrated that inter-regional coactivation of ensemble pairs occurred dominantly within network fast oscillations (Fig. 4), which indicates that ensemble activities are not correlated most of the time. This explains why the correlation coefficients of inter-regional ensemble activities during NREM were generally small (~0.02), even among the coupled-ensemble-pairs (Figs. 2c, d and 3a). Consistent with this notion, the correlation coefficients were significantly larger when analyses were restricted to the time bins within fast network oscillations (Fig. 4f). In addition, pairwise coactivation could be masked by "noisy" solo activation (Supplementary Fig. 15). Thus, many ensemble pairs with weak correlation may be missed by our conservative criteria (Fig. 2c). Therefore, the proportions of coupled-ensemble-pairs can be underestimated. To minimise bias, we did not use an a priori assumption of the period during which ensemble pairs could coactivate. However, reasonable restriction of the periods for coactivation detection may provide further insights for future research.

Alongside temporally restricted coupling across brain regions, input from other brain regions would also explain the weak correlations in inter-regional ensemble pairs. The BLA, vCA1, and PL5 receive input from many other brain regions. This pattern would naturally restrict the extent of activity in the region of interest (e.g., PL5) owing to activities in another brain region (e.g., BLA or vCA1).

Furthermore, we recorded only a small proportion of cells compared with the number of neurons in the brain region of interest. Indeed, the number of coupled-ensemble-pairs was linearly correlated with the number of analysed ensemble pairs, which, in turn, was linearly correlated with the product of the numbers of simultaneously recorded neurons in the involved regions (Supplementary Fig. 9). Thus, simultaneous recordings from more neurons in more brain regions could provide additional insights into the interaction of neuronal activities across brain regions. Such recordings using recently developing technologies[51] could provide a valuable future research direction.

In this study, we demonstrated that inter-regional coactivation develops after associative learning between neutral (i.e., cues) and aversive (i.e., shocks) stimuli, contrary to the presentation of neutral stimuli alone (Supplementary Fig. 8). This finding implies that association learning may invoke inter-regional coactivation. However, whether an aversive experience without association would also induce inter-regional coactivations remains unclear. Additionally, whether the association of two neutral stimuli induces inter-regional coactivation is unknown. Furthermore, the shock intensity used in this study was greater than that in previous studies (see Methods for more details), which raises the possibility that strong aversive experiences enhance inter-regional coactivation. The combination of a strong aversive experience and consecutive exposure to different chambers in the context- and cue-retention sessions at a short interval might generalise fear memories to multiple contexts or generate a mixture of "tone-place" memory traces[59]. Future studies using multi-regional recordings from animals that experience aversive stimuli without associative cues and those that perform associative learning tasks without aversive stimuli or with milder aversive stimuli would address these questions. These experiments would represent a valuable direction for future studies.

In conclusion, our results suggest that the de novo inter-regional coordination of preconfigured local ensembles forms a new memory. The coordinated ensembles are reactivated together during short bouts of fast network oscillations during the initial stage of memory consolidation as well as during memory retrieval (Fig. 10). Although our findings imply a close association between memory functions and inter-regional ensemble coactivations, the necessity and sufficiency of the coactivations regarding memory functions should be elucidated in future loss/gain-of-function studies. Thus, further studies are warranted to elucidate how changes in inter-regional ensemble coactivation are involved in memory processes and how such changes are regulated.

## Methods

**Animals.** Fifteen singly housed male Long–Evans rats (9.6–15.0 weeks of age, weighing 330–503 g at the time of surgery; Japan SLC) were maintained in a 12 h light/12 h dark cycle (lights on at 8:00 a.m.). Only male rats were used to exclude any potential effects of oestrous cycles on neural activities and animal behaviours. All procedures of animal care and use were approved by the Institutional Animal Care and Use Committee of the Osaka City University (approval #15030) and were performed in accordance with the National Institutes of Health Guide for the Care and Use of Laboratory Animals.

**Surgery for chronic implants of wires and probes.** All surgical procedures were performed under isoflurane anaesthesia (1–3% in 50% air/50% oxygen mixture gas). For each rat, a small incision was made on the pectus skin, and a Teflon insulated stainless wire (AS636, Cooner wire) was sutured on the left intercostal muscles for electrocardiography (ECG) recordings[60]. The other end of the wire was subcutaneously led to a small incision made on the nuchal skin. After suturing the pectus skin, the rats were placed on stereotaxic frames (Model 962, Kopf). Two stainless wires were inserted into the nuchal muscles of each rat for electromyography (EMG) monitoring[43]. A short conductive wire (36 AWG, Phoenix Wire) was soldered on a stainless screw (B002SG89KW, Antrin). The screw was placed on the right olfactory bulb [mediolateral from the midline (ML) +0.5 mm, anteroposterior from bregma (AP) +9.0 mm from bregma] for electro-olfactography (EOG) recordings, which reflect respiration[61]. Wires for ECG, EMG, and EOG were gathered on single connectors for 16-channel differential input pre-amplifiers (C3323, Intan). Two additional screws with 36 AWG wires placed on the cerebellum through small holes made in the skull were used as ground and reference wires, respectively, in all electrophysiological recordings. Two tungsten wires (100 μm in diameter, California Fine Wire) were implanted into each eyelid, and the free ends were placed on single connectors for stimulation. Silicon probes (Buzsaki64sp and Buzsaki64spL from Neuronexus or F6-64 from Cambridge Neurotech) were attached on three-dimensional printed microdrives (STL data are available at https://github.com/Mizuseki-Lab/microdrive) and then coated with poly (3,4-ethylenedioxythiophene) conducting polymer by applying a direct current (0.1 μA for 3 s for each channel) controlled by a nanoZ impedance tester (White Matter). Three 1 × 2 mm rectangular craniotomies centred at (ML +1.0 to +1.5 mm, AP +2.90 to +3.25 mm), (ML +4.60 to +4.80 mm, AP −2.60 to −3.00 mm), and (ML +2.80 to +3.00 mm, AP −4.95 to −5.55 mm) were performed for inserting probes targeting the prefrontal cortex, amygdala, and vHPC in the right hemisphere, respectively. The tips of the probes for the prefrontal cortex were aligned on the parasagittal plane and those for the amygdala and vHPC were aligned on the coronal plane, followed by the insertion of the probes into the brains with angles of −14°, 0°, and 14° from the D–V axis (pointing probe tips medially is negative) for prefrontal cortex, amygdala, and vHPC recordings, respectively, and all shanks were perpendicular to the A–P axis. A small Faraday cage composed of copper mesh was secured on the skull with dental cement (Orthofast, GC) to reduce electrical noise and protect the implants.

**Electrophysiological recordings.** All implanted probes and the connectors hosting ECG, EMG, and EOG signals were connected to a recording system (C3100 256 ch acquisition board from Intan or 512 ch acquisition board from Open Ephys) via pre-amplifiers (C3323 or C3325, Intan). Accelerations of the head were recorded by accelerometers located on the pre-amplifiers. All signals were recorded continuously at 20 kHz with a 16-bit resolution using the Open Ephys GUI software (https://open-ephys.org). Positive polarity is presented upwards throughout this paper.

**Fear conditioning.** After the recovery period (3–15 days; median = 7 days; n = 15 rats), during which the animals had fully recovered from the surgery and the silicon probes were slowly moved downward to the target areas (typically 70–280 μm/day), we performed behavioural tests with electrophysiological recordings. The behaviours of the rats were recorded at 25 frames/s using a video camera (CM3-U3-31S4C-CS, Flir) with an 8 mm lens (LENS-80T4C, Tamron) mounted on the ceiling. Shutter timing was controlled by a stimulator (SEN-7203, Nihon Kohden), and transistor–transistor logic pulses sent from the camera were captured by the electrophysiological recording system to obtain the acquisition timing of individual frames.

The experiment consisted of the following five behavioural sessions: baseline, conditioning, context-retention, cue-retention/extinction, and retention-of-extinction (Fig. 1c, d). All sessions occurred on the same day (Fig. 1d). The cue-retention test and extinction were performed as a continuous session, without separation, except for the analysis shown in Supplementary Fig. 22, where the cue-retention and extinction parts of the session were separated at the onset of the 9th cue. Conditioning and context-retention tests were performed in a tube (diameter, 30 cm; depth, 51 cm) with horizontal stripes placed on metal grids scented with 1% acetate. Other behavioural sessions were conducted in a rectangular box (27 × 33 cm; depth, 40 cm) with vertical stripes placed on a white plastic floor scented with 70% ethanol.

Thirty-second 5 kHz pips[62] (250 ms on, 750 ms off, 74 dB) were used as conditioned stimuli (CS). Trains of 2 ms electrical pulses [lasting 2 s; 4.6–5.1 mA at 8 Hz for each eyelid; the left and right eyelids were stimulated alternatively with half-cycle temporal shift; generated with isolators (SS-202J, Nihon Kohden)] applied through eyelid wires[49] were used as unconditioned stimuli (US). The intensity of the electrical pulse was greater than those in previous studies that used eyelid stimulations as US[49,62–64]. There were 700–750 ms traces between the offsets of the last pips and the onsets of the first shocks. Each session started with a 4-min free exploration periods, during which no tone was presented, followed by the presentation of 4 CS for the baseline, 12 CS for the conditioning[62], 0 CS for the context-retention test, 40 CS for the cue-retention test/extinction, and 8 CS for the test of the retention-of-extinction. CSs were presented at pseudo-random intervals uniformly distributed in the range of 180–240 s, except the last 32 tones in the cue-retention/extinction sessions, where intervals were uniformly distributed in the range of 60–120 s. The inter-CS intervals were determined using the rule described above for the first animal recorded; then, the same inter-CS intervals were used for all the subsequent animals. The US was administered only in the conditioning sessions. The duration of context-retention sessions was 4 min, and the other sessions ended 4 min after the offset of the last CS presentation.

Context-retention sessions were followed by cue-retention/extinction sessions with a short interval during which the rats were kept in the homecage (2.3–3.8 min; median = 3.1 min; n = 15 rats), and other behavioural sessions were separated by 2.5–2.6 h rest/sleep sessions in the homecages (Fig. 1d). Animal behaviour and electrophysiological activity recordings in rest/sleep sessions were performed for >2.5 h prior to the baseline sessions and continued for >2.5 h after the test of the retention-of-extinction. The baseline sessions started at 8:40 a.m., and the retention-of-extinction test sessions ended at 7:45 p.m.; all behavioural and interleaved homecage sessions were conducted during the light cycle, whereas the first and last homecage sessions were predominantly conducted during the dark cycle. Immediately after the recordings, animals were anaesthetised with isoflurane (1–3% in 50% air/50% oxygen mixture gas), and electrode positions were marked by micro-lesioning by applying DC currents (3 μA for 10 s, A365, World Precision Instrument) through the top- and bottom-most recording sites of each shank.

**Histological reconstruction of electrode positions.** After 12–36 h following micro-lesioning, the rats were perfused with 4% paraformaldehyde (441244, Sigma-Aldrich), and the brains were removed from the skulls. After 24–48 h post-fixation in 4% paraformaldehyde, the brains were sliced at a thickness of 50 or 75 μm using a vibratome (VT1200S, Leica). The slices were permeabilised with 0.3% Triton X-100 (35501, Nacalai Tesque) in phosphate-buffered saline (PBS) for 30 min at room temperature and stained sequentially using NeuroTrace Red fluorescent Nissl Stain Solution (200× dilutions in PBS, overnight at 4 °C; N21482, Thermo Fisher) and DAPI (0.5 μg/mL in PBS, for 30 min at room temperature; D1306, Thermo Fisher). The slices were washed with PBS for 30 min at room temperature before and after each staining. Micrographs of the slices were obtained using a confocal (LSM 700, Zeiss) or fluorescence (BZ-X800, Keyence) microscope, and the position of electrodes was reconstructed by visual detection of the sites having micro-lesions (Fig. 1a). The reconstructed positions of the electrode tips are summarised on diagrams found elsewhere[65] (Supplementary Fig. 1a).

**Spike sorting.** The shapes of the shock artefacts estimated using the third-order Savitzky–Golay filter (5 ms window width) were subtracted from the recorded trace in periods from 10 ms prior to shock onsets to 1 s following shock offsets, with the edges attenuated exponentially (τ = 2 ms and 200 ms for onset and offset, respectively), to avoid potential contamination of shock artefacts. The subtraction was performed before spike detection. Spike detection and automated clustering were performed with Kilosort2[66] (https://github.com/MouseLand/Kilosort2). To ensure that no spikes were contaminated by shock artefacts, we discarded the spikes detected around each shock pulse (0.1 ms prior to onsets to 5 ms following the offset of each 2 ms pulse: in total, 7.1 ms per shock pulse) and then manually curated the clusters on the phy software (https://github.com/cortex-lab/phy). Lastly, we evaluated the cluster quality. The isolation distance[67] was calculated with clusters detected on the same shank. The inter-spike interval (ISI) index[68] was calculated using the following formula: $\frac{ISI_{0.5-2}}{ISI_{2-10}} \times \frac{8}{1.5}$, where $ISI_{x-y}$ is the count of ISI in an $[x, y]$ ms window. The ISI index is useful for most cases but not for some non-bursty cells; thus, we also used the contamination rate in Kilosort2. The contamination rate was calculated as follows:

$$\min\left(\frac{ACG_{0.5-n}}{ACG_{0.5-49.5}} \times \frac{49}{n-0.5}, \frac{ACG_{0.5-n}}{ACG_{250-500}} \times \frac{250}{n-0.5}\right),$$ where $ACG_{x-y}$ is the count of spike auto-correlogram in an $[x, y]$ ms window, and $n$ is shifted from 1.5 to 9.5 ms using 1 ms steps, to find the minima.

The mean waveform of each cluster was calculated on high-pass filtered traces (>300 Hz), a channel with maximum spike amplitude was identified, and the mean waveform of that channel was then upsampled to 200 kHz using the spline function of MATLAB (MathWorks). Spike amplitude (trough depth from the baseline) and width (the duration from the trough to the peak) were estimated using the upsampled mean waveform. We set four criteria for cluster quality: (1) isolation distance >15, (2) ISI index <0.2 or contamination rate <0.05, (3) overall mean firing rates >0.01 Hz, and (4) spike amplitudes >50 μV. Units that met all four criteria were used for further analyses.

**Classification of excitatory and inhibitory neurons**. To classify the recorded cells as excitatory or inhibitory cells, we first detected putative excitatory and inhibitory synaptic connections (Supplementary Fig. 2a) using methods described previously[69], with minor modifications. For each cell pair, the CCG of spike timings of two neurons was calculated in 0.1 ms bins and then smoothed with a Gaussian filter ($\sigma = 0.5$ ms). For each cell pair, each spike of both cells was randomly and independently jittered on a uniform distribution in the range of [−5 ms, +5 ms]. Subsequently, the CCG was calculated for the surrogate via the same procedure used for the original spike trains, and the maxima and minima of CCG in the range of [−5 ms, +5 ms] were detected. The procedure was repeated 1,000 times to obtain both the mean and 99% confidence intervals (CIs) of CCGs; 99% global bands were defined as 0.5th and 99.5th percentiles of the maxima and minima in the range of −5 to +5 ms. If actual smoothed CCG had peak/trough values higher/lower than the upper/lower boundary of 99% global bands in [+1 ms, +4 ms] periods, the cell pair was marked as a candidate pair with a monosynaptic excitatory/inhibitory connection. Smoothed CCGs of the candidate pairs were visually inspected to exclude suspicious connections, such as CCG with broad peaks/troughs or strong peaks/troughs at time 0. The remaining pairs were accepted as pairs with monosynaptic excitation/inhibition. Cells with at least one excitatory/inhibitory innervation and no inhibitory/excitatory ones were labelled as excitatory/inhibitory cells.

In all recorded regions, the spike widths of nearly all CCG-based putative excitatory cells were > 0.6 ms, and those of almost all CCG-based putative inhibitory cells were < 0.5 ms (Supplementary Fig. 2b). As such, to classify cells that were not labelled as either excitatory or inhibitory based on CCGs, we used spike width as described previously[70]. Among the CCG-based non-classified cells whose spike polarity was negative, cells with a spike width >0.6 ms and <0.5 ms were labelled as excitatory and inhibitory cells, respectively. CCG-based non-classified cells with a spike width between 0.5 and 0.6 ms and the ones with positive spikes were categorised as non-classified. The numbers of excitatory, inhibitory, and non-classified cells are summarised in Supplementary Table 1.

**Sleep scoring**. Sleep states were automatically scored based on prefrontal LFP, hippocampal LFP, and head acceleration[71] using Buzcode scripts (https://github.com/buzsakilab/buzcode). The scoring results were visually inspected with the power spectrum in the prefrontal LFP, vHPC LFP, and nuchal EMG, and the scoring was modified if necessary. Microarousals[71,72], i.e., short (<40 s) awake periods that were interleaved in NREM or occurred on transitions from REM to NREM, were treated as a part of NREM.

**Heart rates, EOG power spectrum, nuchal EMG amplitudes, and head accelerations**. Individual heartbeats were detected as peaks on ECG signals, and the mean heart rate was calculated in 0.5 s bins and smoothed using the 5 s window moving average. Multitaper power spectrum analysis of EOG signals was performed in 1 s sliding windows with 0.5 s steps using the Chronux toolbox (http://chronux.org). Nuchal EMG was high-pass filtered (>10 Hz), its envelope was obtained via Hilbert transform, and amplitudes were obtained as the average of an envelope in 0.5 s bins. Accelerometer signals were high-pass filtered (>1 Hz) on $x$-, $y$-, and $z$-axis signals separately to remove the effect of gravity acceleration. Subsequently, head acceleration was calculated as the mean absolute values of acceleration vectors in 0.5 s bins.

**Freeze detection**. Periods of freezing behaviour were detected using the Gaussian mixture hidden Markov model (HMM) with three hidden states. All behavioural sessions were concatenated; if the rat slept during the behavioural sessions, those periods were excluded from detection. In the HMM, heartrate, 0.5–5 Hz and 5–10 Hz band power of EOG, nuchal EMG amplitudes, and the logarithm of head acceleration were used as observed variables. HMM was optimised for each rat. The mean head acceleration in each hidden state was calculated, and periods in the state with the slowest acceleration were labelled as freezing. Freezing periods <5 s were removed, and those separated with a gap <1 s were concatenated. For visualisation in Fig. 1c and Supplementary Fig. 3, the fraction of time in freezing behaviour was calculated in each cue presentation period and in trisected bins of each inter-cue interval.

**Wavelet analysis**. The discrete wavelet transform was computed using the MATLAB wavelet software package (provided by C. Torrence and G. Compo [https://github.com/chris-torrence/wavelets]), and the wavelet power was then z-scored within each scale using means and standard deviations (SDs) within NREM sleep.

**Detection of hippocampal SWRs**. SWRs were detected on LFPs recorded in the vCA1, vHPC CA3 region, or ventral subiculum using a previously described method[43], with a minor modification. We used a slower frequency band and lower power threshold because the ripples were slower and weaker in the vHPC than in the dHPC[16]. To exclude gamma contamination, we adopted ripples co-occurring with sharp-waves. First, for each channel, the LFPs were bandpass-filtered (100–250 Hz), and their root mean squares (RMS) in 13.3 ms windows were z-scored using means and SDs within NREM sleep. Periods with signals >1.5 z were

used as candidate events on each channel. Candidates with peaks <4 z or shorter <30 ms were discarded, and those separated by <10 ms intervals were concatenated. The maxima of z-scored smoothed ripple power within the candidate and its corresponding time were considered as the ripple peak power and the ripple peak time of the candidate, respectively. Overlapped ripple candidates detected on the same shank were concatenated, and candidates >750 ms were discarded.

Sharp waves were detected using the difference between the most superficial and deepest channels within each shank. The difference was bandpass-filtered (2–40 Hz) and z-scored with the mean and SD within NREM sleep, and periods in which the signals were <−2.5 z for >20 ms were accepted as sharp-waves. Sharp waves >400 ms were removed, and sharp-wave troughs were determined as timepoints with the minima of the filtered signal. If no sharp-wave troughs were detected on the same shank during ripple candidates, the candidates were discarded. Moreover, overlapped candidates detected on different shanks were concatenated, and events <750 ms were regarded as SWRs.

For cases in which the candidates detected on multiple channels/shanks were concatenated, the ripple peak time of the candidate with maximum ripple peak power (in z-score) and the corresponding channel were regarded as the ripple peak time and the maximum ripple power channel for the resultant SWRs, respectively. Otherwise, a channel on which the candidate of interest was detected was regarded as the maximum ripple power channel of the SWRs. For each SWR, ripple troughs were detected in the bandpass-filtered (100–250 Hz) signals on the maximum ripple power channel, and the SWR peak time was defined as the time point of the ripple trough closest to its ripple peak time.

**Detection of amygdalar HFOs**. Amygdalar HFOs[18] were detected as previously described for amygdalar high-gamma detection[73], with a minor modification. The median of LFPs for each shank was bandpass-filtered (90–180 Hz), and RMSs of the filtered signals (window = 20 ms) were then converted to z-scores using the mean and SD within NREM sleep. Periods with a z-score >2 were classified as candidate events. Candidates with peaks <4 z were discarded, those with duration <30 ms were removed, and those separated by intervals <20 ms were concatenated. Detection was performed for each shank. The peak time of an HFO candidate was determined as the time of the peak on RMSs used for detection, and its peak height (in z-score) was used as the peak power. Overlapped candidate events on different shanks were concatenated, and events >750 ms were discarded. Because HFOs occur mainly during NREM[18], we accepted only candidates detected during NREM periods as HFOs, and those detected within awake periods were classified as aHFOs. For HFOs/aHFOs detected across multiple shanks, the shank with the highest peak power (in z-score) was labelled as the maximum amplitude shank. When HFOs/aHFOs were detected on one shank only, that shank was labelled as the maximum amplitude shank.

For each HFO/aHFO event, peak power and time were defined as those on the maximum amplitude shank. To estimate the peak frequency of each HFO, we calculated wavelet power (in 90–180 Hz band, 11 scales) as described above on the bandpass-filtered median LFP of the maximum amplitude shank. The wavelet power peak was then detected, and its corresponding frequency was assigned as the peak frequency of the HFO of interest.

**Detection of prefrontal gamma and ripple oscillations**. Prefrontal $\gamma_{slow}$, $\gamma_{fast}$, and cRipple oscillations[15] were detected on LFPs in the prelimbic cortex. The LFPs of each channel were bandpass-filtered (30–60, 60–90, and 90–180 Hz for $\gamma_{slow}$, $\gamma_{fast}$, and cRipples, respectively), and RMSs (window = 20 ms) of the filtered signal were z-scored with mean and SD within NREM epochs. Periods with a z-score >3 were used as candidate events, and the maxima of the RMS within the candidate and its corresponding time points were classified as the peak power and the peak time of the candidate, respectively. Candidates with a peak power <5 z were discarded, those with duration <50 ms were removed, and those separated by <30 ms intervals were concatenated. Detection was conducted for each channel separately; subsequently, overlapped events on different channels were concatenated, and events >750 ms were discarded. When concatenating multiple candidates, the peak time corresponding to the highest peak power (in z-score) across channels was assigned as the peak time of the oscillatory events.

**Detection of prefrontal slow-waves**. The mean LFPs across channels in the PL were calculated, and slow-waves were detected as positive deflections of bandpass-filtered (0.5–6 Hz) mean LFPs during NREM[9,32,43]. Sequences of upward–downward zero crossings of the z-scored bandpass-filtered signals were detected as onsets and offsets of candidates of slow-waves, and peaks were taken as maximum deflection between the onsets and offsets. If the deflections of the filtered signal at the peaks were <1.5 z or the durations from the onsets to the offsets were <100 ms or >1 s, these candidates were discarded. We rejected candidates unless the filtered signal monotonically increased from the onset to the peak and monotonically decreased from the peak to the offset. Then, we accepted candidates that were accompanied by a significant decrease of gamma activity as previously reported[34]. LFP on the PL channel with the highest delta power was bandpass-filtered (30–600 Hz), squared, and then smoothed with moving average filter (100 ms windows), which gives instantaneous PL gamma power. To compensate for the non-stationarity of gamma activity, we z-scored the gamma power locally (± 20 s from each slow-wave candidate peak). If the

minima of the locally normalised gamma power around a candidate peak (±50 ms) was smaller than $-0.5$ z, the candidate was regarded as a slow-wave. The typical intervals from onset to peak, peak to offset, and offset to the next peak were calculated as the median of these intervals within individual rats. For calculating the typical interval between slow-wave offset and the next slow-wave peak, we excluded events with an interval from slow-wave offset to the subsequent slow-wave onset of >4 s. We did not consider spiking activity in detecting slow-waves because the detection heavily depended upon the number of simultaneously recorded neurons, and a particular type of interneuron is active during DOWN states[34]. Instead, we detected OFF states solely defined based on PL spiking activity as described in the following section. To examine the modulation of spiking activity, we classified the detected slow-waves into tertiles within each session according to their amplitudes, and slow-wave peak-triggered average of spike counts pooled across cells were obtained in 3 ms bins. The spike counts were normalised with means in the concatenated periods of [−2000 ms, −1500 ms] and [+1500 ms, +2000 ms] from the slow-wave peaks in each session.

**Detection of prefrontal OFF states.** In sessions with >20 PL well-isolated cells, OFF states (cortical silent periods) were determined purely based on spiking activities[32,33]. Spikes from all recorded neurons were pooled and counted in 5-ms bins, then smoothed with a Gaussian filter ($\sigma = 30$ ms). To detect a significant decrease of the multiunit activity, we calculated the mean and SD of the smoothed spike counts within NREM and defined the edge and trough thresholds as mean − 1.5 SD and mean − 2 SD, respectively. Where the threshold(s) was <0 spike, it was set as a 0 spike.

We selected the periods during which the smoothed spike counts were equal to or lower than the edge threshold for >100 ms as candidates of OFF states. Candidates >1000 ms were discarded, and those whose minima of the spike counts were equal to or lower than the trough threshold were accepted as OFF states. Timestamp with the lowest spike counts within each OFF state was taken as the centre of the OFF state. If multiple bins contained minimum spike counts within each OFF state, the median of their timestamps was taken as the centre of the OFF states. OFF state modulation of spiking activity was examined by calculating OFF centre-triggered average of spike counts across cells in 3 ms bins. The results were normalised with means for the concatenated periods of [−1500 ms, −1000 ms] and [+1000 ms, +1500 ms] from the centre of OFF states within each session.

**Ensemble detection by independent component analysis.** The instantaneous activation strength of cell ensembles was estimated using ICA as described previously[24,25,74]. First, for each brain region, the z-scored firing-rate matrix of recorded neurons during a template epoch was obtained in 20 ms bins. Next, principal components analysis (PCA) was performed on this matrix, and significant components with corresponding eigenvalues exceeding the Marchenko–Pastur threshold[5] were identified. Next, the firing-rate matrix was projected onto the significant components, yielding the dimension-reduced projected matrix. An ICA was performed on the projected matrix using the FastICA package for MATLAB (available at https://research.ics.aalto.fi/ica/fastica/). Because all resultant independent components were used for the subsequent analysis, the number of independent components were equivalent to the number of significant principal components determined by PCA. The projection vector of each ensemble was calculated as the product of the PCA projection matrix and the weight vector of ICA, then normalised to unity length, and the sign of the largest absolute weight was set as positive, given that both the sign and the scale of the ICA output are arbitrary[74]. On the z-scored firing-rate matrix (in 20 ms bins) in a matched epoch **M**, the instantaneous ensemble activation strength at time $t$, $A_k(t)$, was calculated as $\mathbf{M}(t)^T\mathbf{P}_k\mathbf{M}(t)$. Here, $\mathbf{P}_k$ is the outer product of the normalised projection vector with its diagonal set to 0 to prevent high activation strength caused by the isolated activity of a single neuron with high weight to that pattern. We used the activities during conditioning sessions as templates unless otherwise specified. We performed these analyses in each brain region separately. The summary of the numbers of detected ensembles is presented in Supplementary Table 2.

We assessed the member cells of each ensemble as the cells with the top five weights of the projection vector. Cells that were members of at least one ensemble detected in conditioning sessions were labelled as ensemble-participating-cells.

**Explained variance.** Explained variance (EV) and reverse EV (REV) were calculated as described previously[6,75]. First, we calculated spike counts of each cell in 20- or 100 ms bins during pre-cond NREM, conditioning sessions, and post-cond NREM, then obtained correlation coeffects of the spike counts within each of the epochs mentioned above for each BLA–PL5 and vCA1–PL5 cell pairs. Next, we calculated EV as the square of the partial correlation coefficient between the correlation coefficients during conditioning and post-cond NREM under the control of correlation coefficients during pre-cond NREM. Similarly, REV was calculated as the square of the partial correlation coefficient between the correlation coefficients during conditioning and pre-cond NREM under the control of correlation coefficients during post-cond NREM. EV and REV were calculated for each rat.

**Ensemble coactivation.** To evaluate the coactivation of cell ensembles, we calculated CCGs between instantaneous ensemble activation strengths (bin size =

20 ms) as

$$\mathrm{CCG}(\tau) = \frac{1}{T}\sum_{t=1}^{T} f(t)g(t+\tau) \quad (1)$$

where $T$ is the number of the analysed bin and $f(t)$ and $g(t)$ are z-scored instantaneous ensemble activation strengths at time $t$. Because the signals were z-scored, $CCG(\tau)$ gives the Pearson correlation coefficient between the signal $f$ and the time-shifted signal $g$. For presentation, CCGs were smoothed with a Gaussian kernel ($\sigma = 20$ ms), except for the data reported in Fig. 2c. Subsequently, the maximum deflection of each CCG was detected within a ±100 ms window around time 0.

The significance of the deflection was evaluated based on chunk shuffling. First, one of the signal pairs was divided into 2 s chunks. Then, the order of the chunks were randomly shuffled, and CCG was calculated. This method preserves the finer (<2 s) structure of auto-correlograms (ACGs), which is important because the non-uniformity of ACGs may be inherited to the CCG[76]. The shuffling was iterated 500 times, and 99% CIs of the maxima and minima of the CCG in a range of ±100 ms were then estimated. The CCG peak/trough was identified as the point at which the absolute values of differences between the actual CCG and the shuffled mean reached the maximum. If the actual CCG value of the peak/trough was larger or smaller than the 99% CIs, the peak or trough was regarded as significant.

Ensemble pairs with significant peaks and troughs during post-cond NREM were labelled as coupled-ensemble-pairs and inverse-coupled pairs, respectively. Ensemble pairs that did not have significant peaks during post-cond NREM were labelled as non-coupled pairs, including inverse-coupled pairs, unless stated otherwise.

To confirm the consistency of the results across animals in the changes of the proportion of coupled-ensemble-pairs between pre- and post-cond NREM, we performed "leave-one-out" analysis of the proportion changes in which one rat was excluded from the analysis (Supplementary Fig. 5a). To check the robustness of the results against the thresholding, we repeated the "leave-one-out" analysis for a proportion of coupled- and inverse-coupled-ensemble-pairs detected with 95 and 99% CIs. The 95% CI criterion was used only for Supplementary Fig. 5a.

To assess the contribution of SWRs, HFOs, or cRipples to the coactivations of coupled-ensemble-pairs, we calculated event-excluded CCGs between the instantaneous ensemble activation strength of ensembles by removing time bins that contained the oscillatory events of interest from the CCG calculation. To investigate whether significant coactivation occurred outside the oscillatory events of interest, we performed chunk shuffling (500 times), as described above, on the event-excluded CCGs to test whether significant peaks were preserved.

To examine whether the coactivation occurred preferentially within the oscillatory events of interest, we calculated peak drops as differences in the peak height between event-excluded CCGs and CCGs with entire NREM, and their significances were tested as follows: First, we performed a surrogate event-excluded CCG analysis with randomly jittered events (jitters distributed uniformly in ranges of ±500 to ±2500 ms) 500 times for each event type, and the peak drop of each CCG was calculated. Finally, we tested whether the actual peak drops were larger than the 99.5th percentiles of the peak drops in jittered events.

To compare the proportions of significantly coactivated ensemble pairs in conditioning versus cue-retention/extinction sessions, CCGs of instantaneous ensemble activation strengths during these sessions were calculated. To match the analysed duration, conditioning sessions (45.8 min), together with the cue-retention part (28.4 min) combined with the first 17.4 min of the extinction part of the cue-retention/extinction sessions were used. To determine the reappeared-pairs, the CCGs of instantaneous ensemble activation strengths during the entire cue-retention/extinction session were calculated. The significance of each ensemble pair's CCG peak was examined using chunk shuffling (500 times) as described above.

**Triple-activation of ensembles across brain regions.** To evaluate simultaneous reactivation across the BLA, vCA1, and PL5 ensembles, we extended the CCG analysis to ensemble triplets in a manner similar to the methodology previously established for spike triplet analysis[77]. First, we defined triple CCG as follows:

$$\mathrm{Triple\text{-}CCG}(\tau, \sigma) = \frac{1}{T}\sum_{t=1}^{T} f(t)g(t+\tau)h(t+\sigma) \quad (2)$$

where $T$ is the number of the analysed bin (bin size = 20 ms) and $f(t)$, $g(t)$, and $h(t)$ are z-scored instantaneous ensemble activation strengths at time $t$. For presentation, triple CCGs were smoothed using a two-dimensional Gaussian kernel ($\sigma = 10$ ms). The maxima of triple CCGs with corresponding time gaps in the range of $|\tau|$, $|\sigma|$, $|\tau-\sigma|$ <100 ms, which corresponded to triple-activation with all participant ensembles activated within 100 ms time gaps, were taken as peak values.

To examine the significance of peak values, we performed chunk shuffling (as in the CCG analysis) of all 3 signals to generate 500 surrogate data. When the peak value of actual data was larger than the top 0.5th percentile of the peaks of the surrogate data, the triplet was labelled as a coupled-ensemble-triplet.

The triple CCG defined above can be interpreted as an $h(t)$ weighted CCG between $f(t)$ and $g(t)$ [if $h(t)$ is constant (i.e., $h(t) = \bar{h}$), $\mathrm{Triple\text{-}CCG}(\tau, \sigma) = \bar{h} \cdot \mathrm{CCG}_{f,g}(\tau)$, where $\mathrm{CCG}_{f,g}(\tau)$ is pairwise CCG between $f(t)$ and $g(t)$]. Because $h(t)$ is the z-scored instantaneous activation strength, which remains near 0 most of the time (Fig. 2b), $h(t)$ works as a mask of non-structured "solo" activation, and only the coactivation between

$f(t)$ and $g(t)$ that occurred when $h(t)$ is much greater than 0 contributes to positive values (i.e., peaks) in the triple CCG. Due to this masking effect, triple CCG can be more robust than pairwise CCG of partial-pairs against "solo" activation; an ensemble triplet may have a significant peak in the triple CCG even when none of the partial-pairs has a significant CCG peak (Supplementary Fig. 15).

To compare the proportions of significantly triple-activated ensemble triplets during conditioning versus cue-retention/extinction sessions, triple-CCGs of instantaneous ensemble activation strengths during these sessions were calculated. The analysed duration was matched by truncating cue-retention/extinction sessions as was performed for CCG analyses (see "Ensemble coactivation" subsection). The significance of triple-CCG peaks in these behavioural sessions was examined using chunk shuffling as described above.

**Triple CCG and CCG of partial-pairs on simulated data.** To examine how the effect of triple-activation, coactivation, and solo activation events impact the triple CCG and CCGs of partial-pairs, we generated three 60 min long binary trains (bin size = 20 ms). For simulation with triple-activation, triple-activation events were randomly added at the rate of 0.1 min$^{-1}$, which was approximately matched with triple-activation event rates of coupled-ensemble-triplets in post-cond NREM. For simulation with coactivation, coactivation events were added randomly at the rate of 1 min$^{-1}$ for each combination. Solo activation events were added randomly at rates of 1, 2, 5, 10, 20, 50, or 100 min$^{-1}$ for each signal train separately. For simplicity, zero time-lag was assumed for triple-activations and coactivations. For each condition, the simulation was repeated 500 times to obtain the distribution of maxima of triple CCG and CCGs of partial-pairs.

**Detection of the activation, coactivation, and triple-activation timings of the ensembles.** The instantaneous activation strength of each ensemble was z-scored; then, the activation events were detected as activation strength peaks >5 z, and the timepoints of the corresponding local maxima were assigned as timestamps of the ensemble activation events. To examine the significance of activation event rates, we calculated the instantaneous activation strength of surrogate ensembles (500 surrogates for each ensemble) obtained by random permutation of the projection vector. Subsequently, activation events were detected for surrogate ensembles as activation strength peaks >5 z, and their event rates were obtained during pre- and post-cond NREM. Significantly activated ensembles were determined as ensembles whose event rates were higher than the top 2.5th percentile of the event rates of the corresponding surrogates.

To detect the time points of coactivations, first, we determined the optimal temporal shift of each ensemble pair by detecting peak timing on the CCG of the instantaneous ensemble activation strengths during post-cond NREM. Next, we calculated the instantaneous coactivation strength as the product of the z-scored instantaneous ensemble activation strength with the optimal temporal shift. Coactivation events were detected as peaks >25 z$^2$ on the instantaneous coactivation strength. For each coactivation event, the time corresponding to the maxima of the coactivation strength was classified as the timestamp of the coactivation event.

Similarly, the instantaneous triple-activation strength was calculated as the product of the time-shifted activation strength, with shifts determined based on the time gaps of triple CCG peak during post-cond NREM, followed by the detection of triple-activation events as peaks >125 z$^3$ on it. For each triple-activation event, the time corresponding to the maxima of the triple-activation strength was determined as its timestamp. To confirm the consistency of the enhancement of triple-activation after fear conditioning, we performed "leave-one-out" analysis in which rats were excluded one by one from the analysis.

The time gaps between coactivated and triple-activated ensembles were not necessarily the same during sleep and awake periods. Thus, for analysis of coactivation/triple-activation during behavioural sessions, we re-calculated the optimal time shift using CCGs/triple CCGs within the behavioural sessions of interest, followed by the detection of coactivation/triple-activation events, as described above.

Shock- and cue-triggered histograms of the activation/coactivation/triple-activation occurrence rate were calculated in 20 ms bins and then smoothed with a Gaussian kernel ($\sigma = 20$ ms). In the homecage sessions that preceded and followed the conditioning sessions, all NREM epochs were concatenated within pre- and post-cond homecage sessions, and the occurrence rate of activation/coactivation events was calculated in 5 min bins. That of triple-activation events was calculated in wider (10 min) bins, given their low event rates.

**Freezing behaviour and cue presentation modulation of cell firing, ensemble activation, and ensemble coactivation.** The number of spikes of each neuron, instantaneous activation strength of ensembles that were identified based on activity during conditioning sessions, and instantaneous coactivation strength among the ensembles were calculated in 20 ms bins during the entire conditioning or entire cue-retention/extinction session after the first tone onsets. Bins containing sleep epochs were excluded from the analysis. Each bin was labelled as freezing or non-freezing, and the means within each bin type were obtained. The differences were calculated to evaluate freezing modulation. To investigate the significance of the difference, we randomly shuffled bin labels 500 times and estimated the

distribution of difference of the surrogate data. If the actual difference was larger/smaller than 97.5/2.5 percentiles of the estimated distribution, these cells/ensembles/ensemble pairs were regarded as positively/negatively modulated by freezing behaviour.

Similarly, modulation of cue presentation was tested by labelling bins as within or outside cue presentation and repeating the same procedure for the modulation by freezing behaviour. Modulations by cue presentation in the baseline sessions were assessed using the same procedure. To examine modulations by the movement of the animals in the homecage sessions, we labelled awake bins as immobile or moving and repeated the procedure described above. We detected immobile wakefulness as awake periods during which head acceleration was <mean + 2 SD of head acceleration during freezing behaviour in behavioural sessions. The remaining awake bins were treated as moving periods.

**Coactivation-triggered average of the wavelet power.** We selected one channel for each probe, performed wavelet transform on the LFP in the 0.5–330 Hz band (94 scales), and normalised wavelet transforms as described above (see "Wavelet analysis"). Channels with a maximum theta (6–10 Hz) and delta (0.5–4 Hz) power were used for hippocampal and prefrontal wavelet analysis, respectively. For amygdalar wavelet analysis, the channel with maximum power in the gamma band (50–100 Hz) among BLA channels was used. If no BLA channels were available, lateral nucleus of the amygdala channels were used instead. The coactivation-event-triggered average of the wavelet power was calculated for each coupled-ensemble-pair and was then averaged within each region pair.

**Oscillatory-event-triggered average of the coactivation/activation strength.** The SWR-, HFO/aHFO-, PL slow-wave-, PL $\gamma_{slow}$-, PL $\gamma_{fast}$-, and PL cRipple-peak-triggered average, and PL OFF centre-triggered average of the coactivation was calculated as a peri-event-triggered average of the instantaneous coactivation strength. The normalised coactivation strength was obtained as a z-score of the peri-event-triggered average in a ±2 s window. Peaks of the SWR-, HFO-, and cRipple-triggered average of the coactivation strength were detected in the range of [−100 ms, +100 ms] from the oscillatory event peaks. Peaks of the PL slow-wave peak-triggered average of the BLA–PL5 and vCA1–PL5 ensemble-coactivation strengths were detected in the range of [−400 ms, 0 ms] and [−200 ms, +200 ms] from PL slow-wave peaks, respectively.

Similarly, the SWR-, HFO-, and cRipple-peak-triggered average of the activation was calculated as the peri-event-triggered average of the instantaneous ensemble activation strength. The peri-event-triggered average was calculated in a ±2 s window for pre- and post-cond NREM. For visualisation, the results were z-scored within each ensemble using the mean and SD across pre- and post-cond results and plotted in the range of ±400 ms.

The peak times of SWRs and HFO were used to calculate the PL slow-wave peak-triggered average of the SWR and HFO event rates, and the event rate peaks were detected in the range of [−300 ms, 0 ms] from PL slow-wave peaks.

**Partner region of ensembles and cells.** If an ensemble participated in at least one coupled-ensemble-pair, the ensemble was considered a coactivation-participating-ensemble, and its coupled region was determined as a region from which the partner ensemble was identified. Member cells of coactivation-participating-ensembles were labelled as ensemble-coactivation-contributing cells, and the region of the partner ensemble was defined as the coupled region of the member cells. Note that a given cell can be a member of multiple ensembles and may have more than one coupled region.

**Modulation of cell firing.** Because the firing rates of individual neurons are typically log-normally distributed[38], the logarithm values were used to calculate the mean and SD of firing rates and to compare firing rates across cell populations and behavioural states. For analyses that used the ratio of firing rates, such as firing-rate modulation and the modulation index, the mean firing rates were calculated on a linear scale to avoid negative values.

The firing-rate modulation by SWRs and HFOs was measured as follows: $\frac{\overline{FR_{event}}}{\overline{FR_{NREM}}}$, where $\overline{FR_{event}}$ and $\overline{FR_{NREM}}$ are the mean firing rates within the events of interest (SWRs or HFOs) and the entire NREM, respectively.

**Statistics.** To assess the significance of CCG peaks/troughs of instantaneous ensemble activations (range [−100 ms, +100 ms]) and spike times (range [+1 ms, +4 ms]), we estimated the 99% CI of maxima and minima of CCGs (range for ensemble activation [− 100 ms, + 100 ms] and for spike times [−5 ms, +5 ms]) of the surrogates (global bands). We then tested the significance against the global bands to avoid multiple comparison issues (see also "Classification of excitatory and inhibitory neurons" and "Ensemble coactivations"). Similarly, the significance of the actual triple CCG peak was tested against 99% CI of triple CCG peaks of the surrogate data (see also "Triple-activation of ensembles across brain region"). To examine the non-uniformity of triple CCG peak position, first, we compared the distribution with uniform distribution with the $\chi^2$ test. Then, the significance of defection in each bin was evaluated using a Poisson distribution with Bonferroni corrections.

As the firing rates of individual neurons are log-normally distributed[38], we performed two-way ANOVA and the post hoc Tukey–Kramer test (TK test) for firing rates on logarithms of the data. Our measurements other than firing rates were also highly skewed. Thus, we tested the statistical significance using non-parametric tests, such as the Wilcoxon rank sum test, WSR-test, Friedman test, Kruskal–Wallis test, and two-way ANOVA on the ranks. Only when a significant effect was detected with the Friedman test, Kruskal–Wallis test, or ANOVA, a post hoc TK test was performed.

Non-parametric Fisher's exact test and Kolmogorov–Smirnov test were used to assess the difference in proportions and distributions, respectively. Fisher's exact test with n × m contingency table was performed with a custom MATLAB script developed by Guangdi Li (available at https://jp.mathworks.com/matlabcentral/fileexchange/24379-fisher-s-exact-test-with-n-x-m-contingency-table).

For the analysis of coupled-ensemble-pairs (except the detection of coupled-ensemble-pairs itself), we included the BLA–PL5 and vCA1–PL5 ensemble pairs only for statistical rigour because significant changes in the CCGs of ensemble activities in the population level were detected only in these region pairs.

The Friedman test, Kruskal–Wallis test, ANOVA, and $\chi^2$ test were one-sided, and all other statistical tests performed in the study were two-sided. For box plots, data points > upper quartile + 1.5 interquartile range (IQR) and those < lower quartile − 1.5 IQR were regarded as outliers.

**Reporting summary**. Further information on research design is available in the Nature Research Reporting Summary linked to this article.

## Data availability
The datasets supporting this study will be deposited to a public repository when the ongoing studies using the same dataset are published. Meanwhile, the data are available from the corresponding authors upon request. Source data are provided with this paper.

## Code availability
All codes for the manuscript are available at https://github.com/HiroMiyawaki/Miyawaki2022_NatCommun [78].

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

## Acknowledgements

We thank Joshua Johansen for his advice on eyelid electrical stimulation and Sakura Okada and Nobuyoshi Matsumoto for their advice on ECG recordings. We are grateful to Kamran Diba, Antonio Fernández-Ruiz, Nathaniel Kinsky, Takuma Kitanishi, Hideyuki Matsumoto, Sébastien Royer, Yuichi Takeuchi, and Brendon O. Watson for discussion and comments on the manuscript. We thank the Research Support Platform, Osaka City University (OCU) Graduate School of Medicine for letting us use the microscopes. This work was supported by JSPS KAKENHI (20K06860, 20H05477, and 19H04986 to H.M.; 21H00209, 20H03356, 19H05225, 16H04656, and 16H01279 to K.M.), GSK Japan Research Grant (to H.M.), The Naito Foundation (to H.M. & K.M.), The Takeda Science Foundation (to H.M. & K.M.), Toray Science Foundation (to K.M.), The Uehara Memorial Foundation (to H.M. & K.M.), OCU 'Think globally, act locally' Research Grant for young researchers 2018 through the hometown donation fund of Osaka City (to H.M.), OCU Strategic Research Grant 2019 for young researchers (to H.M.), and OCU Strategic Research Grant 2021 for basic research (to H.M.).

## Author contributions

Conceptualisation: H.M. and K.M.; investigation: H.M.; visualisation: H.M.; supervision: K.M.; writing—original draft: H.M.; writing—review & editing: H.M. and K.M.; funding acquisition: H.M. and K.M.

## Competing interests

The authors declare no competing interests.
