## [Peer review file · Nature Communications]

REVIEWER COMMENTS

Reviewer #1 (Remarks to the Author):

Report

The aim of the study was to analyze co-activation of different brain areas involved in the consolidation of a memory trace. Subjects were trained first in Contextual Fear Conditioning and their neuronal activity was recorded (this continued all throughout the experimental procedure). Afterwards they were tested for context and cue and subjected to an extinction session, followed by a retention of extinction test. Between acquisition and testing, animals were returned to their homecages and activity during sleep was registered. From the analysis of single unit activity, ensembles of neurons active at the same time were detected in the three structures of interest (amygdala, ventral hippocampus and prelimbic cortex) though they showed different temporal dynamics. High frequency oscillations and ripples were found after acquisition, specially in sleep periods, where this type of activity is associated with the consolidation of memories (replay of the events). Besides, at the time of memory retrieval, the emergence of slow gamma activity was observed.

The key findings are the ensembles recorded, the type of inter-regional activation following the acquisition and how this serves to consolidate a recently acquired memory trace in a preexistent neuronal network (memory co-opts an existing neuronal ensemble). Furthermore, at the time of retrieval, network activity is coordinated by fast oscillations.

What is most remarkable is that they show that a memory is consolidated in a preexistent network. Authors mention that their findings are also compatible with the hypothesis of the Hp as an indexer for memories. Overall, the ideas presented here are interesting and neuronal activity recording and analysis is solid.

Major comments

On the analysis of memory formation:

The technique used here (like all the other techniques to register freely moving animals) to record neuronal activity has a time limitation: the activity of a particular group of neurons can be followed over the hours but not over longer periods like days or weeks. Authors comment on this issue in line 423. However it should be noted that memory acquisition and consolidation are a processes that have many facets, many of them taking several hours or even days or weeks to be completed. Perhaps it is recommendable to make a mention to the temporal aspect of memory formation, specially considering that the whole experiment was carried out in less than 24 h. In that sense, it would be reasonable to refer to "the initial stages" of memory formation rather than taking it as a finished process. Moreover, in the present work there are many citations related to memory in the references (~50% of the total) and it is even cited Squire's classical paper (1986), so it is desirable to mention in the discussion the relation of present results in function of more traditional terminology of the field (particularly the evolution of memory traces over time).

In the abstract and the discussion it is mentioned that the results are compatible with the "index theory", this can certainly be the case but it cannot be ruled out the chance of the vHp being not a mere indexer but also a structure involved in the storage of the trace.

On the behavioral training:

There are no references to explain the type of training used here nor a validation of the parameters selected to train the animals (amount and intensity of the shocks, intensity (dB) of the cue, etc.). In the present work animals receive 12 trains of pulses of 4.6-5.1 mA and 32 (?) tones are used for extinction of the memory trace. For example, in a previous work by Hugues and colleagues (Synapse, 2006 Sep 15;60(4):280-7, doi: 10.1002/syn.20291) a similar methodology is used but less shocks of minor intensity are used (5 US presentations of 3.5 mA and they are extinguished by 15 CS presentations). It is a possibility that this stronger training is fundamental to observe the activity at

the ensembles' level but it is not mentioned. It would be suitable to include more detailed data on this subject.

After a training session in the CFC, subjects are tested and the context elicits a >50% freezing response. In the cue retention test there is also a high freezing response. Though rats are later exposed to an extinction session, it appears to be insufficient to extinguish the initial memory trace because high percentages of freezing are still observed in the extinction test. Another possibility is that subjects are forming a mixed "tone-place" engram, given the strong saliency of the initial training, and this is generalized in the context retention exposure to the chamber B that is followed by Cue retention and extinction in chamber A. A similar phenomenon is described in Sachella et. al (2021, <https://doi.org/10.1101/2020.07.12.197319>).

On the ensembles:

Supplementary figure 3 is important to understand the amount of ensembles were distinguished from the total number of registered cells. The graphs and numbers are not very clear to follow and they are key to the paper.

Minor comments

There are a few omissions in the methodology that would be necessary to include:

- how long is the recovery period after surgery? Are animals housed singly? Are the implants chronically fixed or is it an acute intervention (i.e.: does the surgery take place the same day of the experiments or are there recovery days before the beginning of the recordings?).
- the experimental protocol is hard to follow (both in the methods and in Figure 1d). It would be useful to clarify it a little more. The use of the word "session" is a bit confusing because at a first read, sessions appear to happen in different days.

- Graphs in Figure 1 c and d are expressed in seconds or minutes making them not clear enough. Particularly in Figure 1d, it would be recommendable to include a separate timeline expressed in hours to show the elapsed time between training, testing and extinction.

Conclusion

The results shown here are very interesting. All the electrophysiology recordings and the analysis are robust and clearly explained. In case authors want to extend their studies, it would be interesting to analyze the freezing response induced by the initial training and evaluate how long does this memory last (only behavioral testing, no electrophysiological recordings) and if there is a mixed "tone-place" memory to achieve a deeper characterization of the behavior.

Reviewer #2 (Remarks to the Author):

Miyawaki and Mizuseki investigated interactions between neuron ensembles in the prefrontal cortex, amygdala, and ventral hippocampus during and after fear conditioning in rats. With large-scale multi-site electrophysiological recording, they identified that all three regions contained groups of neurons that fired together during fear conditioning as well as subsequent non-REM sleep. Notably, such ensemble co-activation in the mPFC was temporally coupled with that in the amygdala and hippocampus. The inter-region co-activation was detected more frequently after fear conditioning than the homecage period and reappeared during the retention test. In the amygdala and mPFC, but not the hippocampus, the co-activation was also detected before fear conditioning. Based on these findings, the authors argue that frightening experiences are rapidly encoded by a set of amygdala and prefrontal neurons with robust baseline connectivity and that these neurons are coupled with hippocampal neurons during subsequent sleep.

This study provides the first evidence for fine-time scale, inter-region interactions during the

acquisition and consolidation of aversive associative memory. The experiments are technically challenging, and the authors' expertise in sophisticated analytical approaches are instrumental to uncovering cell assembly dynamics supporting fear memory. In addition, detailed characterizations of oscillations and spiking activity during sleep are of great value to researchers interested in neuronal mechanisms supporting consolidation during sleep. The analyses appeared to be conducted carefully with appropriate control measures. I only have a few concerns about data interpretation.

1) Based on the results in Fig. 7, the authors argue that "co-activation of BLA-PL5 coupled ensemble pairs and triple-activation of BLA-vCA1-PL5 coupled ensemble triplets form rapidly during memory acquisition, whereas co-activation of vCA1-PL5 coupled ensemble pairs develops during the sleep period that follows those experiences." This argument does not convince me for two reasons. Firstly in Fig. 7b, the co-activation rate of vCA1-PL5 pairs was increased after the shock. Although the rate was not different from that of non-coupled pair, this finding suggests that vCA1-PL5 pairs (and other cell pairs) were rapidly formed during the conditioning. Secondly, vCA1-PL5 (and other) pairs possibly encode other task features such as the CS, conditioning chamber, and freezing. Therefore, the formation of vCA1-PL5 pairs may be triggered by these features rather than the shock. The authors need to strengthen evidence if they want to claim that vCA1-PL5 ensembles develop only after learning.

2) The model proposed in Fig. 10 needs additional clarifications.

a. The model gives me an impression that the pre-configured cell assembly includes local neurons in each region, and the binding of the assembly between the regions occurs during experiences. This does not fit with the data in Supplementary Fig. 16c showing the activation of BLA-PL neurons before the conditioning. Please clarify.

b. I do not follow how the present data support that cell assembly in the vHPC functions as an index for a specific experience. The point of confusion is that the authors argue "co-activation of vCA1-PL5 coupled ensemble pairs develops during the sleep period." According to the original theory, the index should be formed during experiences to facilitate the co-activation of other assemblies during subsequent consolidation processes.

c. The authors described their model very briefly in the first paragraph of the discussion. The paragraph is quite dense. I would suggest that they move this paragraph toward the end of the discussion and elaborate on how the present data fit with each prediction/argument.

Reviewer #3 (Remarks to the Author):

The manuscript by Miyawaki & Mizuseki examined the inter-regional coactivations of preconfigured local ensembles following a fear conditioning task. The data show evidence of inter-regional activation during a brief period of sleep following learning. These coactivations relied on fast oscillations – Bla HFOs and vCA1 SWR contribute to BLA-PL5 and vCA1-PL5 ensemble coactivations during sleep. cRipples contributed to both coactivations. Coactivation in the BLA and PL5 were configured prior to memory but developed with experience in the vCA1. The authors also examined triple activation which was present during specific time windows and was also enhanced by SWRs, HFOs, and cRipples. The likely activation followed the order of vCA1, Bla and PL5. Interestingly, coactivated events were greater to shock presentation in BLA-PL5 pairs and triplets but not in vCA1-PL5 pairs. These findings are important for understanding how three critical regions involved in fear learning are coactivated or work together as part of a network to support learning and memory.

Uncovering how different neural areas coactivate to support memory is a fundamental question for system neuroscience. The data presented directly address this question. They are exciting and the

analyses are thorough, clear and appropriate. Below I have a few minor comments/questions for the authors.

The authors compare the presence of the coactivated pairs following fear learning against a neutral condition that equate for novelty. This is a great comparison which shows that the proportion of coactivated pairs changed between pre- and post-baseline NREM sessions but not the CCG peak height of ensemble pairs. One question for the authors is whether they think that the CCG peak height of ensemble pairs would be modulated if the neutral experience consisted of associative learning between two neutral stimuli (the HPC has been implicated in sensory learning) or whether just mere exposure to an emotional event could do the same (shock exposure). While I am not suggesting these studies need to be done, I do feel the authors should discuss how specific the signal is to associative fear learning in particular.

Cue retention seems to run into the extinction session as depicted in Fig 1 given the x-axis, but the methods suggests that there was an interval of 2.5-2.6h rest/sleep. It is unclear if cue retention and cue extinction were run in one session or not and if the former then how the distinction is made at the level of the analyses is also unclear. If indeed these sessions were separate then they should be depicted as such in the Figure.

Additional information within the text on how the cells were 'modulated' by fear (p.4). This is important to clarify because there is no information in the text nor in the supplemental figure legend that explains how modulation was examined.

What is the functional significance of the vCA1-PL5 coactivation developing rapidly after the experience? There is a lot of discussion/speculation as to why and how this happens at the network level but less so at the functional level.

The authors should add a table showing the number of co-active pairs and co-active triplets per animal.

Reviewer #4 (Remarks to the Author):

This is a well conducted study that aims to address some timely key open questions in the learning and memory field such as how inter-regional neural ensemble interactions emerge, are reactivated during post-learning sleep, and relate to fast network oscillations. Towards that end, fear conditioning and fear memory retrieval were used as model paradigm in rats, while large scale electrophysiological recordings were performed in brain regions implicated in this behavior, including BLA, prelimbic cortex layer 5 (PL5) and ventral hippocampus CA1 (vCA1) followed by in depth analyses. The study uses a paradigm with a relatively short consolidation period (several hours) before the fear memory test and excludes analysis of the ensuing extinction sessions. The key novel findings that are generally well supported by the data are: Identification of inter-regional BLA-PL5 and vCA1-PL5 and BLA-vCA1-PL5 ensembles during learning, that are coactivated during post-learning Non-REM sleep and associated with distinct fast network oscillations also during memory retrieval. Coactivation-participating ensembles developed with distinct time-courses and were preconfigured in BLA and PL, but emerged later in vCA1. Moreover, ensemble-coactivation contributing cells displayed faster firing rates, undermining the notion that higher excitability may determine allocation of neurons to memory traces.

I have several comments and questions:

1. Generally, it may be worthwhile to streamline the manuscript to the core three regions for which ensembles, ensemble pairs and triplets were analyzed (BLA, PL-Layer 5 and vCA1).
2. Although deducible from the tables, it would help to mention the number of animals for which

individual analyses were done throughout. It would also be helpful to know if the behavior was similar in the subsets of analyzed animals.

3. For the initial identification of FC ensembles, was the whole conditioning session used (c.f. Suppl Fig 16)? Which part of the cue-retention/extinction session was used for analysis of modulated ensembles? It should be noted that changes in neural activity can precede behavioral changes (i.e. in extinction). Are the same or different ensembles modulated by cue and/or freezing? It would be very interesting to see what happens to the FC-ensembles and pairs/triplets during extinction.

4. The identified ensembles/ensemble pairs also include inhibitory cells. What is their proportion and does this differ between regions and regional ensemble pairs?

5. The proportion of ensemble pairs is low in REM and does not differ between pre- and post-cond. REM. I was wondering if this has to do with the brevity of REM compared to NREM and the fact that generally NREM dominates the early sleep phases, also given the time evolution of ensemble activation?

6. The temporal delays between BLA-PL5 and vCA1-PL5 ensemble pairs of 20-23 ms on average appear quite long for assuming monosynaptic connectivity and seem to be longer than those in the cited studies. Despite the well-established direct interconnectivity between these regions, there may be other explanations for coactivation of these ensembles.

7. The authors show shock-triggered averages of ensembles/ensemble pairs and triplets events during FC. I am not sure they can simply suggest "that these ensembles are related to memory of shock events" and that the vCA1-PL5 pairs are not involved in the initial acquisition. It would be nice to see cue-triggered averages during FC and freezing-triggered averages during retention (expanding on Figs 7g,h and 8f-h).

8. In Suppl Fig 19b, I did not understand which homecage session(s) this refers to. The NREM firing data seem to differ from those shown in Fig 9c (at least for BLA) and data seem only significant when wake, REM and NREM are considered together. Label in middle graph BLA excitatory cells "coupled to BLA" should read "coupled to PL5".

9. The results on firing modulation by SWRs and HFOs should be reworded including the conclusion statement, according to presented data/stats. Suppl Fig 20a,b suggests that this modulation is enhanced in excitatory BLA ensemble cells coupled to PL5 and not "largely unchanged". For PL5 ensembles, there is a general increase in modulation of excitatory cells within coupled and other ensembles, and it looks like there is a specific increase in modulation of inhibitory cells in coupled ensembles.

10. Could the authors based on their data speculate on the roles of inhibitory cells in ensembles?

We wish to express our gratitude to the reviewers for their insightful comments on our manuscript, which we have revised thoroughly in response to their suggestions.

In response to Reviewer #1's comments, we clarified that we investigated the initial stage of memory consolidation and our data cannot rule out the possibility that the hippocampus itself stores the memory traces. We also reviewed the literature to verify the parameters of our behavioural experiments, added references to the Methods, and discussed the possibility that strong shock intensity may induce mixed "tone-place" memory traces. Additionally, we revised Fig. 1 and the Methods section to clarify the details of the experiment.

In response to suggestions by Reviewer #2, we re-evaluated the shock modulation of vCA1–BLA ensemble coactivation and revised the Results, Discussion, and Fig. 10. We also examined the cue/freeze modulation of ensemble coactivation during conditioning (Supplementary Fig. 19). Additionally, we have discussed the extent to which our results are consistent with the hippocampal index theory and added a detailed explanation of the presented model.

In response to comments from Reviewer #3, we discussed that it is still an open question whether the coactivation observed in this study can also be induced by various types of learning, rather than an association involving strong aversive experience. Further, we have analysed the cue and freeze triggered modulation of coactivation separately in cue-retention and extinction parts of the sessions (Supplementary Fig. 22 and Supplementary Table 6).

In response to comments from Reviewer #4, the similarity of the behaviour across subset of animals was confirmed (Supplementary Fig. 3). We analysed the cue and freeze modulation of coactivations in cue-retention and extinction parts of the sessions separately (Supplementary Fig. 22 and Supplementary Table 6). We also analysed the time evolution of coactivation during conditioning (Supplementary Fig. 17 and Supplementary Table 5). Further, we compared the proportion of inhibitory members across ensembles (Supplementary Fig. 24b). We also showed that the lack of coactivation during REM sleep is not due to the fact that REM sleep, compared with NREM sleep, is shorter and temporally more distant from the conditioning (Supplementary Figs. 7, 20). Lastly, we incorporated the SWR- and HFO-modulation on firing of cells in the BLA and vCA1 to the main figure (Fig. 9) and revised the description of these modulations.

Our point-by-point responses follow below, with the reviewers' comments shown in bold Arial font and our responses in regular Times New Roman font. The text added or edited in the revised manuscript is indicated in blue font, both in the rebuttal and revised manuscript.

Reviewer #1 (Remarks to the Author):

Report

The aim of the study was to analyze co-activation of different brain areas involved in the consolidation of a memory trace. Subjects were trained first in Contextual Fear Conditioning and their neuronal activity was recorded (this continued all throughout the experimental procedure). Afterwards they were tested for context and cue and subjected to an extinction session, followed by a retention of extinction test. Between acquisition and testing, animals were returned to their homecages and activity during sleep was registered. From the analysis of single unit activity, ensembles of neurons active at the same time were detected in the three structures of interest (amygdala, ventral hippocampus and prelimbic cortex) though they showed different temporal dynamics. High frequency oscillations and ripples were found after acquisition, specially in sleep periods, where this type of activity is associated with the consolidation of memories (replay of the events). Besides, at the time of memory retrieval, the emergence of slow gamma activity was observed.

The key findings are the ensembles recorded, the type of inter-regional activation following the acquisition and how this serves to consolidate a recently acquired memory trace in a preexistent neuronal network (memory co-opts an existing neuronal ensemble). Furthermore, at the time of retrieval, network activity is coordinated by fast oscillations.

What is most remarkable is that they show that a memory is consolidated in a preexistent network. Authors mention that their findings are also compatible with the hypothesis of the Hp as an indexer for memories. Overall, the ideas presented here are interesting and neuronal activity recording and analysis is solid.

RESPONSE: We greatly appreciate the Reviewer's expert assessment of our study and invaluable suggestions.

Major comments

On the analysis of memory formation:

The technique used here (like all the other techniques to register freely moving animals) to record neuronal activity has a time limitation: the activity

of a particular group of neurons can be followed over the hours but not over longer periods like days or weeks. Authors comment on this issue in line 423. However it should be noted that memory acquisition and consolidation are a processes that have many facets, many of them taking several hours or even days or weeks to be completed. Perhaps it is recommendable to make a mention to the temporal aspect of memory formation, specially considering that the whole experiment was carried out in less than 24 h. In that sense, it would be reasonable to refer to “the initial stages” of memory formation rather than taking it as a finished process. Moreover, in the present work there are many citations related to memory in the references (~50% of the total) and it is even cited Squire’s classical paper (1986), so it is desirable to mention in the discussion the relation of present results in function of more traditional terminology of the field (particularly the evolution of memory traces over time).

RESPONSE: Thank you for raising these salient points, including the discrepancies between our and more traditional terminology. We have carefully revised the manuscript to clarify that we mainly investigated the “initial stages” of the consolidation process in this study. Indeed, distinguishing the initial stage of consolidation from the remaining process of systems consolidation is very important, which is reflected in the following addition to our manuscript:

The index theory assumes that a hippocampal index develops during memory acquisition—presumably due to the rapid synaptic changes taking place within the hippocampus³⁹. Consistently, the coactivation-contributing-ensembles in the hippocampus developed in an experience-dependent manner (Fig. 9a, b). By contrast, during memory acquisition, the coupled and non-coupled vCA1–PL5 ensemble pairs were similarly coactivated by shock presentations (Fig. 7b). Based on these observations, we hypothesise that the hippocampal ensembles developed through experience are non-selectively and weakly linked with cortical ensembles during the process of memory acquisition, then the subset of ensemble pairs are selected and stabilised later (Fig. 10). The recurring coactivation of the selected hippocampal-cortical ensemble pairs, associated with fast network oscillations in the involved regions (Fig. 4), may further invoke changes in the neocortical circuitry during the initial stage of memory consolidation, which in turn supports systems consolidation.

We also clarified that the entire consolidation process would take longer than our recording sessions and consequently, our current study cannot investigate the long-term changes due to technical limitations.

In turn, these fast changes underlying the initial stage of consolidation may drive slower changes that support systems consolidation, a process which may take

several days, weeks, or months. Because fear-memory induced with eyelid stimulation lasts for at least 6 days after the conditioning⁴⁹, the neuronal dynamics taking place across the consolidation process may be examined by adopting recently developed extra-long duration recording techniques^{50,51} in future studies.

In the abstract and the discussion, it is mentioned that the results are compatible with the “index theory”, this can certainly be the case but it cannot be ruled out the chance of the vHp being not a mere indexer but also a structure involved in the storage of the trace.

RESPONSE: We agree that our data cannot rule out the possibility that the vCA1 stores memory traces. We have clarified this point in the Discussion section:

Although we cannot rule out the possibility that the vCA1 stores memory contents, this hypothesis is consistent with the following observations; (1) the conditioning induced activation of vCA1 ensembles coupled with PL5 (Fig. 9b), (2) PL5 cells displayed increases in firing rates and modulation by SWRs between pre- and post-cond NREM (Fig. 9c, d), (3) coactivation of vCA1–PL5 coupled-ensemble-pairs hosted by SWRs (Figs. 4, 6, and 8) were enhanced during post-cond NREM (Figs 2, 3, and 7e), and (4) vCA1–PL5 ensemble coactivations reappeared during memory retrieval (Fig. 8 and Supplementary Fig. 21).

On the behavioral training:

There are no references to explain the type of training used here nor a validation of the parameters selected to train the animals (amount and intensity of the shocks, intensity (dB) of the cue, etc.). In the present work animals receive 12 trains of pulses of 4.6-5.1 mA and 32 (?) tones are used for extinction of the memory trace. For example, in a previous work by Hugues and colleagues (Synapse, 2006 Sep 15;60(4):280-7, doi: 10.1002/syn.20291) a similar methodology is used but less shocks of minor intensity are used (5 US presentations of 3.5 mA and they are extinguished by 15 CS presentations). It is a possibility that this stronger training is fundamental to observe the activity at the ensembles' level but it is not mentioned. It would be suitable to include more detailed data on this subject.

RESPONSE: As the Reviewer noted, the protocols vary across studies, thus we have compiled parameters for fear conditioning with eyelid stimulation used in previous studies, which were compared with those in the current study for the Reviewer's convenience.

	Blair et al 2005 J. Neurosci.	Hugues et al., 2006 Synapse	Johansen et al et al., 2010 Nat. Neurosci.	Uematsu et al., 2017 Nat. Neurosci.	Current study
Rat strain	Long–Evans	Wistar	Long–Evans	Long–Evans	Long–Evans
Cue type	White noise pips	2.5 kHz tone	White noise pips	5-kHz tone pips	5-kHz tone pips
Cue duration	20 s	30 s	20 s	20 s	30 s
Cue loudness	80 dB	75 dB	70 dB	74 dB	74 dB
Shock intensity	2.5 mA	3.5 mA	2.0 mA	2.0 mA	4.6-5.1 mA
Pulse width	2 ms	Not known	2 ms	2 ms	2 ms
Pulse frequency	10 Hz	5 Hz	6.66 Hz	7 Hz	8 Hz
Shock duration	2 s	1.6 s?	2 s	1 s	2 s
Number of pulses per shock	20	8	13?	7	16
Number of cue-shock pairing	16	5	16	12	12
Number of cue presentation for extinction	N.A.	15	NA	20	40

The table shows that our protocol is mostly similar to those of previous studies, although the shock intensity used in the current study appears to be greater than in the others. As the Reviewer pointed out, this increased intensity of the shock pulses may enhance inter-regional ensemble coactivation. We have now discussed the possibility of “tone-place memory” engrams as the Reviewer suggested in the following comments.

This finding implies that association learning may invoke inter-regional coactivation. However, whether an aversive experience without association would also induce inter-regional coactivations remains unclear. Additionally, it is unknown whether the association of two neutral stimuli induces inter-regional coactivation. Further, the shock intensity used in this study was greater than that in previous studies (see Methods for more details), which raises the possibility that strong aversive experiences enhance the inter-regional coactivation. It is also possible that the combination of a strong aversive experience and consecutive exposure to different chambers in the retention sessions with a short interval generalised fear memories to multiple contexts or generated a mixture of “tone-place” memory traces⁵⁸. Future studies using multi-regional recordings from animals that experience aversive stimuli without associative cues, and animals that

perform associative learning tasks without aversive stimuli or with milder aversive stimuli, would address these questions. These experiments would represent a valuable direction for future studies.

We also added reference for the cue duration and number of pairings to the Methods section. Additionally, it is now clearly mentioned that the intensity of stimulation was greater than in previous studies.

The intensity of the electrical pulse was greater than in previous studies that used eyelid stimulations as US^{49,61-63}.

After a training session in the CFC, subjects are tested and the context elicits a >50% freezing response. In the cue retention test there is also a high freezing response. Though rats are later exposed to an extinction session, it appears to be insufficient to extinguish the initial memory trace because high percentages of freezing are still observed in the extinction test. Another possibility is that subjects are forming a mixed “tone-place” engram, given the strong saliency of the initial training, and this is generalized in the context retention exposure to the chamber B that is followed by Cue retention and extinction in chamber A. A similar phenomenon is described in Sachella et. al (2021, <https://doi.org/10.1101/2020.07.12.197319>).

RESPONSE: Thank you for this valuable suggestion. Indeed, we cannot exclude the possibility that a combination of strong shock pulses and exposure to the two chambers sequentially may generalise memory by generating a memory engram similar to the recently proposed mixed “tone-place” engram. Although we acknowledge that this issue is scientifically important, investigating this possibility would require conducting a series of newly designed and well-controlled experiments, which appear to be beyond the scope of this manuscript. Thus, we now explicitly discuss this possibility mentioned by the Reviewer and cite the pre-print suggested, but leave the issue as an open question, as mentioned in response to the previous comment.

Related to this issue, we realised that the description of the experimental procedure was not completely accurate. The original version of the manuscript mentioned that the cue-retention/extinction sessions immediately followed the context-retention test, however, there was an ~3-min interval for changing the equipment setting, during which the rats stayed in their homecages. Although 3 min is considerably shorter than the intervals used between other sessions (2.5–2.6 hours), we regret the inaccurate wording in the original manuscript. The Methods section was revised as follows:

Context-retention sessions were followed by cue-retention/extinction sessions with a short interval during which the rats were kept in the homecage (2.3–3.8 min,

median = 3.1 min [n = 15 rats]), and other behavioural sessions were separated by 2.5–2.6-h rest/sleep sessions in the homecages (Fig. 1d).

On the ensembles:

Supplementary figure 3 is important to understand the amount of ensembles were distinguished from the total number of registered cells. The graphs and numbers are not very clear to follow and they are key to the paper.

RESPONSE: The figure legend in the original version reported the proportions of positively or negatively modulated cells and ensembles which were expressed as the height of the bar plots. In addition to this redundant information, the original legend had some errors in pointing cells and ensembles in the figure. We apologise for the errors. Now the legend reads as follows:

Proportion of ensemble-participating-cells (cells that were members of at least one ensemble; **a**; n = 200/127/359 cells for BLA/vCA1/PL5, respectively) and proportion of ensembles (**b**; n = 60/38/93 for BLA/vCA1/PL5, respectively) whose firing rates/activation strength were significantly modulated by freezing behaviour (left) or cue presentation (right) during cue-retention/extinction sessions. The significance of the modulation of each cell firing/ensemble activation was determined by comparing the distribution of modulation strength in surrogates obtained using bin label shuffling (refer to Methods for detailed information). Horizontal bars indicate a chance level (5%). The numbers of cells and ensembles are superimposed on the bars.

Minor comments

There are a few omissions in the methodology that would be necessary to include:

- how long is the recovery period after surgery? Are animals housed singly? Are the implants chronically fixed or is it an acute intervention (i.e.: does the surgery take place the same day of the experiments or are there recovery days before the beginning of the recordings?).

RESPONSE: We implanted all wires and probes chronically during surgery. Behavioural test and electrophysiological recording were performed after a 3–15 day recovery period, during which animals were fully recovered from the surgery and the positions of silicon probes were slowly adjusted to the target areas. The animals were singly-housed after being transferred to the animal care facility of the University. This information has been added to the Methods section.

In the Animals subsection:

Fifteen singly-housed male Long–Evans rats ...

The title of the Surgery subsection was revised to:

Surgery for chronic implants of wires and probes

In the Fear conditioning subsection:

After the recovery period (3 – 15 days, median = 7 days [n = 15]), during which animals were fully recovered from the surgery and the silicon probes were slowly moved downward to the target areas (typically 70 – 280 $\mu\text{m}/\text{day}$), we performed behavioural tests with electrophysiological recordings.

- the experimental protocol is hard to follow (both in the methods and in Figure 1d). It would be useful to clarify it a little more. The use of the word “session” is a bit confusing because at a first read, sessions appear to happen in different days.

RESPONSE: We have clarified that all behavioural sessions took place on the same day in the Methods section.

The experiment consisted of five behavioural sessions: baseline, conditioning, context-retention, cue-retention/extinction, and retention-of-extinction (Fig. 1c, d). All sessions took place on the same day (Fig 1d).

Also, in the legend of Fig 1d

Electrophysiological recording, behavioural sessions, and micro-lesions were performed on the same day.

- Graphs in Figure 1 c and d are expressed in seconds or minutes making them not clear enough. Particularly in Figure 1d, it would be recommendable to include a separate timeline expressed in hours to show the elapsed time between training, testing and extinction.

RESPONSE: Thank you for this suggestion. Indeed, inconsistency of measurement units across panels make the timeline unclear. To clarify the timeline, the time is now consistently expressed in minutes across the panels in Fig 1. We further added additional timelines in Fig 1c, showing time elapsed from the beginning of the recording, which is now clarified in the legend:

(c) Proportion of time spent in freezing in each behavioural session. The black lines and grey shaded areas indicate the mean and SE, respectively (n = 15 rats). Time from the session onset is shown on the x-axes and time from the beginning of the recording is superimposed beneath the x-axes, corresponding to the x-axis in (d).

Conclusion

The results shown here are very interesting. All the electrophysiology recordings and the analysis are robust and clearly explained. In case authors want to extend their studies, it would be interesting to analyze the freezing response induced by the initial training and evaluate how long does this memory last (only behavioral testing, no electrophysiological recordings) and if there is a mixed “tone-place” memory to achieve a deeper characterization of the behavior.

RESPONSE: Reportedly, fear-memory induced by eyelid stimulation lasts at least 6 days after conditioning (Ref 49; Johansen et al et al., 2010 Nat. Neurosci.). Based on similarity of the methods, we think it is reasonable to assume that fear memories induced with our protocol would last >6 days. We added this to the Discussion as follows:

Because fear-memory induced with eyelid stimulation lasts for at least 6 days after the conditioning⁴⁹, the neuronal dynamics taking place across the consolidation process may be examined by adopting recently developed extra-long duration recording techniques^{50,51} in future studies.

As mentioned above, we agree that it would be interesting to determine whether a mixed “tone-place memory” engram results from our procedure. We believe that pursuing this possibility is beyond the scope of this manuscript but is certainly an area of research that we intend to pursue in the near future. We are grateful for the Reviewer’s suggestion.

Reviewer #2 (Remarks to the Author):

Miyawaki and Mizuseki investigated interactions between neuron ensembles in the prefrontal cortex, amygdala, and ventral hippocampus during and after fear conditioning in rats. With large-scale multi-site electrophysiological recording, they identified that all three regions contained groups of neurons that fired together during fear conditioning as well as subsequent non-REM sleep. Notably, such ensemble co-activation in the mPFC was temporally coupled with that in the amygdala and hippocampus. The inter-region co-activation was detected more frequently after fear conditioning than the homecage period and reappeared during the retention test. In the amygdala and mPFC, but not the hippocampus, the co-activation was also detected before fear conditioning. Based on these findings, the authors argue that frightening experiences are rapidly encoded by a set of amygdala and prefrontal neurons with robust baseline connectivity and that these neurons are coupled with hippocampal neurons during subsequent sleep.

This study provides the first evidence for fine-time scale, inter-region interactions during the acquisition and consolidation of aversive associative memory. The experiments are technically challenging, and the authors' expertise in sophisticated analytical approaches are instrumental to uncovering cell assembly dynamics supporting fear memory. In addition, detailed characterizations of oscillations and spiking activity during sleep are of great value to researchers interested in neuronal mechanisms supporting consolidation during sleep. The analyses appeared to be conducted carefully with appropriate control measures. I only have a few concerns about data interpretation.

RESPONSE: We thank the Reviewer for the acknowledgement of the manuscript's importance and appreciate the invaluable suggestions for interpreting the presented data.

1) Based on the results in Fig. 7, the authors argue that “co-activation of BLA–PL5 coupled ensemble pairs and triple-activation of BLA–vCA1–PL5 coupled ensemble triplets form rapidly during memory acquisition, whereas co-activation of vCA1–PL5 coupled ensemble pairs develops during the sleep period that follows those experiences.” This argument does not convince me for two reasons. Firstly in Fig. 7b, the co-activation rate of vCA1-PL5 pairs was increased after the shock. Although the rate was not different from that of non-coupled pair, this finding suggests that vCA1-PL5 pairs (and other cell pairs) were rapidly formed during the conditioning.

RESPONSE: Thank you for pointing out this important issue. We agree with the Reviewer and we have investigated the details of coactivation dynamics within the conditioning sessions (Supplementary Fig. 17, and Supplementary Table 5). These analyses revealed that the coactivation of vCA1–PL5 coupled-ensemble-pairs slowly developed during the conditioning sessions, although the change was not statistically significant. These observations were added to the following description:

Contrastingly, coactivation event rates in inter-shock periods gradually increased in vCA1–PL5 coupled-ensemble-pairs, although the change was not statistically significant (Supplementary Fig. 17). Coactivation event rates in vCA1–PL5 ensemble pairs were not different between the coupled and non-coupled pairs, and both coupled and non-coupled vCA1–PL5 ensemble pairs were weakly but significantly activated at shock onset (Fig. 7b).

Secondly, vCA1-PL5 (and other) pairs possibly encode other task features such as the CS, conditioning chamber, and freezing. Therefore, the formation of vCA1-PL5 pairs may be triggered by these features rather than the shock.

The authors need to strengthen evidence if they want to claim that vCA1-PL5 ensembles develop only after learning.

RESPONSE: Thank you for this comment. We examined whether cue/freeze modulate ensemble coactivation and triple-activation during the conditioning sessions (Supplementary Fig. 19). The results were added to the manuscripts as follows:

Additionally, BLA–PL5 coupled-ensemble-pairs and BLA–vCA1–PL5 coupled-ensemble-triplets, but not vCA1–PL5 ensemble pairs, were weakly but significantly activated at cue onsets (Supplementary Fig. 19). However, neither the coactivation of ensemble pairs nor triple-activation of ensemble triplets was enhanced at freeze-onset (Supplementary Fig. 19).

We also assessed whether CCGs between instantaneous activation strengths of vCA1 and PL5 ensembles had significant peaks during conditioning, resulting in the detection of only one coupled pair with a significant peak (Fig. 7h). These results indicate that vCA1–PL5 coactivation, if any, was very weak during the conditioning sessions.

2) The model proposed in Fig. 10 needs additional clarifications.

a. The model gives me an impression that the pre-configured cell assembly includes local neurons in each region, and the binding of the assembly between the regions occurs during experiences. This does not fit with the data in Supplementary Fig. 16c showing the activation of BLA-PL neurons before the conditioning. Please clarify.

RESPONSE: Indeed, Supplementary Fig 16c (now Supplementary Fig. 18c in the revised version of the manuscript) shows that BLA ensembles coupled with PL5, compared with other ensembles, seem to be more strongly activated during pre-cond NREM. Further, coactivation event rates were slightly but significantly higher for BLA–PL5 coupled-ensemble-pairs compared with non-coupled pairs during pre-cond NREM (Fig. 7d). By contrast, a similar difference was not detected in vCA1–PL5 ensemble pairs (Fig. 7e). We incorporated these aspects in Fig. 10 by adding “weak inter-regional coactivation” represented by double lines.

b. I do not follow how the present data support that cell assembly in the vHPC functions as an index for a specific experience. The point of confusion is that the authors argue “co-activation of vCA1–PL5 coupled ensemble pairs develops during the sleep period.” According to the original theory, the index should be formed during experiences to facilitate the co-activation of other assemblies during subsequent consolidation processes.

RESPONSE: As mentioned in response to the above comment, we realised that both coupled and non-coupled vCA1–PL5 ensemble pairs weakly but significantly responded to the shock presentation, and later the coupled and non-coupled pairs were differentiated. We now hypothesise that the selection of coupled vCA1–PL5 pairs is an “initial step” of memory consolidation, which is required for later systems consolidation. We added these points in our manuscript as follows:

The index theory assumes that a hippocampal index develops during memory acquisition—presumably due to the rapid synaptic changes taking place within the hippocampus³⁹. Consistently, the coactivation-contributing-ensembles in the hippocampus developed in an experience-dependent manner (Fig. 9a, b). By contrast, during memory acquisition, the coupled and non-coupled vCA1–PL5 ensemble pairs were similarly coactivated by shock presentations (Fig. 7b). Based on these observations, we hypothesise that the hippocampal ensembles developed through experience are non-selectively and weakly linked with cortical ensembles during the process of memory acquisition, then the subset of ensemble pairs are selected and stabilised later (Fig. 10). The recurring coactivation of the selected hippocampal-cortical ensemble pairs, associated with fast network oscillations in the involved regions (Fig. 4), may further invoke changes in the neocortical circuitry during the initial stage of memory consolidation, which in turn supports systems consolidation.

c. The authors described their model very briefly in the first paragraph of the discussion. The paragraph is quite dense. I would suggest that they move this paragraph toward the end of the discussion and elaborate on how the present data fit with each prediction/argument.

RESPONSE: Thank you for the suggestion. We have added several instances related to the model shown in Fig. 10 in the Discussion. We also added a brief summary of the model in the last paragraph of the Discussion as follows:

In conclusion, our results suggest that the *de novo* inter-regional coordination of preconfigured local ensembles forms a new memory. The coordinated ensembles are reactivated together during short bouts of fast network oscillations during the initial stage of memory consolidation as well as during memory retrieval (Fig. 10).

To clarify the model presented in Fig 10, we further added an elaborated explanation to the legend:

Schematic summary of inter-regional ensemble coactivation development, suggesting that elements of memories are instantly embedded in preconfigured amygdalar and prefrontal cortical ensembles, whereas the inter-regional network that binds the distributed information develops in an experience-dependent manner (left two panels). The coactivation between hippocampal and prefrontal cortical ensembles is enhanced through an initial stage of memory consolidation occurring

during the subsequent sleep period (second right panel). During memory retrieval, prefrontal cortical ensembles are coactivated with amygdalar/hippocampal ensembles (rightmost panel). Inter-regional ensemble coactivations during the post-learning sleep and memory retrieval are associated with fast network oscillations (right two panels).

Reviewer #3 (Remarks to the Author):

The manuscript by Miyawaki & Mizuseki examined the inter-regional coactivations of preconfigured local ensembles following a fear conditioning task. The data show evidence of inter-regional activation during a brief period of sleep following learning. These coactivations relied on fast oscillations – Bla HFOs and vCA1 SWR contribute to BLA-PL5 and vCA1-PL5 ensemble coactivations during sleep. cRipples contributed to both coactivations. Coactivation in the BLA and PL5 were configured prior to memory but developed with experience in the vCA1. The authors also examined triple activation which was present during specific time windows and was also enhanced by SWRs, HFOs, and cRipples. The likely activation followed the order of vCA1, Bla and PL5. Interestingly, coactivated events were greater to shock presentation in BLA-PL5 pairs and triplets but not in vCA1-PL5 pairs. These findings are important for understanding how three critical regions involved in fear learning are coactivated or work together as part of a network to support learning and memory.

Uncovering how different neural areas coactivate to support memory is a fundamental question for system neuroscience. The data presented directly address this question. They are exciting and the analyses are thorough, clear and appropriate. Below I have a few minor comments/questions for the authors.

RESPONSE: We appreciate the Reviewer's kind praise, including describing the manuscript as exciting and the analyses as clear. We are also grateful for the Reviewer's constructive feedback which allowed us to improve our manuscript further.

The authors compare the presence of the coactivated pairs following fear learning against a neutral condition that equate for novelty. This is a great comparison which shows that the proportion of coactivated pairs changed between pre- and post-baseline NREM sessions but not the CCG peak height

of ensemble pairs. One question for the authors is whether they think that the CCG peak height of ensemble pairs would be modulated if the neutral experience consisted of associative learning between two neutral stimuli (the HPC has been implicated in sensory learning) or whether just mere exposure to an emotional event could do the same (shock exposure). While I am not suggesting these studies need to be done, I do feel the authors should discuss how specific the signal is to associative fear learning in particular.

RESPONSE: These are indeed very interesting questions that, unfortunately, cannot be answered in this study. We appreciate these points and proposed them as the direction of future studies in the Discussion section as follows:

In this study, we demonstrated that inter-regional coactivation develops after associative learning between neutral (i.e., cues) and aversive (i.e., shocks) stimuli, contrary to the presentation of neutral stimuli alone (Supplementary Fig. 8). This finding implies that association learning may invoke inter-regional coactivation. However, whether an aversive experience without association would also induce inter-regional coactivations remains unclear. Additionally, it is unknown whether the association of two neutral stimuli induces inter-regional coactivation. Further, the shock intensity used in this study was greater than that in previous studies (see Methods for more details), which raises the possibility that strong aversive experiences enhance the inter-regional coactivation. It is also possible that the combination of a strong aversive experience and consecutive exposure to different chambers in the retention sessions with a short interval generalised fear memories to multiple contexts or generated a mixture of “tone-place” memory traces⁵⁸. Future studies using multi-regional recordings from animals that experience aversive stimuli without associative cues, and animals that perform associative learning tasks without aversive stimuli or with milder aversive stimuli, would address these questions. These experiments would represent a valuable direction for future studies.

Cue retention seems to run into the extinction session as depicted in Fig 1 given the x-axis, but the methods suggests that there was an interval of 2.5-2.6h rest/sleep. It is unclear if cue retention and cue extinction were run in one session or not and if the former then how the distinction is made at the level of the analyses is also unclear. If indeed these sessions were separate then they should be depicted as such in the Figure.

RESPONSE: We performed cue-retention and extinction as one continuous behavioural session. In the analyses for Fig 8, data obtained during the entire cue-retention/extinction sessions were used. We have clarified this important point in the legend of Fig. 8.

For the analyses shown in (b-h), data obtained from the entire cue-retention/extinction session were used.

Also, we performed additional analyses where the cue-retention part of the session was separated from the extinction part of the session (Supplementary Fig. 22 and Supplementary Table 6). In these analyses, cue-retention and extinction were separated at the onset of the 9th cue, as presented in Supplementary Fig. 22a, and also clarified in the Methods:

The cue-retention test and extinction were performed as a continuous session, without separation, except for the analyses shown in Supplementary Fig. 22, where the cue-retention and extinction parts of the session were separated at the onset of the 9th cue.

Additional information within the text on how the cells were ‘modulated’ by fear (p.4). This is important to clarify because there is no information in the text nor in the supplemental figure legend that explains how modulation was examined.

RESPONSE: This information was only presented in the Methods section of the original version of the manuscript. We have added a brief explanation of the methods to the main text as follows:

First, by utilising bin label shuffling (see Methods for details), we assessed whether the firing of each ensemble-participating-cell (i.e., a cell that was a member of at least one ensemble determined during the conditioning session) was modulated by freezing behaviour/cue presentation.

The information was also added to the legend of Supplementary Fig. 4 (previously Supplementary Fig. 3) as well:

The significance of the modulation of each cell firing/ensemble activation was determined by comparing the distribution of modulation strength in surrogates obtained using bin label shuffling (refer to Methods for detailed information).

What is the functional significance of the vCA1-PL5 coactivation developing rapidly after the experience? There is a lot of discussion/speculation as to why and how this happens at the network level but less so at the functional level.

RESPONSE: We have carefully reviewed our results and now provide a more accurate description of the evolution of vCA1–PL5 coactivation as follows: The vCA1–PL5 ensemble pairs coactivated weakly but significantly at the shock onset, and later the coupled and non-coupled ensemble pairs were differentiated. Now we postulate that selecting the subset of vCA1–PL5 ensemble pairs as coupled ones during post-cond NREM

may trigger further changes in the cortical circuit that supports the initial stage of memory consolidation, which in turn supports further systems consolidation. These issues are now discussed as follows:

The index theory assumes that a hippocampal index develops during memory acquisition—presumably due to the rapid synaptic changes taking place within the hippocampus³⁹. Consistently, the coactivation-contributing-ensembles in the hippocampus developed in an experience-dependent manner (Fig. 9a, b). By contrast, during memory acquisition, the coupled and non-coupled vCA1–PL5 ensemble pairs were similarly coactivated by shock presentations (Fig. 7b). Based on these observations, we hypothesise that the hippocampal ensembles developed through experience are non-selectively and weakly linked with cortical ensembles during the process of memory acquisition, then the subset of ensemble pairs are selected and stabilised later (Fig. 10). The recurring coactivation of the selected hippocampal-cortical ensemble pairs, associated with fast network oscillations in the involved regions (Fig. 4), may further invoke changes in the neocortical circuitry during the initial stage of memory consolidation, which in turn supports systems consolidation.

The authors should add a table showing the number of co-active pairs and co-active triplets per animal.

RESPONSE: Thank you for this advice, however this information was summarised in Supplementary Table 4.

Reviewer #4 (Remarks to the Author):

This is a well conducted study that aims to address some timely key open questions in the learning and memory field such as how inter-regional neural ensemble interactions emerge, are reactivated during post-learning sleep, and relate to fast network oscillations. Towards that end, fear conditioning and fear memory retrieval were used as model paradigm in rats, while large scale electrophysiological recordings were performed in brain regions implicated in this behavior, including BLA, prelimbic cortex layer 5 (PL5) and ventral hippocampus CA1 (vCA1) followed by in depth analyses. The study uses a paradigm with a relatively short consolidation period (several hours) before the fear memory test and excludes analysis of the ensuing extinction sessions. The key novel findings that are generally well supported by the data are: Identification of inter-regional BLA-PL5 and vCA1-PL5 and BLA-vCA1-PL5 ensembles during learning, that are coactivated during

post-learning Non-REM sleep and associated with distinct fast network oscillations also during memory retrieval. Coactivation-participating ensembles developed with distinct time-courses and were preconfigured in BLA and PL, but emerged later in vCA1. Moreover, ensemble-coactivation contributing cells displayed faster firing rates, undermining the notion that higher excitability may determine allocation of neurons to memory traces.

RESPONSE: We greatly appreciate the Reviewer’s expert assessment of our study and invaluable suggestions, which have greatly strengthened our manuscript.

I have several comments and questions:

1. Generally, it may be worthwhile to streamline the manuscript to the core three regions for which ensembles, ensemble pairs and triplets were analyzed (BLA, PL-Layer 5 and vCA1).

RESPONSE: Thank you for this suggestion. We have carefully refined the manuscript and minimised the description of results in non-core regions. Additionally, the information for vCA1, BLA, and PL5 in Supplementary Tables 1 and 2 was moved to the first three columns— highlighting the core regions.

2. Although deducible from the tables, it would help to mention the number of animals for which individual analyses were done throughout.

RESPONSE: To clarify this point, we have added the number of rats for each region pair in Supplementary Table S3 as well in the text:

Significant changes in ensemble coactivation between pre- and post-cond NREM were observed only in BLA–PL5 pairs (examined in 7 rats) and vCA1–PL5 pairs (examined in 7 rats).

The information has been added for the triplets as well:

In rats with implants in all three regions (BLA, vCA1, and PL5; 6 rats), ...

It would also be helpful to know if the behavior was similar in the subsets of analyzed animals.

RESPONSE: Thank you for this suggestion. We repeated the same analyses for subsets of the animals and the results are shown in Supplementary Fig. 3, which confirmed that the proportions of time in freezing behaviour were similar across subsets, which is now mentioned in the main text.

The behaviour of subsets of rats with implants in the BLA and PL5 or in the vCA1 and PL5 did not differ significantly from the average of all the examined rats (Supplementary Fig. 3).

3. For the initial identification of FC ensembles, was the whole conditioning session used (c.f. Suppl Fig 16)?

RESPONSE: We apologise that the periods for ensemble detection were unclear in several figures. As described in the Methods, we used the entire conditioning session to detect FC ensembles. We have clarified this point in the legend for Fig. 7 as follows:

Ensembles were detected based on neuronal firings in entire conditioning sessions and divided into coupled and non-coupled ensemble pairs/triplets based on their coactivation/triple-activation in post-cond NREM.

As well in Supplementary Fig. 18 (previously Supplementary Fig. 16)

Ensembles were detected based on neuronal firings during entire conditioning sessions, then coupled regions of coactivation-participating-ensembles were determined in post-conditioning NREM.

Which part of the cue-retention/extinction session was used for analysis of modulated ensembles? It should be noted that changes in neural activity can precede behavioral changes (i.e. in extinction).

RESPONSE: We used the entire cue-retention/extinction sessions for the analyses in Fig. 8, which we clarified in the revised manuscript.

... we defined “reappeared-ensemble-pairs” as coupled-ensemble-pairs that had significant CCG peaks also during cue-retention/extinction sessions (peaks were assessed on CCGs in whole cue-retention/extinction sessions; Supplementary Table 6).

We also added the information for cue and freeze modulation analyses as follows:

... we calculated the proportion of reappeared-ensemble-pairs whose coactivations were significantly modulated by freezing behaviour or cue presentation during the entire cue-retention/extinction sessions (Fig. 8f).

And clarified the legend of Fig. 8 as well.

For the analyses shown in **(b-h)**, data obtained from the entire cue-retention/extinction session were used.

In addition to analyses shown in Fig. 8, we performed the same analyses in cue-retention (first 8 cue presentations) and extinction (the remaining cue presentations) separately, and our findings are presented in the new Supplementary Fig. 22 (Please also see our response on the next page).

Are the same or different ensembles modulated by cue and/or freezing?

RESPONSE: Thank you for this suggestion. We analysed how cue/freeze modulated ensembles overlapped but found no significant interactions.

The bar plots shown here summarise the overlaps in cue/freeze modulation of the ensembles. The black horizontal bars indicate chance levels of modulation, under the assumption that cue and freeze modulations were independent. The p-values of the assumption are superimposed on the top-right (obtained with the chi-square test). The numbers of ensembles are superimposed on the bars. Since these results seem to go beyond the scope of this paper, we did not include them. Should the reviewers and editors think these plots should be in the manuscript, we will happily add them.

It would be very interesting to see what happens to the FC-ensembles and pairs/triplets during extinction.

RESPONSE: As described in our response to the Reviewer's comments above, additional analyses for extinction were added (Supplementary Fig. 22). However the triplets were not included in these analyses because most of the triplets had ≤ 2 events during whole cue-retention/extinction sessions, as mentioned in the text. We found that the cue-onset coactivation of BLA–PL5 ensemble pairs and cue-onset activation of PL5 ensembles were specifically significant in the cue-retention—but not in the extinction—part of the session. By contrast, we observed no significant freeze-onset modulation in coactivations of ensemble pairs. This has been added to the manuscripts as follows:

The BLA–PL5 reappeared-ensemble-pairs displayed transient enhancement of coactivation events at cue onset (Fig. 8g). The cue-onset enhancement of coactivation of BLA–PL5 reappeared-ensemble-pairs was also significant when the analysis was restricted to the cue-retention—but not in the extinction—part of the session (Supplementary Fig. 22). The activation of PL5 ensembles participating reappeared-ensemble-pairs showed similar patterns (Supplementary Fig. 22). Such cue-onset modulation was not observed in the coactivation of vCA1–PL5

reappeared-ensemble-pairs (Fig. 8h and Supplementary Fig. 22). Alternatively, we did not observe an enhancement of ensemble activation/coactivation at freeze onset during the cue-retention or extinction parts of the session (Supplementary Fig. 22).

4. The identified ensembles/ensemble pairs also include inhibitory cells. What is their proportion and does this differ between regions and regional ensemble pairs?

RESPONSE: We compared the proportion of inhibitory members across coactivation-participating and other ensembles. The results are presented in the new Supplementary Fig. 24b. We also added a description of inhibitory cells as follows:

Inhibitory cells had more coupled regions compared with excitatory cells (Supplementary Fig. 24a), whereas the proportion of inhibitory cells among ensemble-coactivation-contributing cells were similar to those of other ensembles (Supplementary Fig. 24b).

5. The proportion of ensemble pairs is low in REM and does not differ between pre- and post-cond. REM. I was wondering if this has to do with the brevity of REM compared to NREM and the fact that generally NREM dominates the early sleep phases, also given the time evolution of ensemble activation?

RESPONSE: To determine if its short duration could explain the absence of coactivation in REM, we examined whether coactivations can be detected when using a subset of NREM data matching the duration of REM. The significant BLA–PL5 coactivation was also detected in the “duration matched” NREM (Supplementary Fig. 7d-f) and this trend was also detected in vCA1–PL5 pairs ($p = 0.078$), suggesting that the duration difference alone cannot explain the lack of coactivation in REM. These results have been added to the manuscript as follows:

Although REM was shorter than NREM in both pre- and post-cond homecage sessions (REM/NREM durations were $16.7 \pm 1.7 / 81.5 \pm 3.3$ min and $12.4 \pm 2.0 / 67.8 \pm 6.0$ min in pre- and post-cond sessions, respectively, mean \pm SE, $n = 15$ rats), the absence of coactivation during REM cannot be attributed solely to its short duration since BLA–PL5 ensemble coactivations during NREM were still significant when the NREM duration analysed was matched to REM duration, and the same analyses on vCA1–PL5 ensemble pairs showed an enhancing trend for the coactivation (Supplementary Fig. 7 d-f).

To assess whether lack of coactivation in REM may be due to a longer lag from the conditioning, we compared coactivation events in the first REM in post-cond homecage sessions versus those in NREM following the first REM. Even when using data from

NREM following the first REM, temporally more distant from the conditioning compared to the first REM, coactivation event rates were found to be significantly higher in the NREM than in REM (Supplementary Fig. 20a). Additionally, chunk shuffling analyses revealed that a significant proportion of ensemble pairs were coactivated in the NREM following the first REM (Supplementary Fig. 20b). By contrast, the proportion with significant coactivation was on the chance level in the preceding REM, except for coupled BLA–PL5 pairs, for which only 2 out of 38 pairs had significant coactivation events (Supplementary Fig. 20b). We added a paragraph to report these results as follows:

The coactivation/triple-activation event rates were found to decay with time (Fig. 7d-f). Because early phases of sleep are dominated by NREM, this raises the possibility that coactivations were not detected during REM (Supplementary Fig. 7) due to its temporal delay from the fear conditioning instead of the sleep state difference. To explore this possibility, we compared the coactivation/triple-activation event rates and proportion of pairs/triplets that coactivated significantly more often than chance level in the post-cond first REM *versus* following NREM (Supplementary Fig. 20). Among the coupled-ensemble-pairs/-triplets, coactivation/triple-activation event rates during the first REM were significantly lower than the subsequent NREM (Supplementary Fig. 20a). The proportion of significantly coactivated ensemble pairs during the first REM did not differ from chance level (0.5%), except for the BLA–PL5, where only 5.3% of the coupled-ensemble-pairs (2/38 pairs) were coactivated (Supplementary Fig. 20b). Contrastingly, a significant proportion of BLA–PL5 and vCA1–PL5 coupled-ensemble-pairs were coactivated during the NREM epoch following the first REM (36.8% [14/38 pairs] and 25.0% [3/12 pairs] for BLA–PL5 and vCA1–PL5 coupled-ensemble-pairs, respectively). For the vCA1–BLA–PL5 triplets, the proportion of triplets with significant triple-activation was not higher than chance level both during the first REM and the following NREM (Supplementary Fig. 20b). These results suggest that BLA–PL5 and vCA1–PL5 coactivation taking place during REM, if any, were significantly reduced compared with NREM, and this difference cannot be attributed solely to the longer temporal delay from the fear conditioning for REM *versus* NREM.

6. The temporal delays between BLA-PL5 and vCA1-PL5 ensemble pairs of 20-23 ms on average appear quite long for assuming monosynaptic connectivity and seem to be longer than those in the cited studies. Despite the well-established direct interconnectivity between these regions, there may be other explanations for coactivation of these ensembles.

RESPONSE: The word “monosynaptic” may be misleading. What we meant is, in any case, that the observed delays between BLA–PL5 and vCA1–PL5 ensemble activations are

consistent with the idea that the direct projections from the vCA1 and BLA to PL5 support inter-regional coactivation. Thus, we have revised the Results section as follows:

Considering that monosynaptic transmission latencies from the vHPC²⁶ and BLA²⁷ to the prefrontal cortex are approximately 15 ms and 8 ms, respectively, and a brain region could require tens of milliseconds to activate local ensembles through local circuit interactions in response to direct inputs received from an upstream region²⁸, these temporal delays between the ensemble activations suggest that direct projections from the vCA1/BLA to the PL5 support the inter-regional coactivations.

And the Discussion section:

... which is consistent with our notion that the observed time lags of ensemble coactivations reflect the effect of direct inter-regional projections (Fig. 3d).

7. The authors show shock-triggered averages of ensembles/ensemble pairs and triplets events during FC. I am not sure they can simply suggest “that these ensembles are related to memory of shock events” and that the vCA1-PL5 pairs are not involved in the initial acquisition. It would be nice to see cue-triggered averages during FC and freezing-triggered averages during retention (expanding on Figs 7g,h and 8f-h).

RESPONSE: Thank you for raising these important points. As mentioned in response to the Reviewer 3, we have carefully reviewed our results and now provide a more accurate description of the evolution of vCA1–PL5 coactivation as follows: The vCA1–PL5 ensemble pairs were coactivated weakly but significantly at the shock onset, and later the coupled and non-coupled ensemble pairs were differentiated. Accordingly, we revised our discussion extensively. Also, as mentioned in response to Reviewer 2, we have investigated the details of coactivation dynamics within the conditioning sessions (Supplementary Fig. 17 and Supplementary Table 5). These analyses revealed that the coactivation of vCA1–PL5 coupled-ensemble-pairs slowly developed during the conditioning sessions, although the change was not statistically significant. Please see our response to the Reveiwer 2 (on page 110-11) and response to the Reveiwer 3 (on page 15-16) for more information.

Further, cue triggered averages during fear conditioning have been added in a new Supplementary Fig. 19a-c. We found that BLA–PL5 coupled-ensemble-pairs showed weak but significant enhancement of coactivation at cue-onset, whereas such coactivation was not detected in vCA1–PL5 coupled-ensemble-pairs. These results were added to the manuscript as follows:

Additionally, BLA–PL5 coupled-ensemble-pairs and BLA–vCA1–PL5 coupled-ensemble-triplets, but not vCA1–PL5 ensemble pairs, were weakly but significantly activated at cue onsets (Supplementary Fig. 19). However, neither the

coactivation of ensemble pairs nor triple-activation of ensemble triplets was enhanced at freeze-onset (Supplementary Fig. 19).

We also added freeze triggered averages during cue-retention as Supplementary Fig. 22b. We did not see clear modulation of coactivation event rates at onset of freezing, which is now mentioned in the manuscript as follows:

... we did not observe an enhancement of ensemble activation/coactivation at freeze onset during the cue-retention or extinction parts of the session (Supplementary Fig. 22).

8. In Suppl Fig 19b, I did not understand which homecage session(s) this refers to. The NREM firing data seem to differ from those shown in Fig 9c (at least for BLA) and data seem only significant when wake, REM and NREM are considered together.

RESPONSE: For the analyses in Supplementary Fig. 19b (Now Supplementary Fig. 24c), we concatenated REM/NREM/wakefulness periods detected in all the homecage sessions. We clarified the analysed periods in the legend of Supplementary Fig. 24 as follows:

Mean firing rates of ensemble-coactivation-contributing excitatory/inhibitory cells and other excitatory/inhibitory cells during wakefulness, NREM, and REM. The wakefulness/NREM/REM periods detected in all five homecage sessions were concatenated to calculate mean firing rates for each brain state.

Label in middle graph BLA excitatory cells “coupled to BLA” should read “coupled to PL5”.

RESPONSE: Thank you for pointing this out, we have fixed this.

9. The results on firing modulation by SWRs and HFOs should be reworded including the conclusion statement, according to presented data/stats. Suppl Fig 20a,b suggests that this modulation is enhanced in excitatory BLA ensemble cells coupled to PL5 and not “largely unchanged”. For PL5 ensembles, there is a general increase in modulation of excitatory cells within coupled and other ensembles, and it looks like there is a specific increase in modulation of inhibitory cells in coupled ensembles.

RESPONSE: We agree with the Reviewer. We have incorporated the plots previously shown in Supplementary Fig. 20 into Fig. 9d, and revised the description in the main text as follows:

Because BLA–PL5 and vCA1–PL5 coactivations were mediated by HFOs and SWRs, we hypothesise that HFO- and SWR- modulations of

ensemble-coactivation-contributing cell firings are enhanced after fear conditioning. In the BLA, the firings of excitatory cells coupled with PL5 were enhanced by HFOs and SWRs more strongly in post- than in pre-cond NREM (Fig. 9d). Similar trends were observed for vCA1 excitatory cells, although the changes for cells coupled with the PL5 did not reach statistical significance ($p = 0.08$ and 0.053 for HFO- and SWR-modulation, respectively, $n = 19$ cells for each, WSR-test). Different from the BLA, SWR-modulation on inhibitory cells coupled with PL5 was also enhanced in the vCA1 (Fig. 9d). In the PL5, HFO- and SWR-modulation of excitatory cells and ensemble-coactivation-contributing inhibitory cells, but not of other inhibitory cells, was enhanced by fear conditioning (Fig 9d). Overall, these results suggest that the inter-regional ensemble coactivations after the conditioning are associated with the enhanced recruitment of ensemble-coactivation-contributing cells to HFOs and SWRs.

10. Could the authors based on their data speculate on the roles of inhibitory cells in ensembles?

RESPONSE: This is an interesting question. Inhibitory cells, compared with excitatory cells, had more coupled regions (Supplementary Fig 24a). This finding might imply that inhibitory cells are implicated in the inter-regional synchronization by coordinating the timings of ensemble activation. However, we have not added this possibility to the manuscript since it seemed too speculative. Of course, if the reviewers and editors think this should be a part of the manuscript, we will happily add it.

REVIEWERS' COMMENTS

Reviewer #1 (Remarks to the Author):

The authors introduced valuable changes in the manuscript that allow a clearer understanding of the methodology used in their study and further enlighten the significance of their findings. I am thankful for their dedication in the responses to the questions raised in the review.

In particular, references to previous works on the duration of a memory of similar training conditions are important to get a complete picture of the processes that might be taking place in the ensembles. Besides, the addition of Suppl. Fig 3 and the information included in the methods greatly contribute to clarify and appreciate the technical procedure (which is not something to underestimate, since there was a lot of hard work behind the experiments).

In all, changes in text, figures and supplementary figures are favorable and contribute with valuable information to the study (for instance, details on the co-activation properties and dynamics –line 319 onwards- or Suppl. Figures 17 and 22).

Regarding the discussion, the addition of new paragraphs and the modifications of this section greatly enrich it, particularly bridging the gap between habitual terms derived from memory and behavior to ephys techniques and neuronal activation (the explanation on memory index theory, etc.).

Once again, I would like to highlight the meaning of the results shown here in relation to memory formation and the allocation in a preexistent network.

Reviewer #2 (Remarks to the Author):

The authors fully addressed my comments and concerns. I do not have any further questions on this excellent work.

Reviewer #3 (Remarks to the Author):

The authors have addressed all my comments. The only outstanding comment is the lack of specifics regarding cell 'modulation'. The authors need to make this clear in the main text and methods - what are they referring to when they speak of 'modulation'.

Reviewer #4 (Remarks to the Author):

The authors have clarified my questions and concerns by additional analyses and text revisions. I have no further comments.

Of note, it may be worth checking for typos and wording, e.g. in some of the new Suppl Figs/panels (Suppl Figs 17 and 22 "ensemble") or in the text e.g. pg 18 line 401 I was not clear on the use of "Alternatively" here.

Thank you for the opportunity to submit our manuscript, which has undergone further revision. We wish to express our gratitude to the reviewers for their time and effort to assess our manuscript. Our point-by-point responses are below, with the reviewers' comments shown in bold Arial font and our responses in regular Times New Roman font. The revised text in the manuscript is indicated in blue font.

Reviewer #1 (Remarks to the Author):

The authors introduced valuable changes in the manuscript that allow a clearer understanding of the methodology used in their study and further enlighten the significance of their findings. I am thankful for their dedication in the responses to the questions raised in the review.

In particular, references to previous works on the duration of a memory of similar training conditions are important to get a complete picture of the processes that might be taking place in the ensembles.

Besides, the addition of Suppl. Fig 3 and the information included in the methods greatly contribute to clarify and appreciate the technical procedure (which is not something to underestimate, since there was a lot of hard work behind the experiments).

In all, changes in text, figures and supplementary figures are favorable and contribute with valuable information to the study (for instance, details on the co-activation properties and dynamics –line 319 onwards- or Suppl. Figures 17 and 22).

Regarding the discussion, the addition of new paragraphs and the modifications of this section greatly enrich it, particularly bridging the gap between habitual terms derived from memory and behavior to ephys techniques and neuronal activation (the explanation on memory index theory, etc.).

Once again, I would like to highlight the meaning of the results shown here in relation to memory formation and the allocation in a preexistent network.

Response: We greatly appreciate the reviewer's expert assessment of the work. We would like to thank the reviewer for acknowledging the importance of our study.

Reviewer #2 (Remarks to the Author):

The authors fully addressed my comments and concerns. I do not have any further questions on this excellent work.

Response: It is our great pleasure that our revisions addressed all the reviewer's concerns.

Reviewer #3 (Remarks to the Author):

The authors have addressed all my comments.

Response: We are happy to know that our revisions addressed the reviewer's concerns.

The only outstanding comment is the lack fo specifics regarding cell 'modulation'. The authors need to make this clear in the main text and methods - what are they referring to when they speak of 'modulation'.

Response: Thank you for pointing out the lack of specifics. Here we meant modulations of "cell firing". We have clarified this point as follows:

In the cue-retention/extinction sessions, firing rates of most BLA, vCA1, and PL5 ensemble-participating-cells were modulated by freezing behaviour/cue presentation (Supplementary Fig. 4a)

For further clarification, we added an explanation of how the modulation of cell firing is assessed:

Firing-rate modulations were measured as the ratio of firing rates within events of interest (e.g. HFOs or SWRs) versus mean firing rates across NREM (see Methods for detail).

Reviewer #4 (Remarks to the Author):

The authors have clarified my questions and concerns by additional analyses and text revisions.

I have no further comments.

Response: We would like to acknowledge the constructive suggestions that the reviewer had made.

Of note, it may be worth checking for typos and wording,

e.g. in some of the new Suppl Figs/panels (Suppl Figs 17 and 22 "ensemble") or in the text e.g. pg 18 line 401 I was not clear on the use of "Alternatively" here.

Response: Thank you for the detailed assessment of the paper. The typographical errors pointed out here were fixed. Additionally, we have carefully reviewed the wording again and edited the manuscript, including the use of “alternatively,” as pointed out by the reviewer:

On the other hand, we did not observe an enhancement of ...

In addition, we have checked the typographical errors in the figures and have had the manuscript files reviewed once again by experienced editors who are native English speakers.